# Inspecting discrepancy between multivariate distributions using half-space depth-based information criteria

**Pratim Guha Niyogi**                                                       *pguhaniyogi@umc.edu*
*Department of Data Science*
*J. D. Bower School of Population Health*
*University of Mississippi Medical Center*
*Jackson, MS 39216, USA*

**Subhra Sankar Dhar**                                                        *subhra@iitk.ac.in*
*Department of Mathematics and Statistics*
*Indian Institute of Technology*
*Kanpur, UP 208016, India*

**Reviewed on OpenReview:** *https://openreview.net/forum?id=i12oaLvy1I*

## Abstract

This article inspects whether a multivariate distribution differs from a specified distribution and tests the equality of two low-dimensional multivariate distributions. In this study, a graphical tool-kit using well-known half-space depth-based information criteria is proposed, which is a two-dimensional plot, regardless of the dimension of the data. The simple interpretability of the proposed graphical tool-kit motivates us to formulate test statistics to carry out the corresponding testing of hypothesis problems. It is established that the proposed tests based on the same information criteria are consistent. Moreover, the asymptotic distributions of the test statistics under contiguous/local alternatives are derived, which enables us to compute the asymptotic power of these tests. Empirical studies demonstrate that these tests outperform several existing methods across a range of distributions, which indicates that the proposed methodology is robust as well. The practical utility of the proposed tool-kit and tests is further illustrated through applications to two benchmark real-world datasets.

## 1 Introduction

In many scientific investigations, data are collected for multiple variables, and therefore, some techniques are needed to visualize and compare multivariate observations. In this spirit, this article addresses the problem of goodness-of-fit to validate the assumption of the underlying multivariate distribution, which is commonly encountered in many statistical data analyses. Moreover, we propose statistical tests to compare two multivariate probability distributions along with appropriate graphical tool-kits. It is needless to mention that such testing of hypothesis problems has a plethora of applications. For example, to understand the dynamics of the human physical activity patterns, the distribution of diurnal activity and the rest period is assumed to be a double Pareto distribution, and in order to validate it, a goodness-of-fit test is needed (Paraschiv-Ionescu et al., 2013). In psychology, the study of maternal depression and examination of maternal and infant sleep throughout the first year of life is an example of a two-sample testing problem (Newland et al., 2016).

Let us now formulate the problem with notation. Consider the data $\mathcal{X} = \{\mathbf{X}_1, \cdots, \mathbf{X}_n\}$ associated with unknown distribution $F$, and $F_0$ is the specified distribution function on $\mathbb{R}^d$ ($d \geq 1$), where $d$ is finite and much smaller than the sample size $n$. We now want to test that $H_0 : F = F_0$ against $H_1 : F \neq F_0$, and a few more technical assumptions on $F$ and $F_0$ will be stated at appropriate places. Next, the two-sample

problem for comparing multivariate distributions can also be formulated in the following way. Suppose that $\mathcal{X} = \{\mathbf{X}_1, \cdots, \mathbf{X}_n\}$ and $\mathcal{Y} = \{\mathbf{Y}_1, \cdots, \mathbf{Y}_m\}$ are two independent datasets associated with two unknown multivariate distributions $F$ and $G$, respectively. Here, we now want to test $H_0 : F = G$ against $H_1 : F \neq G$. For $d = 1$, i.e., in the case of univariate data, the aforesaid problems have already been investigated in many articles. For example, the readers may refer to Kolmogorov-Smirnov (KS) (Daniel, 1990), Cramér-von Mises (CvM) (Anderson, 1962), Anderson-Darling (AD) (Anderson & Darling, 1952) and a few more tests. In this context, for comparing univariate distributions, a few graphical tool-kits have also been used in literature for a long period of time, such as the quantile-quantile plot (Gnanadesikan, 2011; Chambers et al., 2018), the Lorenz curve (Lorenz, 1905).

For $d > 1$, one may look at AD test (Paulson et al., 1987), CvM test (Koziol, 1982), Doornik-Hansen (DH) test (Doornik & Hansen, 2008), Henze-Zirkler (HZ) test (Henze & Zirkler, 1990), Royston (R) test (Royston, 1992) and a series of $\chi^2$-type tests (for example McCulloch (M), Nikulin-Rao-Robson (NRR), Dzhaparidze-Nikulin (DN) (Dzaparidze & Nikulin, 1975)). These tests validate whether the data are obtained from a multivariate normal distribution or not. For two-sample multivariate problems, similar studies have been investigated by Chen & Friedman (2017) and see a few references therein. Besides, like univariate data, there have been a few attempts to compare two multivariate distributions in two-dimensional plots (see, for example, Dhar et al. (2014)).

In this article, we also study similar hypothesis problems based on the difference of some functions derived from the center-outward ranking (Tukey, 1975), which is known as data-depth (for details see Liu et al. (1999)), and the proposed test procedure is called data-depth discrepancy (DDD). The main idea of the discrepancy measure DDD is that it is large if and only if the null hypothesized assumption on the parent distribution is likely to be false (in a goodness-of-fit test) or the samples are likely to be from different distributions (in a two-sample test). Strictly speaking, in this work, the DDD is defined based on the $L_2$ and $L_\infty$ distances between two relevant data-depth functions for KS and CvM test statistics, respectively. However, to measure the DDD, one may use other suitable distance functions (such as $L_p, p \geq 1$ distance) in principle.

## 1.1 Contribution

This article proposes DDD based on Tukey's half-space depth (Tukey, 1975) since under some regularity conditions, it uniquely determines a uniformly absolutely continuous distribution with compact support under minimal assumptions. In addition, this can be computed using freely available packages in various statistical software such as R (R Core Team, 2025). In order to carry out the test, we replace the distribution function by half-space depth in the KS and CvM type test statistics for the goodness-of-fit and the two-sample tests, respectively. To summarize, the following are the main contributions of this article.

- First, we define a discrepancy measure based on half-space depth information criteria, which can be used to create an alternative graphical tool-kit to visualize the disparity of two distribution functions.

- Second, we propose two test statistics based on the proposed discrepancy measure for both goodness-of-fit and two-sample testing procedures.

- Third, most of the existing test statistics are available for the normality test only; our testing procedure provides a computationally feasible unified solution for any underlying distribution for any dimension.

- Fourth, we have theoretically shown that the proposed tests are consistent, i.e., the power of the tests tends to one as the sample size tends to infinity. Moreover, the asymptotic distributions of the test statistics under the contiguous alternative are derived, which enables us to compute the asymptotic power of the tests under contiguous alternatives.

- Fifth, a new graphical tool-kit based on the information criteria associated with half-space depth is introduced here with proper theoretical justification, where the diagram related to our proposed graphical tool-kit lies on the two-dimensional plane irrespective of the dimension of the data.

- Sixth, this half-space depth-based graphical tool-kit and the tests have potential applications related to modern Machine Learning problems. Exemplary, for out-of-distribution detection (OOD), the half-space depth-based methodologies perform reasonably well since half-space depth-based estimators can achieve a high breakdown point. For instance, the half-space based location estimator's breakdown point can achieve $\frac{1}{3}$ (see Propositions 3.2 and 3.3 in Donoho & Gasko (1992)), and this large value of breakdown point indicates that the graphical tool-kit and tests based on half-space depth proposed have good performance against the outliers. For this purpose, see the results obtained for *gilgais* data analysis in Section 6.2 as this dataset has many outliers or influential observations (see Figure 7).

### 1.2 Organization

The rest of the paper is organized as follows. In Section 2, we briefly review the different well-known notions of data-depth, the usefulness of Tukey's half-space depth. Section 3 is dedicated to the proposed methods in statistical testing for goodness-of-fit and two-sample situations. First, we establish the heuristic for the graphical tool-kit, and in the latter part of Section 3, we proceed with the formal definition of DDD, concluding with two multivariate tests based on data-depth. In Section 4, we investigate the asymptotic properties of the proposed testing procedures. The finite sample performance is presented in Section 5. Finally, in Section 6, we implement the proposed test on two popular benchmark datasets. Section 7 concludes the article with a discussion. Appendix contains all technical details and mathematical proofs. The codes of all numerical studies are made available upon request.

## 2 Some preliminaries

For a given multivariate data cloud $\mathcal{X} = \{\mathbf{X}_1, \cdots, \mathbf{X}_n\}$, a point $\mathbf{x}$ in the same Euclidean space becomes the representative of $\mathcal{X}$ through the function $D_{\mathcal{X}}(\mathbf{x})$, which measures how 'close' $\mathbf{x}$ is to the center of $\mathcal{X}$. Mathematically, a function, viz., depth function, will be a bounded and non-negative function of the form $D : \mathbb{R}^d \times \mathcal{F} \to \mathbb{R}$, where $\mathcal{F}$ is the class of all distributions on $\mathbb{R}^d$. Half-space depth is one of the depth functions that does not impose any moment conditions on the data, and it can characterize a certain family of distributions. The technical definition of the half-space depth is as follows. Let $F$ be a probability distribution on $\mathbb{R}^d$ for $d \geq 1$, and $\mathcal{H}$ be the class of closed half-spaces $H$ in $\mathbb{R}^d$. The Tukey's depth (or half-space depth) of a point $\mathbf{x} \in \mathbb{R}^d$ with respect to $F$ is defined by $D_F(\mathbf{x}) = \inf\{F(H) : H \in \mathcal{H}, \mathbf{x} \in H\}$. One may also look at $D_F(\cdot)$ in the following way. Let $\mathcal{S}^{d-1} = \{\mathbf{u} \in \mathbb{R}^d : \|\mathbf{u}\| = 1\}$ be the unit sphere of $\mathbb{R}^d$, then for $\mathbf{u} \in \mathcal{S}^{d-1}$ and $\mathbf{x} \in \mathbb{R}^d$, we consider the closed half-space passing through $\mathbf{x} \in \mathbb{R}^d$ as $H[\mathbf{x}, \mathbf{u}] = \{\mathbf{y} \in \mathbb{R}^d : \mathbf{u}^{\mathrm{T}}\mathbf{x} \leq \mathbf{u}^{\mathrm{T}}\mathbf{y}\}$. Now, Tukey's half-space depth with respect to the distribution function $F$ can be defined as $D_F(\mathbf{x}) = \inf_{\mathbf{u} \in \mathcal{S}^{d-1}} F(H[\mathbf{x}, \mathbf{u}])$.

The sample version of $D_F(\mathbf{x})$, which is denoted as $D_{\mathcal{X}}(\mathbf{x})$, based on the empirical distribution $\widehat{F}_n$ of the sample $\mathcal{X} = \{\mathbf{X}_1, \cdots, \mathbf{X}_n\}$ and is defined as $D_{\mathcal{X}}(\mathbf{x}) = \min_{\mathbf{u} \in \mathcal{S}^{d-1}} \frac{1}{n} \sum_{i=1}^{n} \mathbf{1}\{\mathbf{u}^{\mathrm{T}}\mathbf{X}_i \leq \mathbf{u}^{\mathrm{T}}\mathbf{x}\}$. Observe that, for $d = 1$, $D_F(x) = \min\{F(x), S(x)\}$, where $S(x) = 1 - F(x)$. Since $\widehat{F}_n(x)$ is the right-continuous empirical cumulative distribution function, we define $\widehat{S}_n(x) = 1 - \widehat{F}_n(x)$, and consequently, the sample version of half-space depth for univariate data becomes $D_{\mathcal{X}}(x) = \min\{\widehat{F}_n(x), \widehat{S}_n(x)\}$. Based on this definition, it can easily be observed that the depth function achieves the maximum at the median of the distribution and monotonically decays to zero when $x$ deviates from the median.

## 3 Data-depth discrepancy and associated statistical tests

In this section, we propose a data-depth discrepancy (DDD) describing a graphical tool-kit and associated tests. The DDD is defined in terms of the difference of the depth functions, and we begin this discussion in Section 3.1 for any probability distribution. We conclude this section with the motivation for defining a graphical tool-kit to visualize the dissimilarities of the distributions. We introduce new multivariate goodness-of-fit and two-sample tests based on half-space depth in Section 3.2.

### 3.1 Data-depth discrepancy (DDD)

Our goal is to construct statistical tests that can answer the following questions.

**Problem 1.** *(One-sample/Goodness-of-fit multivariate problem) For $d \geq 1$, let $\mathbf{X}$ be a d-dimensional random variable with distribution function $F$ on $\mathbb{R}^d$, where $\mathbf{X} = (X_1, \cdots, X_d)^T$. Suppose that $F_0$ is a pre-specified distribution function. Given observations $\mathcal{X} = \{\mathbf{X}_1, \cdots, \mathbf{X}_n\}$ independent and identically distributed (i.i.d.) from $F$, can we decide whether $F \neq F_0$?*

**Problem 2.** *(Two-sample multivariate problem) For $d \geq 1$, let $\mathbf{X}$ and $\mathbf{Y}$ be two random variables with distribution functions $F$ and $G$, respectively, where $\mathbf{X} = (X_1, \cdots, X_d)^T$ and $\mathbf{Y} = (Y_1, \cdots, Y_d)^T$. Given the two independent datasets $\mathcal{X} = \{\mathbf{X}_1, \cdots, \mathbf{X}_n\}$ and $\mathcal{Y} = \{\mathbf{Y}_1, \cdots, \mathbf{Y}_m\}$ independently and identically distributed from $F$ and $G$, respectively, can we decide whether $F \neq G$?*

To answer the above-mentioned questions, we propose the data-depth discrepancy (DDD) between two distributions $F$ and $G$ at $\mathbf{x} \in \mathbb{R}^d$ based on the data-depth as follows.

$$\text{DDD}(\mathbf{x}; F, G) = D_F(\mathbf{x}) - D_G(\mathbf{x}) \tag{1}$$

The sample version of these measures can be obtained by replacing $D$ with its empirical data-depth values based on the specific sample sizes. Moreover, the discrepancy between the empirical and theoretical distribution is defined as $\text{DDD}(\mathbf{x}; \mathcal{X}, F_0) = D_{\mathcal{X}}(\mathbf{x}) - D_{F_0}(\mathbf{x})$. Note that $\text{DDD}(\cdot; \cdot, \cdot)$ is a valid criterion to resolve Problems 1 and 2 as long as the depth functions (here, half-space depth) characterizes the distribution (i.e., $D_F(\mathbf{x}) = D_G(\mathbf{x})$ for all $\mathbf{x} \in \mathbb{R}^d \Leftrightarrow F = G$). Note that for $d = 1$, the characterization of the distribution using data-depth always holds. For details on the characterization of the distributions based on half-space depth for $d > 1$, see the following proposition.

**Proposition 1.** *(Struyf & Rousseeuw, 1999; Nagy et al., 2019; Laketa & Nagy, 2021) Let $P$ be a probability measure on $\mathbb{R}^d$, with $d > 1$, and let $F$ be the corresponding distribution function. If any one of the following holds, then the half-space depth characterizes the distribution.*

    *(a) $P$ is an finite atomic measure[1] in $\mathbb{R}^d$,*
    *(b) measures $P$ whose all Dupin's floating bodies[2] $D_{[\delta]}(P)$ exist with $\delta \in (0, 1/2)$.*

Proposition 1 indicates that the difference of the half-space depth values can provide a valid measure of the discrepancy, which is defined as DDD in equation 1. Now, we propose a graphical method where we plot DDD as a scatter plot across a horizontal axis and measure the deviation from the horizontal axis. Needless to say, the advantage of this graphical tool-kit is that it remains unaffected by the dimension of data. This graphical approach can be used for any distribution and based on any depth function, but to produce a valid graphical tool-kit for addressing the Problems 1 and 2, we continue our discussion with half-space depth since we can exploit the characterization properties discussed in Proposition 1. As a result, we can conclude that, if $\text{DDD}(\mathbf{x}; F, G) = 0$ for all $\mathbf{x}$, and graphically, if the majority of points are concentrated on the horizontal axis, we can conclude that the two underlying distributions are expected to be identical, where $F$ and $G$ belong to a certain family of distribution functions. The illustrative examples provide the different patterns of deviations from the horizontal line for both goodness-of-fit and two-sample problems when the null hypothesis is not true.

In the illustrative example in the following subsections, for a given sample $X_1, \cdots, X_n$, DDD is calculated for each $i$, and plotting it against enables a clear assessment of whether the discrepancy deviates from zero across datapoints. Each discrepancy corresponds to a specific datapoint $X_i$, plotting against provides direct visualization of how the discrepancy varies across the sample. We use the observed datapoints as test points because the objective is to assess discrepancies within the support of the underlying data-generating distribution. Evaluating data-depth at points drawn from the same distribution ensures that depth values are meaningful, stable, and comparable, since data-depth is inherently a relative notion defined with respect

---

[1]A measure $P$ on $\mathbb{R}^d$ is called finitely atomic if the support of $P$ consists of a finite point set.
[2]For any depth function $D_F$, define the upper level sets of $D_F(\cdot)$ as $D_\delta(P) = \{\mathbf{x} \in \mathcal{X}; D_F(\mathbf{x}) \geq \delta\}$, for $\delta > 0$. This forms a nested set, non-increasing with growing $\delta$ and $D_0(P) = \mathcal{X}$. Dupin's floating body of the measure $P$ at level $\delta > 0$ is defined as a convex set $D_{[\delta]}(P)$ such that each supporting half-space $H \in \mathcal{H}$ of carries mass $P(H) = \delta$, i.e., $D_{[\delta]}(P) = \cap\{H \in \mathcal{H} : P(H) > 1 - \delta\}$ for $\delta > 0$.

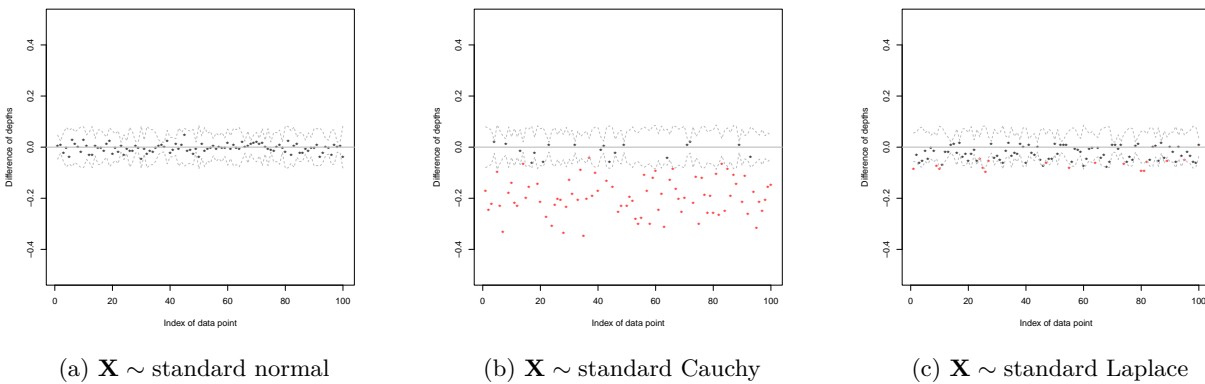

(a) $\mathbf{X} \sim$ standard normal      (b) $\mathbf{X} \sim$ standard Cauchy      (c) $\mathbf{X} \sim$ standard Laplace

Figure 1: The data-depth discrepancy plot for the goodness-of-fit problem where the specified distribution is bivariate normal. The dotted gray curves indicate the two-sigma limits of data-depth discrepancy. Points that are outside the two-sigma limits are color-coded in red.

to the empirical distribution. While data-depth discrepancy can, in principle, be evaluated at arbitrary test points, including those from external distributions, such choices may yield unstable or uninformative depth values, especially in regions with little data support. This can obscure interpretation and introduce variability unrelated to the hypothesis. Our approach follows standard depth-based inference practice by evaluating discrepancies at the observed sample points.

**Remark 1.** *Motivated by an anonymous referee's comment, here we would like to explain that considering the dataset itself as the training dataset, as we have done here, has only a negligible difference with choosing an arbitrary dataset when the size of the dataset is large enough. Consider $[0,1]$ on the horizontal axis and consider a discrete random variable $X$ such that $P[X = \frac{i}{n}] = \frac{1}{n}$ for $i = 1, \ldots, n$. Note that as $n \to \infty$, $X$ will weakly converge to a continuous random variable $Y$, where $Y$ follows uniform distribution over $[0,1]$ (see Example 25.3 in Billingsley (2012)). In view of this fact, when the size of the data is sufficiently large, considering the index of the dataset on the horizontal axis or taking any arbitrary observation $x$ on the horizontal axis has only a negligible difference. Besides, from a practical/computational point of view, computing half-space depth at the datapoint is less demanding than computing half-space depth at an arbitrary point. Therefore, combining all these aforementioned facts (i.e., limiting result as well as avoiding computational burden), we consider the dataset itself as the training dataset.*

### 3.1.1 Illustration of graphical tool-kit: Goodness-of-fit testing problem

For a given sample $\mathcal{X} = \{\mathbf{X}_1, \cdots, \mathbf{X}_n\}$ from an unknown distribution function $F$, we are interested in plotting $\mathrm{DDD}(\mathbf{x}; \mathcal{X}, F_0) = D_{\mathcal{X}}(\mathbf{x}) - D_{F_0}(\mathbf{x})$ with respect to indices of $\mathbf{x}$. Therefore, if $F = F_0$, then the value of $\mathrm{DDD}(\mathbf{x}; \mathcal{X}, F_0)$ will be/close to zero with respect to the observed data. We now consider three simulated datasets, each consisting of 100 i.i.d. observations. Three sets of samples are generated from (1) standard bivariate normal distribution, (2) standard bivariate Cauchy distribution, and (3) standard bivariate Laplace distribution, respectively. For all of them, we consider the bivariate normal distribution as the specified distribution $F_0$ with unknown mean $\boldsymbol{\mu}$ and dispersion $\boldsymbol{\Sigma}$, which are estimated from the sample using the sample mean vector and the sample variance-covariance matrix, respectively. We standardize the sample using these estimates and see the DDD-plots based on standardized data. The data-depth discrepancy plots for the three simulated samples are shown in Figure 1. In those plots, the dotted gray curves in the figures indicate the two-sigma limits[3] of DDD. It is clearly evident from the graphs that the specified distribution fits the data in Figure 1a reasonably well; moreover, the datapoints in this plot are clustered around the horizontal axis, and most of them are in the two-sigma limits. The other two plots, Figures 1b, 1c, indicate deviations from the horizontal axis, which immediately implies that the data are not from a normal distribution.

---

[3]Two sigma limits refer to boundaries set at two standard deviations $\sigma$ away from the mean $\mu$ in a probability distribution.

### 3.1.2 Illustration of graphical tool-kit: Two-sample testing problem

For given samples $\mathcal{X} = \{\mathbf{X}_1, \cdots, \mathbf{X}_n\}$ and $\mathcal{Y} = \{\mathbf{Y}_1, \cdots, \mathbf{Y}_m\}$, here we plot $\text{DDD}(\mathbf{x}; \mathcal{X}, \mathcal{Y}) = D_{\mathcal{X}}(\mathbf{x}) - D_{\mathcal{Y}}(\mathbf{x})$ for $\mathbf{x} \in \mathcal{X} \cup \mathcal{Y}$. If the two distributions are identical, then the value of $\text{DDD}(\mathbf{x}; \mathcal{X}, \mathcal{Y})$ will be zero/close to zero with respect to observed data. Here, we consider two simulated datasets to demonstrate the performance of our proposed graphical tool-kit. In our first problem, we simulate *sample-1* consisting of 100 i.i.d. observations generated from the trivariate normal distribution having zero mean and scatter matrix $\mathbf{\Sigma} = (\sigma_{i,j})_{i,j=1,2,3}$ with $\sigma_{1,2} = 0.9, \sigma_{1,3} = 0.2$ and $\sigma_{2,3} = 0.5$ ($F$) and *sample-2* consisting of 50 i.i.d. observations from the standard trivariate normal distribution denoted as ($G$). In the next problem, *sample-1* consists of 100 i.i.d. observations from the standard trivariate normal distribution ($F$), and *sample-2* consists of 50 i.i.d. observations from a trivariate skew-normal distribution ($G$) (Azzalini & Valle, 1996). The p.d.f. of the trivariate skew-normal distribution is given by $f(\mathbf{x}) = 2\phi_3(\mathbf{x}; \mathbf{\Omega})\Phi(\boldsymbol{\alpha}^{\mathrm{T}}\mathbf{x})$, where, $\boldsymbol{\alpha}^{\mathrm{T}} = \frac{\boldsymbol{\lambda}^{\mathrm{T}}\mathbf{\Psi}^{-1}\mathbf{\Delta}^{-1}}{\sqrt{1 + \boldsymbol{\lambda}^{\mathrm{T}}\mathbf{\Psi}^{-1}\boldsymbol{\lambda}}}$, $\Delta = diag(\sqrt{1-\lambda_1^2}, \sqrt{1-\lambda_2^2}, \sqrt{1-\lambda_3^2})$, $\boldsymbol{\lambda} = \left(\frac{\lambda_1}{\sqrt{1-\lambda_1^2}}, \frac{\lambda_2}{\sqrt{1-\lambda_2^2}}, \frac{\lambda_3}{\sqrt{1-\lambda_3^2}}\right)^{\mathrm{T}}$, and $\mathbf{\Omega} = \mathbf{\Delta}(\mathbf{\Psi} + \boldsymbol{\lambda}\boldsymbol{\lambda}^{\mathrm{T}})\mathbf{\Delta}$. Here $\phi_3(\mathbf{x}; \mathbf{\Omega})$ denotes the probability density function of a trivariate normal distribution with standardized marginals and the correlation matrix $\mathbf{\Omega}$, and $\mathbf{\Psi}$ is the distribution function of the standard univariate normal distribution. In this example, we have considered $\lambda_1 = \lambda_2 = \lambda_3 = 0.9$ and $\mathbf{\Phi} = \mathbf{I}_3$, identity matrix of dimension $3 \times 3$. The data-depth discrepancy plots for these two toy examples are shown in Figure 2. As before, the gray dotted curves in the figures indicate the two-sigma limits of DDD.

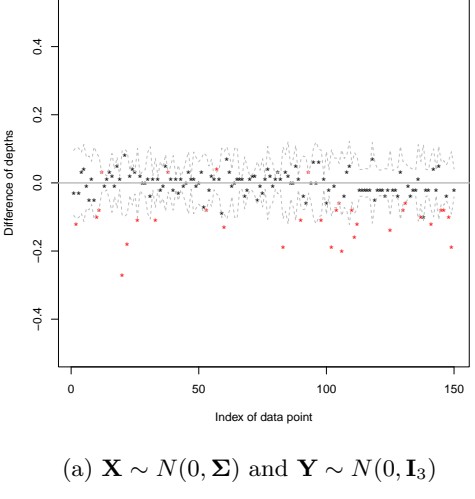
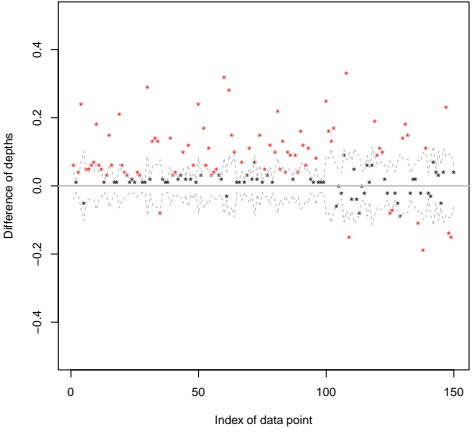

(a) $\mathbf{X} \sim N(0, \mathbf{\Sigma})$ and $\mathbf{Y} \sim N(0, \mathbf{I}_3)$

(b) $\mathbf{X} \sim N(0, \mathbf{I}_3)$ and $\mathbf{Y} \sim f(z)$ where $f(z)$ is the p.d.f. of skew-normal distribution

Figure 2: The data-depth discrepancy plot for the two-sample examples. The dotted gray curves indicate the two-sigma limits of data-depth discrepancy. Points that are outside the two-sigma limits are color-coded in red.

### 3.2 Associated statistical tests

In Problem 1, for given sample $\mathcal{X}$ of size $n$, we are interested to test $H_0 : F = F_0$ against $H_1 : F \neq F_0$, where $F_0$ is a pre-specified distribution function. Due to Proposition 1, for half-space depth function $D$, it is equivalent to test $H_0^* : D_F(\mathbf{x}) = D_{F_0}(\mathbf{x})$ for all $\mathbf{x} \in \mathbb{R}^d$ against $H_1^* : D_F(\mathbf{x}) \neq D_{F_0}(\mathbf{x})$ for some $\mathbf{x} \in \mathbb{R}^d$. The most well-known test statistics for comparing two distributions are the Kolmogorov–Smirnov (KS) test statistics and Cramér–von Mises (CvM) test statistics. These test statistics are based on certain differences between the empirical distribution function $F_n$ and the hypothesized distribution function $F_0$. Here, we propose alternative test statistics based on DDD defined in Section 3.1 so that we can perform the goodness-of-fit test for any arbitrary dimension of the data:

(1) Depth-based KS test statistic:

$$T_{\mathcal{X},F_0}^{\mathrm{KS}} = \sqrt{n} \sup_{\mathbf{x}\in\mathbb{R}^d} |\mathrm{DDD}(\mathbf{x};\mathcal{X},F)| = \sqrt{n} \sup_{\mathbf{x}\in\mathbb{R}^d} |D_{\mathcal{X}}(\mathbf{x}) - D_{F_0}(\mathbf{x})|,$$

(2) Depth-based CvM test statistic:

$$T_{\mathcal{X},F_0}^{\mathrm{CvM}} = n \int \mathrm{DDD}^2(\mathbf{x};\mathcal{X},F_0)dF_0(\mathbf{x}) = n \int (D_{\mathcal{X}}(\mathbf{x}) - D_{F_0}(\mathbf{x}))^2 dF_0(\mathbf{x}).$$

For testing $H_0$ against $H_1$, the null hypothesis will be rejected when the values of the test statistics are very large.

**Remark 2.** *Note that the true depth value $D_{F_0}(\cdot)$ is approximated by its empirical version in computing $T_{\mathcal{X},F_0}^{KS}$ as the empirical half-space depth is a uniformly consistent estimator of population half-space depth, which follows from Corollary 2.3 of Massé (2004). Consequently, for large enough samples, the approximated test statistic provided in the algorithm computes the actual value of the original test statistic arbitrarily well. This kind of idea is often used in computing the values of one sample of Kolmogorov-Smirnov or Cramér-von Mises test statistics.*

**Remark 3.** *Here, we want to discuss the computation of the supremum involved in $T_{\mathcal{X},F_0}^{KS}$. Observe that the main task is to compute $\sup_{\mathbf{x}\in\mathbb{R}^d} |D_{\mathcal{X}}(\mathbf{x}) - D_{F_0}(\mathbf{x})|$ (excluding normalization constant $\sqrt{n}$). We first approximate $\sup_{\mathbf{x}\in\mathbb{R}^d} |D_{\mathcal{X}}(\mathbf{x}) - D_{F_0}(\mathbf{x})|$ by $\sup_{||\mathbf{x}||\leq R} |D_{\mathcal{X}}(\mathbf{x}) - D_{F_0}(\mathbf{x})|$, where $R > 0$ is sufficiently large. In this step, we wanted to convey that the supremum over an unbounded set is approximated by a supremum over a large bounded set, and it makes sense since $R > 0$ can be arbitrarily large. Now, note that in view of Theorem 3.3 in Paindaveine & Bever (2018), $D_{\mathcal{X}}(.)$ and $D_{F_0}(.)$ are quasi concave functions with respect to (.), and in particular, they will be concave functions for elliptically symmetric distributions (e.g., multivariate Gaussian, multivariate Cauchy and many more) in view of Paindaveine & Bever (2018) (see Section 2, p. 3281). Hence, $|D_{\mathcal{X}}(\mathbf{x}) - D_{F_0}(\mathbf{x})|$ will be a convex function with respect to $\mathbf{x}$. Now, due to Bauer maximum principle (which states that a convex, upper semi-continuous function defined on a compact, convex set attains its maximum value at an extreme point of that set; see Theorem 1 in Kružík (2000)), we have*

$$\sup_{||\mathbf{x}||\leq R} |D_{\mathcal{X}}(\mathbf{x}) - D_{F_0}(\mathbf{x})| = \sup_{||\mathbf{x}||=R} |D_{\mathcal{X}}(\mathbf{x}) - D_{F_0}(\mathbf{x})|$$

*Afterwards, due to invariance of half-space depth under arbitrary transformation and monotonicity relative to deepest point (see (i) and (iii) of Theorem 2.1 in Zuo & Serfling (2000)), i.e., for any $R > 0$, $D_{F_{\mathbf{X}}}(\mathbf{x}) = D_{F_{R\mathbf{X}}}(R\mathbf{x})$, where $F_{\mathbf{X}}$ and $F_{R\mathbf{X}}$ denote the cumulative distribution functions of random vectors $\mathbf{X}$ and $R\mathbf{X}$, respectively, and $D_{F_{\mathbf{X}}}(R\mathbf{x}) = C_R D_{F_{\mathbf{X}}}(\mathbf{x})$ for some $C_R \in (0,\infty)$ (in particular, $C_R \in (0,1)$ if $R > 1$), we have*

$$\sup_{||\mathbf{x}||=R} |D_{\mathcal{X}}(\mathbf{x}) - D_{F_0}(\mathbf{x})| = \sup_{||\mathbf{x}||=1} |D_{\mathcal{X}}(R\mathbf{x}) - D_{F_0}(R\mathbf{x})| = \sup_{||\mathbf{x}||=1} |C_R D_{\mathcal{X}}(\mathbf{x}) - C_R D_{F_0}(\mathbf{x})|.$$

*Now, in view of the formula of empirical p-value provided in Step 1.6 in algorithm described in pages 7–8, it is equivalent to consider*

$$\sup_{||\mathbf{x}||=1} |D_{\mathcal{X}}(\mathbf{x}) - D_{F_0}(\mathbf{x})|$$

*as $C_R$ is getting cancelled from both sides. For this reason, to compute the half-space depth based on the KS statistic, the points are sampled from the boundary of the unit ball. Moreover, to generate the points from the set $\{\mathbf{x} : ||\mathbf{x}|| = 1\}$, we obtain the observations uniformly on $\{\mathbf{x} : ||\mathbf{x}|| = 1\}$, as the number of observations tends to infinity, it will be dense in $\{\mathbf{x} : ||\mathbf{x}|| = 1\}$. In this context, we would like to mention that one can achieve some other way to have dense observations on $\{\mathbf{x} : ||\mathbf{x}|| = 1\}$. For instance, one may consider a sequence of rational numbers (component-wise), whose norm equals one. Next, for computing $T_{\mathcal{X},F_0}^{CvM}$, note that*

$$n \int (D_{\mathcal{X}}(\mathbf{x}) - D_{F_0}(\mathbf{x}))^2 dF_0(\mathbf{x}) = \mathbb{E}_{\mathbf{X}\sim F_0}\{\sqrt{n}(D_{\mathcal{X}}(\mathbf{X}) - D_{F_0}(\mathbf{X}))\}^2.$$

*Hence, for a given observations $\mathbf{Y}_1, \cdots, \mathbf{Y}_M$ from $F_0$, using law of large number, $\frac{1}{M}\sum_{i=1}^{M}\{\sqrt{n}(D_{\mathcal{X}}(\mathbf{Y}_i) - D_{F_0}(\mathbf{Y}_i))\}^2$ can approximate the value of $T_{\mathcal{X},F_0}^{CvM}$ arbitrarily well.*

To obtain the empirical p-value based on the proposed test statistics $T_{\mathcal{X},F_0}$, where $T_{\mathcal{X},F_0}$ is either $T_{\mathcal{X},F_0}^{\mathrm{KS}}$ or $T_{\mathcal{X},F_0}^{\mathrm{CvM}}$, we consider a bootstrap procedure (Efron, 1982) that consists of the following steps.

---

**Algorithm 1** Approximation of Test Statistic and Empirical p-value for Problem 1

---

1: **Input:** Observed data $\mathcal{X} = \{\mathbf{X}_1, \ldots, \mathbf{X}_n\}$, null distribution $F_0$, Number of points on the unit balls $M$ and the number of bootstrap samples $B$

2: Generate $M$ points uniformly from the boundary of the $d$-dimensional unit ball; denote this set by $\mathcal{U}_1$

3: Generate $M$ i.i.d. samples from $F_0$; denote this set by $\mathcal{U}_2$

4: Compute half-space depth with respect to $\mathcal{X}$ and $F_0$

    (a) $\widetilde{T}_{\mathcal{X},F_0}^{\mathrm{KS}} \approx \sqrt{n} \max_{\mathbf{x} \in \mathcal{U}_1} |\mathrm{DDD}(\mathbf{x}; \mathcal{X}, F_0)|$

    (b) $\widetilde{T}_{\mathcal{X},F_0}^{\mathrm{CvM}} \approx n \cdot \frac{1}{M} \sum_{j=1}^{M} \mathrm{DDD}^2(\mathbf{x}_j; \mathcal{X}, F_0), \quad \mathbf{x}_j \in \mathcal{U}_2$

5: **for** $b = 1$ to $B$ **do**

6:     Generate bootstrap sample $\mathcal{X}^{*(b)} = \{\mathbf{X}_1^{*(b)}, \ldots, \mathbf{X}_n^{*(b)}\}$ from $F_0$

7:     Compute $\widetilde{T}_{\mathcal{X}^*, F_0}^{(b)}$ using Step 4 with $\mathcal{X}^{*(b)}$

8: **end for**

9: Use $\{\widetilde{T}_{\mathcal{X}^*, F_0}^{(b)} : b = 1, \ldots, B\}$ to approximate the null distribution of $T_{\mathcal{X},F_0}$

10: Compute empirical p-value:

$$\widehat{\mathbb{P}}_{H_0}\left(T_{\mathcal{X},F_0} > \widetilde{T}_{\mathcal{X},F_0}\right) = \frac{1}{B} \sum_{b=1}^{B} \mathbf{1}\left\{\widetilde{T}_{\mathcal{X}^*, F_0}^{(b)} > \widetilde{T}_{\mathcal{X},F_0}\right\}$$

11: **Output:** Test statistic $\widetilde{T}_{\mathcal{X},F_0}$ and empirical p-value

---

Next, we consider the two-sample $d$-dimensional problem for two independent samples $\mathcal{X} = \{\mathbf{X}_1, \cdots, \mathbf{X}_n\}$, and $\mathcal{Y} = \{\mathbf{Y}_1, \cdots, \mathbf{Y}_m\}$ where $\mathbf{X}_i$'s are independently distributed with distribution function $F$ and $\mathbf{Y}_i$'s are independently distributed with distribution function $G$. In Problem 2, we want to test $H_0 : F = G$ against $H_1 : F \neq G$, which is equivalent to test $H_0^* : D_F(\mathbf{x}) = D_G(\mathbf{x})$ for all $\mathbf{x} \in \mathbb{R}^d$ against $H_1^* : D_F(\mathbf{x}) \neq D_G(\mathbf{x})$ for some $\mathbf{x} \in \mathbb{R}^d$, where $D$ is the half-space depth. Similar to the goodness-of-fit test problem, the above equivalence is meaningful due to the characterization property of half-space depth. We construct the KS and CvM type test statistics based on $\mathrm{DDD}(\mathbf{x}; \mathcal{X}, \mathcal{Y})$ and obtain the following test statistics:

(1) Depth-based KS test statistic:

$$T_{\mathcal{X},\mathcal{Y}}^{\mathrm{KS}} = \sqrt{n+m} \sup_{\mathbf{x} \in \mathbb{R}^d} |\mathrm{DDD}(\mathbf{x}; \mathcal{X}, \mathcal{Y})| = \sqrt{n+m} \sup_{\mathbf{x} \in \mathbb{R}^d} |D_{\mathcal{X}}(\mathbf{x}) - D_{\mathcal{Y}}(\mathbf{x})|,$$

(2) Depth-based CvM test statistic:

$$T_{\mathcal{X},\mathcal{Y}}^{\mathrm{CvM}} = (n+m) \int \mathrm{DDD}^2(\mathbf{x}; \mathcal{X}, \mathcal{Y}) dH_{n,m}(\mathbf{x}) = (n+m) \int (D_{\mathcal{X}}(\mathbf{x}) - D_{\mathcal{Y}}(\mathbf{x}))^2 dH_{n,m}(\mathbf{x}),$$

where $H_{n,m}(\cdot)$ is the empirical distribution function based on the combined sample $\mathcal{X} \cup \mathcal{Y}$. Here, we also reject the null hypothesis for the large value of the test statistic.

**Remark 4.** *Similar to the one-sample problem discussed in Remark 3, in order to compute $T_{\mathcal{X},\mathcal{Y}}^{KS}$, one may follow the same approach described there. For instance, we here consider $d$-dimensional unit ball. Next, note that, $T_{\mathcal{X},\mathcal{Y}}^{CvM} = (n+m) \int (D_{\mathcal{X}}(\mathbf{x}) - D_{\mathcal{Y}}(\mathbf{x}))^2 dH_{n,m}(\mathbf{x}) = (n+m) \times \frac{1}{(n+m)} \sum_{j=1}^{(n+m)} (D_{\mathcal{X}}(\mathbf{y}_j) - D_{\mathcal{Y}}(\mathbf{y}_j))^2$, where $\mathbf{y}_j \in \mathcal{X} \cup \mathcal{Y}$ for $j = 1, \cdots, (n+m)$ as $H_{n,m}(\cdot)$ is the empirical distribution function based on the combined sample $\mathcal{X} \cup \mathcal{Y}$. This fact will enable us to compute $T_{\mathcal{X},\mathcal{Y}}^{CvM}$.*

The following algorithm provides an empirical p-value (Tibshirani & Efron, 1993) for Problem 2 based on any of the proposed test statistics $T_{\mathcal{X},\mathcal{Y}}$ where $T_{\mathcal{X},\mathcal{Y}}$ is either $T_{\mathcal{X},\mathcal{Y}}^{\mathrm{KS}}$ or $T_{\mathcal{X},\mathcal{Y}}^{\mathrm{CvM}}$.

---

**Algorithm 2** Approximation of Test Statistic and Empirical p-value for Problem 2

---

1: **Input:** Samples $\mathcal{X} = \{\mathbf{X}_1, \ldots, \mathbf{X}_n\}$, $\mathcal{Y} = \{\mathbf{Y}_1, \ldots, \mathbf{Y}_m\}$, Number of points on the unit balls $M$ and the number of bootstrap samples $B$

2: Generate $M$ points uniformly from the boundary of the $d$-dimensional unit ball; denote this set by $\mathcal{U}_1$

3: Compute half-space depth with respect to $\mathcal{X}$ and $\mathcal{Y}$

   (a) $\widetilde{T}_{\mathcal{X},\mathcal{Y}}^{\mathrm{KS}} = \sqrt{n+m} \max\limits_{\mathbf{x} \in \mathcal{U}_1} |\mathrm{DDD}(\mathbf{x}; \mathcal{X}, \mathcal{Y})|$

   (b) $\widetilde{T}_{\mathcal{X},\mathcal{Y}}^{\mathrm{CvM}} = \sum\limits_{j=1}^{n+m} \mathrm{DDD}^2(\mathbf{z}_j; \mathcal{X}, \mathcal{Y}), \quad \mathbf{z}_j \in \mathcal{X} \cup \mathcal{Y}$

4: Form pooled sample $\mathcal{Z} = \{\mathbf{X}_1, \ldots, \mathbf{X}_n\} \cup \{\mathbf{Y}_1, \ldots, \mathbf{Y}_m\}$

5: **for** $b = 1$ to $B$ **do**

6:     Draw a bootstrap sample of size $(n+m)$ with replacement from $\mathcal{Z}$

7:     Split into $\mathcal{X}^{*(b)} = \{\mathbf{X}_1^{*(b)}, \ldots, \mathbf{X}_n^{*(b)}\}$ and $\mathcal{Y}^{*(b)} = \{\mathbf{Y}_1^{*(b)}, \ldots, \mathbf{Y}_m^{*(b)}\}$ such that first $n$ as the $\mathcal{X}_b^*$ sample and the last $m$ are $\mathcal{Y}_b^*$

8:     Compute $\widetilde{T}_{\mathcal{X}^*,\mathcal{Y}^*}^{(b)}$ using Step 3 with $\mathcal{X}^{*(b)}$ and $\mathcal{Y}^{*(b)}$

9: **end for**

10: Use $\{\widetilde{T}_{\mathcal{X}^*,\mathcal{Y}^*}^{(b)} : b = 1, \ldots, B\}$ to approximate the null distribution of $T_{\mathcal{X},\mathcal{Y}}$

11: Compute empirical p-value:

$$\widehat{\mathbb{P}}_{H_0}\left(T_{\mathcal{X},\mathcal{Y}} > \widetilde{T}_{\mathcal{X},\mathcal{Y}}\right) = \frac{1}{B} \sum_{b=1}^{B} \mathbf{1}\left\{\widetilde{T}_{\mathcal{X}^*,\mathcal{Y}^*}^{(b)} > \widetilde{T}_{\mathcal{X},\mathcal{Y}}\right\}$$

12: **Output:** Test statistic $\widetilde{T}_{\mathcal{X},\mathcal{Y}}$ and empirical p-value

---

Here we would like to mention that $M = 1000$ and $B = 500$ are chosen based on the stability of the results. The larger values of $M$ and $B$ also give similar results with more computational time. In order to maintain computational cost under control with stabilized results, such choices of $M$ and $B$ are considered. Now, the asymptotic results for the proposed test statistics are studied in Section 4.

**Remark 5.** *For computational implementation, we use the `ddalpha` package to evaluate half-space depth. Specifically, `ddalpha` implements the random Tukey's half-space depth approximation proposed by Cuesta-Albertos & Nieto-Reyes (2008), which approximates the exact half-space depth using a finite number of randomly selected one-dimensional projections. By restricting attention to a finite collection of random directions, this approach avoids evaluating all possible projections and substantially reduces the computational burden, making it feasible in moderate- to high-dimensional settings. In our application, the sample size $n$ is considerably larger than the ambient dimension $d$ ($n \gg d$), under which the random projection approximation provides an accurate estimate of the population half-space depth. This is supported by standard uniform convergence results from empirical process theory and consistency properties of depth estimators; see Dyckerhoff (2004) and Briend et al. (2025). In contrast, the exact computation of half-space depth becomes computationally prohibitive as the dimension increases. For example, Chan (2004) showed that exact computation of half-space depth may require computational complexity on the order of $O(n^{d-1})$, which grows exponentially with dimension. The randomized approximation of Cuesta-Albertos & Nieto-Reyes (2008) is significantly more scalable. For completeness, we formally state below the computational complexity of the random Tukey's half-space depth algorithm used in our implementation.*

**Theorem 2.** *Let $\mathcal{X}_n = \{\mathbf{X}_1, \ldots, \mathbf{X}_n\} \subset \mathbb{R}^p$ be the sample of size $n$ in dimension $d$ and let $k \in \mathbb{N}$ be the number of the random directions samples uniformity from $\mathbb{S}^{d-1} = \{\mathbf{u} \in \mathbb{R}^d : \|\mathbf{u}\|_2 = 1\}$. The random Tukey's half-space depth of a query point $\mathbf{x} \in \mathbb{R}^d$ can be computed in time $T(n, d, k) = O(knd + kn \log n)$. When $n \geq d$ and $d < \log n$ (low-dimensional situation), $T(n, d, k) = O(kn \log n)$.*

## 4 Main results

### 4.1 Large sample statistical properties

To investigate the asymptotic properties of the proposed graphical tool-kit and tests, one first needs to assume a few technical conditions viz., (C1)-(C4), which are the following.

(C1) The distribution function $F$ satisfies $F(\partial H) = 0$ for all $H \in \mathcal{H}$, where $\partial H$ is the topological boundary of $H$.

(C2) $\mathcal{F} = \{F : F \text{ is a proper distribution function}\}$ is a totally bounded, and permissible subclass of $L_2(F)$ on $\mathcal{H}$, where $L_2(F) = \{f : \int_{\mathcal{H}} f^2(x)dF(x) < \infty\}$. Moreover, for each $\eta > 0$ and $\epsilon > 0$, there exists a $\delta > 0$ such that, $\limsup \mathbb{P}\left\{\sup_{\mathbb{D}(\delta)} |\mathcal{V}_n(f-g)| > \eta\right\} < \epsilon$, where $\mathbb{D}(\delta) = \{(f,g) : f,g \in \mathcal{F}, \rho_F(f-g) < \delta\}$ for the semi-norm $\rho_F$ on $\mathcal{H}$.

(C3) A minimal closed half-space at $\mathbf{x}$, viz. $H[\mathbf{x}]$ is uniquely defined if $D_F(\mathbf{x}) > 0$.

(C4) The distribution functions are either with finite support or absolutely continuous with continuous probability density function.

**Remark 6.** *Note that for an absolutely continuous distribution (e.g., a Gaussian distribution), Condition (C1) will be satisfied. For details, see Section 2 in Massé (2004). Next, (C2) is a well-known condition for empirical central limit theorems (see Chapter 5 in Pollard (1984)). In this context, observe that the class of Gaussian distribution indexed by the mean parameter, when the variance is fixed, satisfies Condition (C2). Thereafter, Condition (C3)implies the uniqueness of the half-space depth at a certain point, and Condition (C4) is satisfied by a wide range of distributions like multivariate Gaussian distribution, multivariate Cauchy distributions etc. Overall, Conditions (C1)–(C4) are common across the literature of statistical and machine learning related methodologies based on half-space depth, and if the data obtained from absolutely continuous distributions like Gaussian distributions, Conditions (C1)–(C4) will be satisfied, and hence, it has wide practical adaptability.*

We now state the results, which justify the usefulness of the proposed graphical tool-kits for sufficiently large sample sizes. See Theorems 3 and 4:

**Theorem 3.** *For every $\epsilon > 0$, define $\mathbb{C}_\epsilon(F, F_0) = \{(\mathbf{x}, y) : \mathbf{x} \in \mathbb{R}^d, |y - (D_F(\mathbf{x}) - D_{F_0}(\mathbf{x}))| < \epsilon\}$ and for a given sample $\mathcal{X} = \{\mathbf{X}_1, \cdots, \mathbf{X}_n\}$, define $\widehat{\mathbb{C}}(\mathcal{X}, F_0) = \{(\mathbf{x}, D_{\mathcal{X}}(\mathbf{x}) - D_{F_0}(\mathbf{x})) : \mathbf{x} \in \mathcal{X}\}$. Then, under the conditions (C1)-(C4), for every $\epsilon > 0$, we have $\lim_{n \to \infty} \mathbb{P}\left\{\widehat{\mathbb{C}}(\mathcal{X}, F_0) \subset \mathbb{C}_\epsilon(F, F_0)\right\} = 1$.*

**Theorem 4.** *For every $\epsilon > 0$, define $\mathbb{C}_\epsilon(F, G) = \{(\mathbf{x}, y) : \mathbf{x} \in \mathbb{R}^d, |y - (D_F(\mathbf{x}) - D_G(\mathbf{x}))| < \epsilon\}$ and for given independent samples $\mathcal{X} = \{\mathbf{X}_1, \cdots, \mathbf{X}_n\}$ and $\mathcal{Y} = \{\mathbf{Y}_1, \cdots, \mathbf{Y}_m\}$, define $\widehat{\mathbb{C}}(\mathcal{X}, \mathcal{Y}) = \{(\mathbf{x}, D_{\mathcal{X}}(\mathbf{x}) - D_{\mathcal{Y}}(\mathbf{x})) : \mathbf{x} \in \mathcal{X} \cup \mathcal{Y}\}$. Then, under the conditions (C1)-(C4), for positive finite number $\lambda = \lim_{\min(n,m) \to \infty} n/(n+m)$ and for every $\epsilon > 0$, we have $\lim_{\min(n,m) \to \infty} \mathbb{P}\left\{\widehat{\mathbb{C}}(\mathcal{X}, \mathcal{Y}) \subset \mathbb{C}_\epsilon(F, G)\right\} = 1$.*

**Remark 7.** *Theorems 3 and 4 show that for a large sample size, the points cluster around the horizontal axis if and only if the null hypothesis is true for both the goodness-of-fit and two-sample testing problem under some mild conditions.*

**Theorem 5** (Point-wise asymptotic properties). *Let $\star \in \{KS, CvM\}$. Under the conditions (C1)-(C4), the test based on test statistic $T^\star_{\mathcal{X}, F_0}$ for testing $H_0 : F = F_0$ against $H_1 : F \neq F_0$ is point-wise consistent, i.e., $\mathbb{P}_F\{T^\star_{\mathcal{X}, F_0} > s^\star_{1-\alpha}\} \to 1$ as $n \to \infty$ under $H_1$, where $s^\star_{1-\alpha}$ is such that $\lim_{n \to \infty} \mathbb{P}_{H_0}\left\{T^\star_{\mathcal{X}, F_0} > s^{(\star)}_{1-\alpha}\right\} = \alpha \in (0, 1)$.*

Theorem 5 indicates that the power of the proposed test for the goodness of fit problem will converge to the highest possible value, i.e., one, when the sample size is sufficiently large.

**Theorem 6** (Asymptotically uniform power). *Suppose that $\mathcal{X} = \{\mathbf{X}_1, \cdots, \mathbf{X}_n\}$ is a collection of i.i.d. random variables with distribution function $F$. For testing $F = F_0$ against $H_1 : F \neq F_0$, under the conditions*

*(C1)-(C4), the power of the test based on $T^{\star}_{\mathcal{X},F_0}$ tends to 1 uniformly over a sequence of alternatives $F_n$ satisfying $\sqrt{n}L_{\infty}(D_{F_n}, D_{F_0}) \geq \Delta_n$, where $\Delta_n \to \infty$ as $n \to \infty$ where $\star \in \{KS, CvM\}$.*

Not only for the fixed alternative, Theorem 6 indicates that the power of the proposed test for the goodness of fit problem will converge to the highest possible value, i.e., one, when the sample size is sufficiently large, as long as the sequence of alternatives satisfies a certain condition. We now describe the results related to pointwise and uniform consistency of the proposed two-sample testing procedure in a similar spirit to Theorem 5 and 6.

**Theorem 7.** *For $\star \in \{KS, CvM\}$, let $t^{(\star)}_{1-\alpha}$ be such that $\lim\limits_{\min(n,m)\to\infty} \mathbb{P}_{H_0}\left\{T^{\star}_{\mathcal{X},\mathcal{Y}} > t^{\star}_{1-\alpha}\right\} = \alpha \in (0,1)$. Further, suppose that $H(\mathbf{x}) = \lambda F(\mathbf{x}) + (1-\lambda)G(\mathbf{x})$. Then, under the conditions (C1)-(C4), $\mathbb{P}_{H_1}\left\{T^{\star}_{\mathcal{X},\mathcal{Y}} > t^{\star}_{1-\alpha}\right\} \to 1$, as $\min(n,m) \to \infty$. Moreover, for $\star \in \{KS, CvM\}$, the power of the test based on $T^{\star}_{\mathcal{X},\mathcal{Y}}$ tends to one, uniformly over a sequence of alternatives $F_{n,m}$ satisfying $\sqrt{n+m}L_{\infty}(D_{F_{n,m}}, D_{G_m}) \geq \Delta_{m,n}$, where $\Delta_{m,n} \to \infty$ as $\min(m,n) \to \infty$, where $G_m$ is the empirical distribution based on $\mathcal{Y}$.*

### 4.2 Asymptotic local power study

In Section 4.1, we have established that data-depth based KS and CvM tests are all asymptotically consistent, and therefore, a natural question is how the asymptotic power of the tests under local/contiguous alternatives (see Sidak et al. (1999); Dhar et al. (2019)). Let $\mathbb{P}_n$ and $\mathbb{Q}_n$ be the sequences of the probability measures defined on the sequence of probability spaces $(\Omega_n, \mathbb{A}_n)$. Then, $\mathbb{Q}_n$ is said to be contiguous with respect to $\mathbb{P}_n$ when $\mathbb{P}_n\{A_n\} \to 0$ implies that $\mathbb{Q}_n\{A_n\} \to 0$ for every sequence of measurable sets $A_n$. It is important to note that the sequence of set $A_n$ changes with $n$ along with the $\sigma-$field $\mathbb{A}_n$, and hence, it does not directly follow from the definition of contiguity that any distribution function $\mathbb{Q}_n$ is contiguous with respect to $\mathbb{P}_n$. In order to characterize the contiguity, Le Cam proposed some results which are popularly known as "Le Cam's Lemma". A consequence of Le Cam's first lemma is that the sequence $\mathbb{Q}_n$ will be contiguous with respect to the sequence $\mathbb{P}_n$ if $\log(\mathbb{Q}_n/\mathbb{P}_n)$ is an asymptotically normal random variable with mean $-\sigma^2/2$ and variance $\sigma^2$, where $\sigma$ is a positive constant. Moreover, the consequence of Le Cam's third lemma is that for $\mathbf{X}_n \in \mathbb{R}^d$, the joint distribution of $\mathbf{X}_n$ and $\log(\mathbb{Q}_n/\mathbb{P}_n)$ is distributed as multivariate normal with mean vector $(\boldsymbol{\mu}, -\sigma^2/2)^{\mathrm{T}}$ and covariance $\begin{pmatrix} \boldsymbol{\Sigma} & \boldsymbol{\tau} \\ \boldsymbol{\tau}^{\mathrm{T}} & \sigma^2 \end{pmatrix}$ under $\mathbb{P}_n$, then under the alternative distribution $\mathbb{Q}_n$, the asymptotic distribution of $\mathbf{X}$ is also a normal with mean $\boldsymbol{\mu} + \boldsymbol{\tau}$ and covariance $\boldsymbol{\Sigma}$.

In order to test $H_0 : F = F_0$, we consider the sequence of alternatives

$$H_n : F_n = \left(1 - \frac{\gamma}{\sqrt{n}}\right)F_0 + \frac{\gamma}{\sqrt{n}}H \tag{2}$$

for a fixed $\gamma > 0$ and $n = 1, 2, \cdots$. In terms of depth function, this testing of the hypothesis problem can equivalently be written as $H_0^* : D_F = D_{F_0}$, and the sequence of alternatives $H_n^* : D_{F_n} = (1 - \frac{\gamma}{\sqrt{n}})D_{F_0} + \frac{\gamma}{\sqrt{n}}D_H$. It follows from the Proposition 1 that the aforesaid hypothesis statement is valid for the half-space depth function. In the following remark, we have discussed the importance of the local/contiguous alternatives.

**Remark 8.** *Here we would discussion on the importance of the local/contiguous alternatives. The main significance of the local/contiguous alternatives is associated with finding the optimal test among the consistent tests. For example, suppose that there are two tests based on $T_n$ and $V_n$, both are consistent, i.e., the power of both tests based on and $T_n$ and $V_n$ converge to one (maximum possible value) when the sample size is infinity for fixed alternatives. Then the natural question would be which test is better among these two tests. To address this issue, the idea of local alternatives came to the literature. The local alternatives should be such that they are not the same as the null distribution but converge to the null as the sample size converges to infinity (see equation 2). Therefore, in other words, it will give us the power of the test in the local neighborhood of the null distribution, whereas the fixed alternatives provide us the power of the test when the distance between the alternative and null distributions remains fixed/constant (does not depend on the sample size n). As a result, we want to convey that the power property of the test under local alternatives was often overlooked. Further, note that the volume of the neighborhood depends on the choice of $\gamma$ (see*

equation 2). *Moreover, for $\gamma = 0$, we have $F_n = F_0$ for all $n$, and for $\gamma = \sqrt{n}$, we have $F_n = H$ for all $n$ (see equation 2). Therefore, in this sense, the work in Section 4.1 (i.e., under fixed alternative) is a special case of the work in Section 4.2 (i.e., under local alternatives) but not vice-versa.*

Now, Theorems 8 and 9 state the asymptotic power properties of the proposed test under contiguous alternatives.

**Theorem 8.** *Assume that $F_0$ and $H$ (see equation 2) have continuous and positive densities $f_0$ and $h$, respectively on $\mathbb{R}^d$ $(d \geq 2)$ such that $\mathbb{E}_{H_0}\left\{\frac{h(\mathbf{x})}{f_0(\mathbf{x})} - 1\right\}^4 < \infty$, and suppose that the optimal half-space depth associated to $\mathbf{x}$ is unique. In addition, conditions (C1)-(C4) hold. Then the sequence of alternatives is a contiguous sequence. Further, assume that $\mathcal{G}_1(\mathbf{x})$ is a random element associated with a Gaussian process with mean function zero and covariance kernel $F_0(H[\mathbf{x}_1] \cap H[\mathbf{x}_2]) - F_0(H[\mathbf{x}_1])F_0(H[\mathbf{x}_2])$, and $\mathcal{G}_1'(\mathbf{x})$ is a random element associated with a Gaussian process with mean function $-\gamma \mathbb{E}_{\mathbf{x} \sim H}\{D_{F_0}(\mathbf{x})\}$ and covariance kernel $F_0(H[\mathbf{x}_1] \cap H[\mathbf{x}_2]) - F_0(H[\mathbf{x}_1])F_0(H[\mathbf{x}_2])$. Under the alternatives described in equation 2, the asymptotic power of the test based on $T_{\mathcal{X},F_0}^{KS}$ is $\mathbb{P}_\gamma\left\{\sup_{\mathbf{x}}|\mathcal{G}_1'(\mathbf{x})| > s_{1-\alpha}^{(1)}\right\}$, where $s_{1-\alpha}^{(1)}$ is such that $\mathbb{P}_{\gamma=0}\left\{\sup_{\mathbf{x}}|\mathcal{G}_1(\mathbf{x})| > s_{1-\alpha}^{(1)}\right\} = \alpha$. Moreover, under the alternative hypothesis described in equation 2, the asymptotic power of the test based on $T_{\mathcal{X},F_0}^{CvM}$ is $\mathbb{P}_\gamma\left\{\int |\mathcal{G}_1'(\mathbf{x})|^2 dF_0(\mathbf{x}) > s_{1-\alpha}^{(2)}\right\}$, where $s_{1-\alpha}^{(2)}$ is such that $\mathbb{P}_{\gamma=0}\left\{\int |\mathcal{G}_1(\mathbf{x})|^2 dF_0(\mathbf{x}) > s_{1-\alpha}^{(2)}\right\} = \alpha$.*

Now consider the scenario of a two-sample problem. The null hypothesis is given by $H_0 : F = G$ against the sequences of alternatives

$$H_{n,m} : G = \left(1 - \frac{\gamma}{\sqrt{n+m}}\right) F + \frac{\gamma}{\sqrt{n+m}} H \tag{3}$$

for a fixed $\gamma > 0$ and $n, m = 1, 2, \cdots$, which is equivalent to test $H_0^* : D_F = D_G$ against the sequence of alternatives $H_{n,m}^* : D_G = \left(1 - \frac{\gamma}{\sqrt{n+m}}\right) D_F + \frac{\gamma}{\sqrt{n+m}} D_H$.

**Theorem 9.** *Assume $F$ and $H$ (see Statement equation 3) have continuous and positive densities $f$ and $h$, respectively on $\mathbb{R}^d$ $(d \geq 2)$ such that $\mathbb{E}_F\left\{\frac{h(\mathbf{x})}{f(\mathbf{x})} - 1\right\}^4 < \infty$. Suppose that the optimal half-space depth associated to $\mathbf{x}$ is unique and $\lim_{\min(n,m) \to \infty} \frac{n}{m+n} = \lambda \in (0,1)$, and in addition, conditions (C1)-(C4) hold. Then the sequence of alternatives is a contiguous sequence. Furthermore, assume that $\mathcal{G}_2(\mathbf{x})$ is a random element associated with a Gaussian process with mean function zero and covariance kernel $\{F(H[\mathbf{x}_1] \cap H[\mathbf{x}_2]) - F(H[\mathbf{x}_1])F(H[\mathbf{x}_2])\}/\lambda(1-\lambda)$ and $\mathcal{G}_2'(\mathbf{x})$ is a random element associated with a Gaussian process with mean function $\gamma\sqrt{\lambda/(1-\lambda)}\mathbb{E}_{\mathbf{u} \sim H}\{D_F(\mathbf{x})\}$ and covariance kernel $\{F(H[\mathbf{x}_1] \cap H[\mathbf{x}_2]) - F(H[\mathbf{x}_1])F(H[\mathbf{x}_2])\}/\lambda(1-\lambda)$. Under the alternatives described in Statement equation 3, the asymptotic power of the test based on KS is $\mathbb{P}_\gamma\{\sup_{\mathbf{x}}|\mathcal{G}_2'(\mathbf{x}) > t_{1-\alpha}\}$ where $t_{1-\alpha}^{(1)}$ is such that $\mathbb{P}_{\gamma=0}\{\sup_{\mathbf{x}}|\mathcal{G}_2(\mathbf{x})| > t_{1-\alpha}^{(1)}\} = \alpha$. Moreover, under the alternative hypothesis described in Statement equation 3, the asymptotic power of the test based on CvM is $\mathbb{P}_\gamma\left\{\int |\mathcal{G}_2'(\mathbf{x})|^2 d\mathbf{x} > t_{1-\alpha}^{(2)}\right\}$ where $\mathbb{P}_{\gamma=0}\left\{\int |\mathcal{G}_2'(\mathbf{x})|^2 d\mathbf{x} > t_{1-\alpha}^{(2)}\right\} = \alpha$.*

The results in Theorems 8 and 9 provide us the asymptotic power of the proposed tests under the local alternatives described in Statements equation 2 and equation 3, respectively. Using those results, we compute the asymptotic powers of the proposed tests for various choices of $\gamma$. In this study, we consider that $F_0$ is the standard bivariate normal distribution, and $H$ is the standard bivariate Laplace distribution. Figure 3 illustrates the summarized results, and it is clearly indicated by those diagrams that our proposed tests perform well for this example. For the sake of concise presentation, we are not reporting here the results for other choices of $F_0$ and $H$, and some other choices of dimension. Nevertheless, our preliminary investigation suggests that the tests perform well for alternative choices.

## 5 Finite sample level and power studies

In this section, we demonstrate the finite sample performance by reporting the size and power of the proposed tests obtained from multiple studies compared to other existing methods. Precisely speaking, we use `mvtnorm`

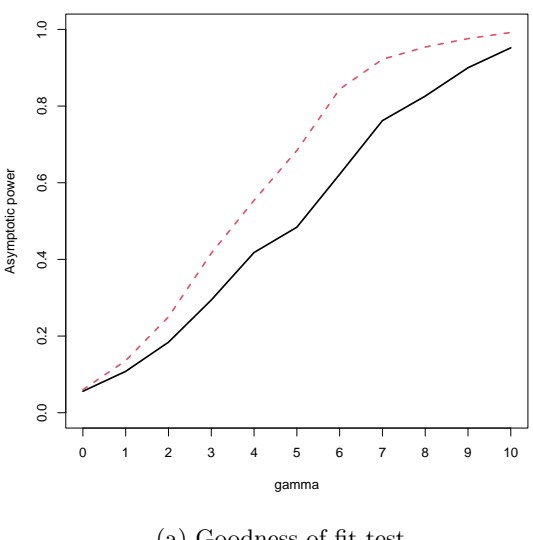 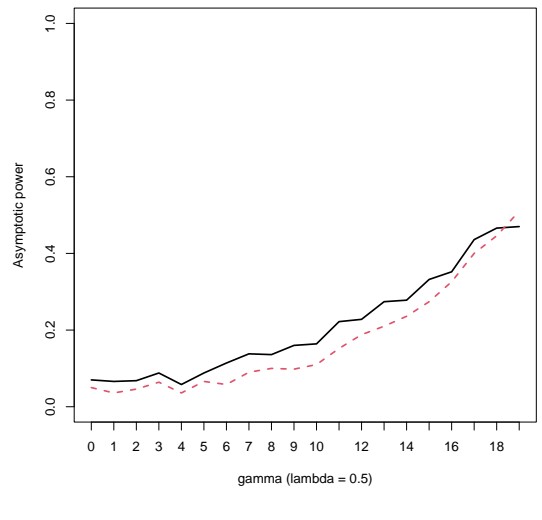

(a) Goodness-of-fit test         (b) Two-sample test

Figure 3: The asymptotic power for (a) goodness-of-fit test and (b) two-sample test with $\lambda = 0.5$ under contiguous alternative described in Statements equation 2 and equation 3, respectively, where $F_0$ is standard bivariate normal distribution and $H$ is the standard Laplace distribution. The black solid line indicates the results based on proposed depth-based KS test statistics, and the red dotted line indicates the results based on proposed depth-based CvM test statistics.

and `LaplacesDemon` to generate the data from multivariate normal, t, and Laplace distributions, respectively. Next, in order to compute the half-space depth, the package, viz., `ddalpha` is used. Next, in order to implement the other tests, `mvnTest` is used to run goodness-of-fit tests. All the simulation results in the following examples are based on 500 replications, and the critical value of the test is estimated by 500 bootstrap samples in each of the simulation runs. We also study the simulation results for cases with different dimensions, choosing $d \in \{2, 6, 10\}$. Furthermore, we denote $\boldsymbol{\mu} \in \mathbb{R}^d$ as a location parameter, and $\boldsymbol{\Sigma}$ is a $d \times d$ positive definite matrix as a scale parameter in those examples.

In the goodness-of-fit problem (see Problem 1), we assume that $F_0$ is a standard $d$-dimensional normal distribution. The choices of unknown distributions described in Problem 1 are listed below.

Model A.1. Standard $d$-variate normal distribution $N_d(\mathbf{0}_d, \mathbf{I}_d)$, where $\mathbf{0}_d = (0, \cdots, 0)^{\mathrm{T}}$ is the $d$-dimensional null vector, and $\mathbf{I}_d$ refers to the $d \times d$ identity matrix.

Model A.2. Mixture of the $d$-variate normal distribution with mixing probability 0.95, where the samples are taken from $0.95 N_d(\mathbf{0}_d, \mathbf{I}_d) + 0.05 N_d(2 \times \mathbf{1}_d, \mathbf{I}_d)$.

Model A.3. Mixture of the $d$-variate normal distribution with mixing probability 0.95, where the samples are taken as a from $0.95 N_d(\mathbf{0}_d, \mathbf{I}_d) + 0.05 N_d(\mathbf{0}_d, \boldsymbol{\Sigma})$. Here, $\boldsymbol{\Sigma}$ be a compound symmetric matrix with off-diagonal elements as 0.5 and diagonal elements as 1.

Model A.4. $d$-variate t-distribution with degrees of freedom 7, $t(\mathbf{0}_d, \mathbf{I}_d, 3)$.

Model A.5. Standard $d$-variate Laplace distribution $L(\mathbf{0}_d, \mathbf{I}_d)$ with p.d.f. $f(\mathbf{x}) = [\Gamma(d/2)/(2\Gamma(d)\pi^{d/2})] \times \exp(-\|\mathbf{x}\|)$, where $\mathbf{x} \in \mathbb{R}^d$.

Here note that Models A.2–A.5 are not too different than the Model A.1, i.e., the alternative models and the null model are close enough so that one can properly evaluate the performance of the tests. In this numerical study, we fix the sample size $n \in \{25, 50, 500\}$ and the nominal significance level $\alpha$ is assumed to be 0.05. We

Table 1: Goodness-of-fit test: Observed relative frequency for rejecting the null along with the computational time (in seconds for single iteration×100) in parentheses, $d = 2$. For Models A.2–A.5, the maximum value in each row is written in bold font.

| $n$ | KS.depth | CvM.depth | AD | CM | DH | HZ | R | McCulloch | NRR | DN |
|---|---|---|---|---|---|---|---|---|---|---|
| Model A.1. (**Null**) $H_0 : \mathbf{X} \sim N_2(\mathbf{0}_2, \mathbf{I}_2)$ against $H_1 : \mathbf{X} \sim N_2(\mathbf{0}_2, \mathbf{I}_2)$ | | | | | | | | | | |
| 25 | 0.036 (8) | 0.060 (8) | 0.040 (8) | 0.024 (8) | 0.054 (8) | 0.042 (2) | 0.056 (3) | 0.036 (3) | 0.036 (2) | 0.038 (3) |
| 50 | 0.050 (14) | 0.064 (15) | 0.052 (22) | 0.048 (21) | 0.054 (22) | 0.060 (05) | 0.050 (07) | 0.030 (04) | 0.056 (05) | 0.058 (06) |
| 500 | 0.054 (62) | 0.056 (61) | 0.036 (71) | 0.036 (68) | 0.062 (32) | 0.042 (39) | 0.056 (38) | 0.052 (41) | 0.044 (42) | 0.060 (44) |
| Model A.2. (**Alternative**) $H_0 : \mathbf{X} \sim N_2(\mathbf{0}_2, \mathbf{I}_2)$ against $H_1 : \mathbf{X} \sim 0.95N_2(\mathbf{0}_2, \mathbf{I}_2) + 0.05N_2(2\mathbf{1}_2, \mathbf{I}_2)$ | | | | | | | | | | |
| 25 | 0.712 (8) | **0.996** (8) | 0.202 (8) | 0.180 (8) | 0.140 (8) | 0.970 (2) | **0.996** (3) | 0.118 (3) | 0.134 (2) | 0.090 (3) |
| 50 | 0.918 (14) | **1.000** (15) | 0.252 (22) | 0.250 (21) | 0.204 (22) | 0.997 (05) | 0.998 (07) | 0.100 (04) | 0.154 (05) | 0.124 (06) |
| 500 | **1.000** (62) | **1.000** (61) | 0.808 (71) | 0.810 (68) | 0.732 (32) | **1.000** (39) | **1.000** (38) | 0.160 (41) | 0.834 (42) | 0.824 (44) |
| Model A.3. (**Alternative**) $H_0 : \mathbf{X} \sim N_2(\mathbf{0}_2, \mathbf{I}_2)$ against $H_1 : \mathbf{X} \sim 0.95N_2(\mathbf{0}_2, \mathbf{I}_2) + 0.05N_2(\mathbf{0}_2, \mathbf{\Sigma})$ | | | | | | | | | | |
| 25 | 0.470 (8) | **0.670** (8) | 0.062 (8) | 0.054 (8) | 0.052 (8) | 0.038 (2) | 0.060 (3) | 0.032 (3) | 0.044 (2) | 0.056 (3) |
| 50 | 0.564 (14) | **0.792** (15) | 0.072 (22) | 0.072 (21) | 0.054 (22) | 0.050 (05) | 0.058 (07) | 0.052 (04) | 0.058 (05) | 0.052 (06) |
| 500 | 0.612 (62) | **0.874** (61) | 0.074 (71) | 0.064 (68) | 0.050 (32) | 0.056 (39) | 0.048 (38) | 0.068 (41) | 0.056 (42) | 0.056 (44) |
| Model A.4. (**Alternative**) $H_0 : \mathbf{X} \sim N_2(\mathbf{0}_2, \mathbf{I}_2)$ against $H_1 : \mathbf{X} \sim t(\mathbf{0}_2, \mathbf{I}_2, 7)$ | | | | | | | | | | |
| 25 | 0.187 (8) | **0.279** (8) | 0.186 (8) | 0.134 (8) | 0.227 (8) | 0.156 (2) | 0.174 (3) | 0.116 (3) | 0.115 (2) | 0.062 (3) |
| 50 | 0.300 (14) | **0.524** (15) | 0.300 (22) | 0.273 (21) | 0.405 (22) | 0.282 (05) | 0.299 (07) | 0.186 (04) | 0.187 (05) | 0.117 (06) |
| 500 | 0.880 (62) | **1.000** (61) | 0.893 (71) | 0.866 (68) | 0.991 (32) | 0.851 (39) | 0.876 (38) | 0.753 (41) | 0.771 (42) | 0.534 (44) |
| Model A.5. (**Alternative**) $H_0 : \mathbf{X} \sim N_2(\mathbf{0}_2, \mathbf{I}_2)$ against $H_1 : \mathbf{X} \sim L(\mathbf{0}_2, \mathbf{I}_2)$ | | | | | | | | | | |
| 25 | 0.474 (8) | **0.652** (8) | 0.562 (8) | 0.492 (8) | 0.484 (8) | 0.530 (2) | 0.402 (3) | 0.254 (3) | 0.324 (2) | 0.224 (3) |
| 50 | 0.808 (14) | **0.914** (15) | 0.884 (22) | 0.852 (21) | 0.740 (22) | 0.818 (05) | 0.702 (07) | 0.610 (04) | 0.722 (05) | 0.536 (06) |
| 500 | 0.996 (62) | **1.000** (61) | 0.995 (71) | 0.987 (68) | 0.991 (32) | 0.997 (39) | 0.978 (38) | 0.943 (41) | 0.915 (42) | 0.882 (44) |

compare the proposed method, denoted as KS.depth and CvM.depth, respectively, with Anderson-Darling (AD), Cramér-von Mises (CvM), Doornik-Hansen (DH), Henze-Zirkler (HZ), Royston (R) tests and a series of $\chi^2$-type tests such as McCulloch (M), Nikulin-Rao-Robson (NRR), Dzhaparidze-Nikulin (DN) tests.

Tables 1, 2, and 3 describe Monte Carlo results for goodness-of-fit tests that represent the observed relative frequency for rejecting $H_0$ for $d = 2, 6$ and 10, respectively. Here, the observed relative frequency for rejecting the null for Model A.1 indicates the size of the test, and for Model A.2-A.5 represent the power of the test for different alternatives. Note that for Models A.2–A.5, in each sample size, i.e., in each row, the maximum values are written in bold font. If we increase the sample size $n$, the power of the test increases for all the methods, and the size of the test is more or less stabilized at the nominal significance level. If we increase the dimension of the data, the aforementioned statement holds true. The power of the proposed tests is consistently better for all the choices of alternative distributions. Moreover, for mixture distributions like Model A.2 and A.3, the competing testing procedures are worse, whereas the proposed methods KS.depth and CvM.depth perform satisfactorily. In terms of computational cost, note that the KS and the CvM tests based on half-space depth are not too demanding up to dimension ten, given the fact that they can achieve good power even when the alternatives are not too different from the null. Besides, another expected fact is that the computational time of all tests depend on the dimension of the data and the sample size of the data but does not depend on the data distribution, which is indicated by the reported values of computational time in Tables 1, 2, and 3.

In the two-sample test scenario (see Problem 2), we use sample sizes $(n, m)$ such that $\lambda = n/(n + m) \in \{0.3, 0.5, 0.8\}$ and fix $n \in \{50, 100, 300\}$. We generate samples from the following distribution family.

Model B. The first samples are taken from the standard multivariate distribution $N_d(\mathbf{0}_d, \mathbf{I}_d)$ and the second sample is taken from the $d$ variate normal $N_d(\mu \times \mathbf{1}_d, \mathbf{I}_d)$ where we chose $\mu \in \{0, 0.5, 01\}$.

Table 4 represents the observed relative frequency for rejecting $H_0$ based on the above model. The values on the table corresponding to $\mu = 0$ are the size of the test, and those to $\mu = \mu_0 \in \{0.5, 1\}$ represent the power

Table 2: Goodness-of-fit test: Observed relative frequency for rejecting the null, along with the computational time (in seconds for single iteration$\times$100) in parentheses, $d = 6$. For Models A.2–A.5, the maximum value in each row is written in bold font.

| $n$ | KS.depth | CvM.depth | AD | CM | DH | HZ | R | McCulloch | NRR | DN |
|---|---|---|---|---|---|---|---|---|---|---|
| Model A.1. (**Null**) $H_0 : \mathbf{X} \sim N_6(\mathbf{0}_6, \mathbf{I}_6)$ against $H_1 : \mathbf{X} \sim N_6(\mathbf{0}_6, \mathbf{I}_6)$ | | | | | | | | | | |
| 25 | 0.064 (23) | 0.032 (10) | 0.048 (22) | 0.048 (21) | 0.060 (23) | 0.040 (11) | 0.050 (09) | 0.044 (10) | 0.052 (09) | 0.070 (10) |
| 50 | 0.044 (26) | 0.054 (12) | 0.054 (25) | 0.050 (25) | 0.044 (27) | 0.064 (12) | 0.048 (13) | 0.048 (11) | 0.070 (12) | 0.074 (13) |
| 500 | 0.044 (73) | 0.040 (50) | 0.036 (75) | 0.034 (74) | 0.042 (76) | 0.048 (51) | 0.054 (52) | 0.036 (52) | 0.040 (52) | 0.046 (52) |
| Model A.2. (**Alternative**) $H_0 : \mathbf{X} \sim N_6(\mathbf{0}_6, \mathbf{I}_6)$ against $H_1 : \mathbf{X} \sim 0.95N_6(\mathbf{0}_6, \mathbf{I}_6) + 0.05N_6(2\mathbf{1}_6, \mathbf{I}_6)$ | | | | | | | | | | |
| 25 | 0.968 (23) | **0.980** (10) | 0.039 (22) | 0.039 (21) | 0.039 (23) | 0.324 (11) | 0.973 (09) | 0.039 (10) | 0.073 (09) | 0.078 (10) |
| 50 | 0.994 (26) | **1.000** (12) | 0.090 (25) | 0.080 (25) | 0.068 (27) | 0.932 (12) | 0.986 (13) | 0.046 (11) | 0.066 (12) | 0.070 (13) |
| 500 | **1.000** (73) | **1.000** (50) | 0.186 (75) | 0.176 (74) | 0.172 (76) | 0.988 (51) | **1.000** (52) | 0.096 (52) | 0.168 (52) | 0.160 (52) |
| Model A.3. (**Alternative**) $H_0 : \mathbf{X} \sim N_6(\mathbf{0}_6, \mathbf{I}_6)$ against $H_1 : \mathbf{X} \sim 0.95N_6(\mathbf{0}_6, \mathbf{I}_6) + 0.05N_6(\mathbf{0}_6, \mathbf{\Sigma})$ | | | | | | | | | | |
| 25 | 0.954 (23) | **0.956** (10) | 0.022 (22) | 0.022 (21) | 0.046 (23) | 0.049 (11) | 0.037 (09) | 0.046 (10) | 0.051 (09) | 0.066 (10) |
| 50 | 0.954 (26) | **0.984** (12) | 0.032 (25) | 0.024 (25) | 0.040 (27) | 0.060 (12) | 0.046 (13) | 0.044 (11) | 0.044 (12) | 0.056 (13) |
| 500 | 0.970 (73) | **1.000** (50) | 0.270 (75) | 0.240 (74) | 0.050 (76) | 0.136 (51) | 0.042 (52) | 0.126 (52) | 0.168 (52) | 0.122 (52) |
| Model A.4. (**Alternative**) $H_0 : \mathbf{X} \sim N_6(\mathbf{0}_6, \mathbf{I}_6)$ against $H_1 : \mathbf{X} \sim t(\mathbf{0}_6, \mathbf{I}_6, 7)$ | | | | | | | | | | |
| 25 | 0.730 (23) | **0.790** (10) | 0.668 (22) | 0.544 (21) | 0.586 (23) | 0.766 (11) | 0.480 (09) | 0.270 (10) | 0.408 (09) | 0.292 (10) |
| 50 | 0.956 (26) | **0.999** (12) | 0.994 (25) | 0.988 (25) | 0.956 (27) | 0.976 (12) | 0.902 (13) | 0.918 (11) | 0.976 (12) | 0.872 (13) |
| 500 | **1.000** (73) | **1.000** (50) | 0.998 (75) | 0.992 (74) | 0.991 (76) | 0.998 (51) | **1.000** (52) | 0.995 (52) | 0.999 (52) | 0.997 (52) |
| Model A.5. (**Alternative**) $H_0 : \mathbf{X} \sim N_6(\mathbf{0}_6, \mathbf{I}_6)$ against $H_1 : \mathbf{X} \sim L(\mathbf{0}_6, \mathbf{I}_6)$ | | | | | | | | | | |
| 25 | 0.656 (23) | **0.912** (10) | 0.828 (22) | 0.590 (21) | 0.568 (23) | 0.896 (11) | 0.438 (09) | 0.080 (10) | 0.400 (09) | 0.418 (10) |
| 50 | 0.910 (26) | **0.996** (12) | 0.992 (25) | 0.990 (25) | 0.942 (27) | 0.985 (12) | 0.854 (13) | 0.682 (11) | 0.966 (12) | 0.930 (13) |
| 500 | **1.000** (73) | **1.000** (50) | 1.000 (75) | 0.990 (74) | 0.998 (76) | 0.997 (51) | **1.000** (52) | 0.989 (52) | 0.995 (52) | 0.996 (52) |

Table 3: Goodness-of-fit test: Observed relative frequency for rejecting the null along with the computational time (in seconds for single iteration $\times$100) in parentheses, $d = 10$. For Models A.2–A.5, the maximum value in each row is written in bold font.

| $n$ | KS.depth | CvM.depth | AD | CM | DH | HZ | R | McCulloch | NRR | DN |
|---|---|---|---|---|---|---|---|---|---|---|
| Model A.1. (**Null**) $H_0 : \mathbf{X} \sim N_{10}(\mathbf{0}_{10}, \mathbf{I}_{10})$ against $H_1 : \mathbf{X} \sim N_{10}(\mathbf{0}_{10}, \mathbf{I}_{10})$ | | | | | | | | | | |
| 25 | 0.040 (51) | 0.040 (49) | 0.056 (54) | 0.062 (46) | 0.052 (41) | 0.042 (42) | 0.072 (47) | 0.048 (46) | 0.152 (45) | 0.190 (48) |
| 50 | 0.040 (63) | 0.018 (62) | 0.050 (66) | 0.048 (50) | 0.048 (51) | 0.054 (51) | 0.056 (52) | 0.052 (53) | 0.076 (52) | 0.100 (52) |
| 500 | 0.022 (88) | 0.014 (85) | 0.058 (84) | 0.070 (76) | 0.042 (74) | 0.064 (77) | 0.044 (79) | 0.050 (78) | 0.056 (77) | 0.066 (75) |
| Model A.2. (**Alternative**) $H_0 : \mathbf{X} \sim N_{10}(\mathbf{0}_{10}, \mathbf{I}_{10})$ against $H_1 : \mathbf{X} \sim 0.95N_{10}(\mathbf{0}_{10}, \mathbf{I}_{10}) + 0.05N_{10}(2\mathbf{1}_{10}, \mathbf{I}_{10})$ | | | | | | | | | | |
| 25 | **1.000** (51) | 0.954 (49) | 0.031 (54) | 0.023 (46) | 0.046 (41) | 0.084 (42) | **1.000** (47) | 0.046 (46) | 0.160 (45) | 0.168 (48) |
| 50 | 0.992 (63) | **1.000** (62) | 0.044 (66) | 0.044 (50) | 0.044 (51) | 0.330 (51) | **1.000** (52) | 0.064 (53) | 0.176 (52) | 0.174 (52) |
| 500 | **1.000** (88) | **1.000** (85) | 0.076 (84) | 0.068 (76) | 0.090 (74) | 0.982 (77) | **1.000** (79) | 0.078 (78) | 0.254 (77) | 0.252 (75) |
| Model A.3. (**Alternative**) $H_0 : \mathbf{X} \sim N_{10}(\mathbf{0}_{10}, \mathbf{I}_{10})$ against $H_1 : \mathbf{X} \sim 0.95N_{10}(\mathbf{0}_{10}, \mathbf{I}_{10}) + 0.05N_{10}(\mathbf{0}_{10}, \mathbf{\Sigma})$ | | | | | | | | | | |
| 25 | **0.698** (51) | 0.392 (49) | 0.578 (54) | 0.446 (46) | 0.082 (41) | 0.308 (42) | 0.070 (47) | 0.114 (46) | 0.274 (45) | 0.250 (48) |
| 50 | **0.870** (63) | 0.678 (62) | 0.312 (66) | 0.424 (50) | 0.192 (51) | 0.182 (51) | 0.076 (52) | 0.190 (53) | 0.287 (52) | 0.342 (52) |
| 500 | **1.000** (88) | 0.992 (85) | 0.623 (84) | 0.627 (76) | 0.031 (74) | 0.253 (77) | 0.069 (79) | 0.272 (78) | 0.376 (77) | 0.407 (75) |
| Model A.4. (**Alternative**) $H_0 : \mathbf{X} \sim N_{10}(\mathbf{0}_{10}, \mathbf{I}_{10})$ against $H_1 : \mathbf{X} \sim t(\mathbf{0}_{10}, \mathbf{I}_{10}, 7)$ | | | | | | | | | | |
| 25 | **0.406** (51) | 0.152 (49) | 0.292 (54) | 0.102 (46) | 0.378 (41) | 0.384 (42) | 0.282 (47) | 0.032 (46) | 0.180 (45) | 0.210 (48) |
| 50 | **0.990** (63) | 0.842 (62) | 0.975 (66) | 0.961 (50) | 0.964 (51) | 1.000 (51) | 0.912 (52) | 0.894 (53) | 0.942 (52) | 0.934 (52) |
| 500 | **1.000** (88) | **1.000** (85) | 0.998 (84) | 0.995 (76) | 0.993 (74) | 0.975 (77) | **1.000** (79) | 0.990 (78) | 0.991 (77) | 0.987 (75) |
| Model A.5. (**Alternative**) $H_0 : \mathbf{X} \sim N_{10}(\mathbf{0}_{10}, \mathbf{I}_{10})$ against $H_1 : \mathbf{X} \sim L(\mathbf{0}_{10}, \mathbf{I}_{10})$ | | | | | | | | | | |
| 25 | **0.742** (51) | 0.284 (49) | 0.738 (54) | 0.284 (46) | 0.456 (41) | 0.736 (42) | 0.358 (47) | 0.224 (46) | 0.418 (45) | 0.362 (48) |
| 50 | **0.878** (63) | 0.414 (62) | 0.834 (66) | 1.000 (50) | 0.858 (51) | 1.000 (51) | **0.878** (52) | 0.410 (53) | 0.872 (52) | 0.850 (52) |
| 500 | **1.000** (88) | **1.000** (85) | 0.996 (84) | 0.992 (76) | 0.997 (74) | 0.991 (77) | **1.000** (79) | 0.983(78) | 0.994 (77) | 0.996 (75) |

Table 4: Two-sample test, namely, KS.depth and CvM.depth: Observed relative frequency for rejecting the null based on Model B. The computational time (in seconds for single iteration $\times 100$) is provided in parentheses.

| | n | m | $\mu = 0$ (**Null**) | | $\mu = 0.5$ (**Alternative**) | | $\mu = 1$ (**Alternative**) | |
| --- | --- | --- | --- | --- | --- | --- | --- | --- |
| | | | KS.depth | CvM.depth | KS.depth | CvM.depth | KS.depth | CvM.depth |
| **$d = 2$** | | | | | | | | |
| | 50 | 117 | 0.056 (15) | 0.056 (14) | 0.914 (15) | **0.970** (14) | **1.000** (15) | **1.000** (14) |
| $\lambda = 0.3$ | 100 | 234 | 0.038 (26) | 0.048 (25) | **1.000** (26) | **1.000** (25) | **1.000** (26) | **1.000** (25) |
| | 300 | 700 | 0.034 (37) | 0.044 (35) | **1.000** (37) | **1.000** (35) | **1.000** (37) | **1.000** (35) |
| | 50 | 50 | 0.086 (11) | 0.052 (11) | 0.802 (11) | **0.844** (11) | **1.000** (11) | **1.000** (11) |
| $\lambda = 0.5$ | 100 | 100 | 0.044 (21) | 0.046 (20) | 0.988 (21) | **0.996** (20) | **1.000** (21) | **1.000** (20) |
| | 300 | 300 | 0.040 (29) | 0.052 (27) | **1.000** (29) | **1.000** (27) | **1.000** (29) | **1.000** (27) |
| | 50 | 13 | 0.086 (09) | 0.036 (07) | 0.392 (09) | 0.344 (07) | 0.888 (09) | **0.920** (07) |
| $\lambda = 0.8$ | 100 | 25 | 0.076 (13) | 0.056 (10) | 0.676 (13) | **0.758** (10) | 0.996 (13) | **0.998** (10) |
| | 300 | 75 | 0.072 (18) | 0.056 (16) | 0.996 (18) | **0.998** (16) | **1.000** (18) | **1.000** (16) |
| **$d = 6$** | | | | | | | | |
| | 50 | 117 | 0.056 (38) | 0.024 (36) | **0.976** (38) | 0.958 (36) | **1.000** (38) | **1.000** (36) |
| $\lambda = 0.3$ | 100 | 234 | 0.062 (53) | 0.028 (49) | **1.000** (53) | **1.000** (49) | **1.000** (53) | **1.000** (49) |
| | 300 | 700 | 0.060 (76) | 0.052 (71) | **1.000** (76) | **1.000** (71) | **1.000** (76) | **1.000** (71) |
| | 50 | 50 | 0.120 (27) | 0.004 (25) | **0.990** (27) | 0.884 (25) | **1.000** (27) | 0.854 (25) |
| $\lambda = 0.5$ | 100 | 100 | 0.068 (33) | 0.010 (27) | **1.000** (33) | **1.000** (27) | **1.000** (33) | **1.000** (27) |
| | 300 | 300 | 0.056 (41) | 0.042 (36) | **1.000** (41) | **1.000** (36) | **1.000** (41) | **1.000** (36) |
| | 50 | 13 | 0.067 (17) | 0.001 (14) | **0.826** (17) | 0.517 (14) | **0.900** (17) | 0.637 (14) |
| $\lambda = 0.8$ | 100 | 25 | 0.055 (25) | 0.004 (20) | **0.954** (25) | 0.738 (20) | **1.000** (25) | 0.888 (20) |
| | 300 | 75 | 0.052 (36) | 0.042 (31) | **1.000** (36) | **1.000** (31) | **1.000** (36) | **1.000** (31) |
| **$d = 10$** | | | | | | | | |
| | 50 | 117 | 0.102 (81) | 0.002 (79) | **0.998** (81) | 0.738 (79) | **1.000** (81) | 0.652 (79) |
| $\lambda = 0.3$ | 100 | 234 | 0.054 (116) | 0.002 (105) | **1.000** (116) | 0.998 (105) | **1.000** (116) | **1.000** (105) |
| | 300 | 700 | 0.044 (152) | 0.010 (144) | **1.000** (152) | **1.000** (144) | **1.000** (152) | **1.000** (144) |
| | 50 | 50 | 0.128 (46) | 0.000 (42) | 0.745 (46) | 0.392 (42) | **1.000** (46) | 0.789 (42) |
| $\lambda = 0.5$ | 100 | 100 | 0.070 (67) | 0.000 (59) | **1.000** (67) | 0.498 (59) | **1.000** (67) | 0.878 (59) |
| | 300 | 300 | 0.042 (93) | 0.004 (81) | **1.000** (93) | **1.000** (81) | **1.000** (93) | **1.000** (81) |
| | 50 | 13 | 0.046 (34) | 0.049 (29) | **0.898** (34) | 0.621 (29) | **0.896** (34) | 0.638 (29) |
| $\lambda = 0.8$ | 100 | 25 | 0.065 (43) | 0.056 (38) | **0.994** (43) | 0.735 (38) | **1.000** (43) | 0.846 (38) |
| | 300 | 75 | 0.068 (55) | 0.004 (54) | **1.000** (55) | 0.986 (54) | **1.000** (55) | **1.000** (54) |

of the test at $\mu_0$. If $\mu$ increases, so do the powers of the tests. Even if the dimension of the data is increased, the performance of the proposed test is not affected. Here we compare the performance of tests proposed in Gretton et al. (2006) (denoted by KMMD), as one of the referees suggested. In the course of computing the size and the power of the KMMD test, the R-function `kmmd` is used in the R-package named `kernlab` with default parameters. Besides, as per advice from a referee, we also consider the energy distance based Cramér-test (see Baringhaus & Franz (2010)), which is implemented using the R-function "cramer.test" in the R package named `cramer`. All results of KMMD and Cramér tests are reported in Table 5. Moreover, observe that in Tables 4 and 5, for each row, corresponding to $\mu = 0.5$ and $\mu = 1$, the maximum power values of the four tests, namely, KS.depth, CvM.depth, KMMD, and Cramér, are written in bold font.

The values reported in Tables 4 and 5 indicate that the proposed half-space depth-based KS and CVM tests perform much better than the MMD-based KMMD test and marginally better than the energy distance-based Cramér test in terms of power. However, in terms of computational time, both KMMD and Cramér tests are marginally superior relative to the half-space depth-based KS and CVM tests when the dimension of the data is fairly large.

Overall, given the good power and robustness properties of the half-space depth-based KS and CVM tests with controllable computational complexity up to moderately large data, one may conclude that these two tests are useful in practice from a Machine Learning and Statistical Science point of view.

Table 5: Two-sample test, namely, KMMD and Cramér: Observed relative frequency for rejecting the null based on Model B. The computational time (in seconds for single iteration $\times 100$) is provided in parentheses.

| | n | m | $\mu = 0$ (**Null**) | | $\mu = 0.5$ (**Alternative**) | | $\mu = 1$ (**Alternative**) | |
| --- | --- | --- | --- | --- | --- | --- | --- | --- |
| | | | KMMD | Cramér | KMMD | Cramér | KMMD | Cramér |
| $d = 2$ | | | | | | | | |
| | 50 | 117 | 0.036 (11) | 0.054 (09) | 0.546 (11) | 0.785 (09) | 0.699 (11) | 0.885 (09) |
| $\lambda = 0.3$ | 100 | 234 | 0.037 (20) | 0.051 (19) | 0.573 (20) | 0.811 (19) | 0.632 (20) | 0.901 (19) |
| | 300 | 700 | 0.040 (63) | 0.048 (56) | 0.622 (63) | 0.855 (56) | 0.701 (63) | 0.924 (56) |
| | 50 | 50 | 0.075 (07) | 0.051 (07) | 0.462 (07) | 0.611 (07) | 0.577 (07) | 0.723 (07) |
| $\lambda = 0.5$ | 100 | 100 | 0.044 (13) | 0.046 (11) | 0.480 (13) | 0.655 (11) | 0.602 (13) | 0.770 (11) |
| | 300 | 300 | 0.040 (21) | 0.052 (20) | 0.555 (21) | 0.721 (20) | 0.658 (21) | 0.869 (20) |
| | 50 | 13 | 0.086 (05) | 0.036 (04) | 0.384 (05) | **0.544** (04) | 0.581 (05) | 0.759 (04) |
| $\lambda = 0.8$ | 100 | 25 | 0.076 (13) | 0.056 (12) | 0.476 (13) | 0.658 (12) | 0.600 (13) | 0.762 (12) |
| | 300 | 75 | 0.072 (24) | 0.056 (23) | 0.601 (24) | 0.832 (23) | 0.798 (24) | 0.845 (23) |
| $d = 6$ | | | | | | | | |
| | 50 | 117 | 0.037 (21) | 0.044 (21) | 0.602 (21) | 0.858 (21) | 0.711 (21) | 0.923 (21) |
| $\lambda = 0.3$ | 100 | 234 | 0.062 (29) | 0.048 (27) | 0.633 (29) | 0.886 (27) | 0.742 (29) | 0.941 (27) |
| | 300 | 700 | 0.057 (46) | 0.051 (42) | 0.689 (46) | 0.900 (42) | 0.804 (46) | 0.976 (42) |
| | 50 | 50 | 0.060 (17) | 0.057 (17) | 0.587 (17) | 0.831 (17) | 0.688 (17) | 0.857 (17) |
| $\lambda = 0.5$ | 100 | 100 | 0.068 (23) | 0.040 (22) | 0.621 (23) | 0.855 (22) | 0.655 (23) | 0.905 (22) |
| | 300 | 300 | 0.057 (33) | 0.043 (33) | 0.701 (33) | 0.903 (33) | 0.769 (33) | 0.943 (33) |
| | 50 | 13 | 0.038 (12) | 0.037 (11) | 0.499 (12) | 0.702 (11) | 0.631 (12) | 0.845 (11) |
| $\lambda = 0.8$ | 100 | 25 | 0.039 (16) | 0.054 (16) | 0.576 (16) | 0.754 (16) | 0.689 (16) | 0.877 (16) |
| | 300 | 75 | 0.052 (32) | 0.047 (33) | 0.606 (32) | 0.823 (33) | 0.711 (32) | 0.945 (33) |
| $d = 10$ | | | | | | | | |
| | 50 | 117 | 0.052 (39) | 0.048 (37) | 0.538 (39) | 0.838 (37) | 0.623 (39) | 0.899 (37) |
| $\lambda = 0.3$ | 100 | 234 | 0.055 (56) | 0.052 (56) | 0.601 (56) | 0.901 (56) | 0.657 (56) | 0.924 (56) |
| | 300 | 700 | 0.054 (83) | 0.051 (80) | 0.666 (83) | 0.914 (80) | 0.700 (83) | 0.941 (80) |
| | 50 | 50 | 0.061 (33) | 0.060 (32) | 0.517 (33) | **0.800** (32) | 0.543 (33) | 0.843 (32) |
| $\lambda = 0.5$ | 100 | 100 | 0.059 (50) | 0.052 (48) | 0.545 (50) | 0.807 (48) | 0.599 (50) | 0.888 (48) |
| | 300 | 300 | 0.045 (64) | 0.049 (64) | 0.583 (64) | 0.841 (64) | 0.645 (64) | 0.899 (64) |
| | 50 | 13 | 0.046 (25) | 0.048 (24) | 0.498 (25) | 0.705 (24) | 0.549 (25) | 0.771 (24) |
| $\lambda = 0.8$ | 100 | 25 | 0.047 (34) | 0.059 (34) | 0.573 (34) | 0.822 (34) | 0.625 (34) | 0.849 (34) |
| | 300 | 75 | 0.058 (49) | 0.064 (50) | 0.600 (49) | 0.856 (50) | 0.711 (49) | 0.926 (50) |

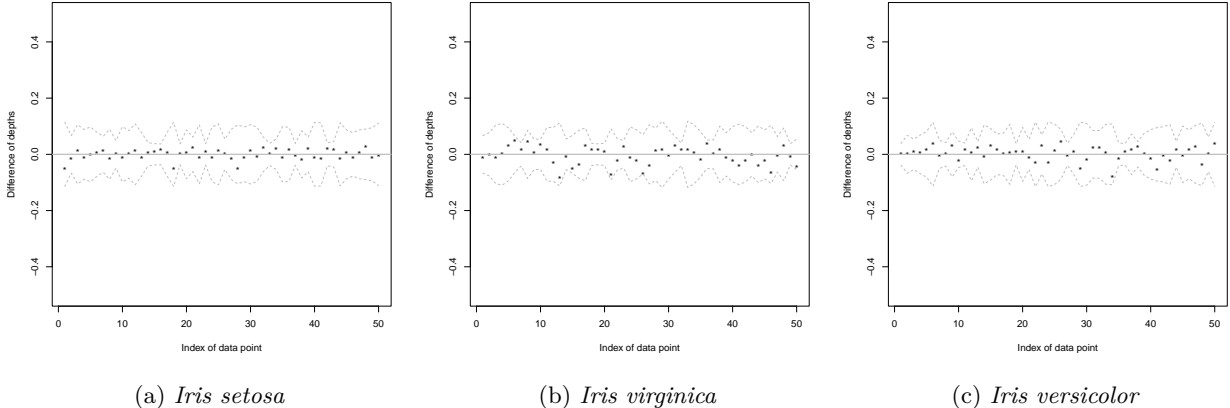

(a) *Iris setosa*       (b) *Iris virginica*       (c) *Iris versicolor*

Figure 4: *Goodness-of-fit test for Iris data:* The data-depth discrepancy plot for *Iris setosa, Iris virginica* and *Iris versicolor* respectively for testing normality. The dotted gray curves indicate the two-sigma limits of data-depth discrepancy

## 6   Real data analysis

To understand the practicability of the proposed tests, two well-known datasets are analyzed here.

### 6.1   *Fisher's Iris data*

This dataset, available in the base R, consists of three multivariate samples corresponding to three different species of Iris, namely *Iris setosa, Iris virginica* and *Iris versicolor*, with sample size 50 for each species. In each species, the length and width of the sepals are measured in centimeters. We would like to determine how close the distribution of each sample associated with sepal length and sepal width is to a bivariate normal sample. This can be formulated as a goodness-of-fit testing problem, with $F_0$ being a bivariate normal distribution with unknown mean $\boldsymbol{\mu}$ and unknown dispersion matrix $\boldsymbol{\Sigma}$. For each species, we estimate these unknown parameters using the corresponding mean vector and covariance matrix. The data-depth discrepancy is graphically represented for three different species in Figure 4, where we observe that most points are tightly clustered around the straight line, indicating that the standardized data are from a standard bivariate normal distribution. Moreover, our goodness-of-fit test for testing $H_0 : F = F_0$ against $H_1 : F \neq F_0$ led to very high empirical p-values for all types of spices (*Iris setosa*: 0.22 (KS) and 0.338 (CvM); *Iris virginica*: 0.35 (KS) and 0.358 (CvM); *Iris versicolor*: 0.192 (KS) and 0.118 (CvM) ). This indicates that $H_0$ is to be accepted, and therefore, bivariate normal distributions show signs of good fit for the data for the sepal length and sepal width of three Iris species. Moreover, many other tests also conclude that the sepal length and the sepal width of the Iris species follow a multivariate normal distribution, after appropriate normalization. For example, see the study in Section 4.2 of Dhar et al. (2014).

### 6.2   *gilgais* data

This dataset is available in the `MASS` package in R. It is collected on a line transect survey in the Gilgai territory in New South Wales, Australia (Webster, 1977). On a 4-meter spaced linear grid, 365 sampling locations are selected, and the Electrical conductivity (in mS/cm), pH, and chloride content (in ppm) are collected at three different depths below the surface, 0-10 cm, 30-40 cm, and 80-90 cm. We would like to investigate how close the joint distribution of three variables at each level is to a trivariate normal distribution. As in the previous example, this can be formulated as a goodness-of-fit problem, where $F_0$ is specified as a trivariate normal distribution with unknown mean $\mu$ and unknown dispersion matrix $\Sigma$. We estimate these unknown parameters using their respective maximum likelihood estimates. We then standardize the data in each sample using the corresponding mean vector and covariance matrix. The proposed data-depth

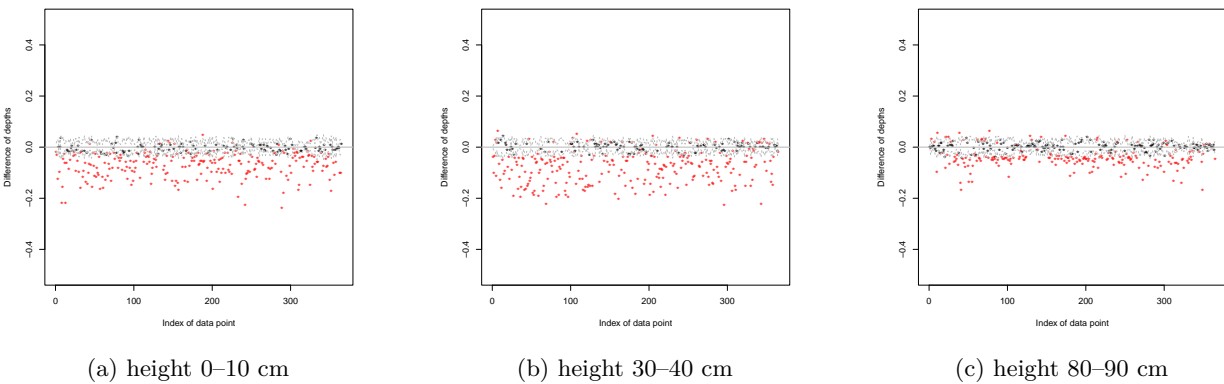

(a) height 0–10 cm                    (b) height 30–40 cm                    (c) height 80–90 cm

Figure 5: Goodness-of-fit test for *gilgais* data: The data-depth discrepancy plot for the tuple at depths below the surface of height 0-10 cm, 30-40 cm, and 80-90 cm, respectively, for testing normality. The dotted gray curves indicate the two-sigma limits of data-depth discrepancy.

dispersion plot is shown in Figure 5, where we observe that a majority of the data cloud is far from the straight line. This phenomenon indicates that the data are not from a trivariate normal distribution. Note that the same finding is concluded in Venables & Ripley (2002). Moreover, the test of $H_0 : F = F_0$ against $H_1 : F \neq F_0$ led to zero empirical p-values for all height levels below the surface, further validating the finding of the DDD plot. Furthermore, we are interested in studying whether the distribution of chemical parameters in a certain range of height is the same as that in some other range of height. Therefore, this can be viewed as a two-sample test where $F$ is the distribution of chemical parameters height at $0 - 10$ cm, and G is that of either at height $30 - 40$ cm or $80 - 90$ cm, respectively. The Figure 6 shows that the data cloud is far from the straight line; this phenomenon indicates that the distributions are different. Moreover, the empirical p-value for each of the situations is close to zero based on the proposed test, which determines that the distributions are significantly different. Furthermore, the scatter plots (see Figure 7) of three samples associated with three ranges of heights further clearly indicate that $F$ and $G$ are different.

## 7    Conclusion

In this paper, we have proposed a data-depth discrepancy and a graphical tool-kit based on Tukey's half-space depth. This device is used to test the equality of two distribution functions/goodness of fit problem, and is applicable to any dimension of the distributions. The motivations behind the graphical device are shown through simulation examples. Influenced by the graphical device, we have proposed test statistics based on the Kolmogorov-Smirnov and Cramér-von Mises tests. For goodness-of-fit and two-sample testing problems, the proposed test statistics can test the equality of two distribution functions, which are common in practice. We have shown that the test statistics are pointwise and also uniformly consistent. Moreover, we have studied the asymptotic power under contiguous alternatives—an important theoretical step that is sometimes overlooked. The applicability of the proposed method is illustrated by simulation studies and real data analysis.

The major advantage of the proposed graphical device is its independence from the data dimension. However, the efficiency of the proposed testing procedure heavily depends on the estimation of half-space depth. The computational efficiency of the proposed test depends on the computational algorithm of the half-space depth. Though the computation of half-space depth is not the main theme in this article, here we would like to discuss this issue. Some algorithms are available in the literature to compute the half-space depth for high-dimensional data. Among them, to the best of our knowledge, the worst-case complexity is $O(n^{d-1} \log n)$, and another couple of algorithms named MTMSAC and MTMSAD are proposed by Shao & Zuo (2020) (see proposition 2 therein). The computational complexity of the MTMSAC algorithm is $O(Nmknd)$, and the MTMSAD algorithm has computation complexity $O(Nmknd + Nmkd^3)$, where $N$ is the length of

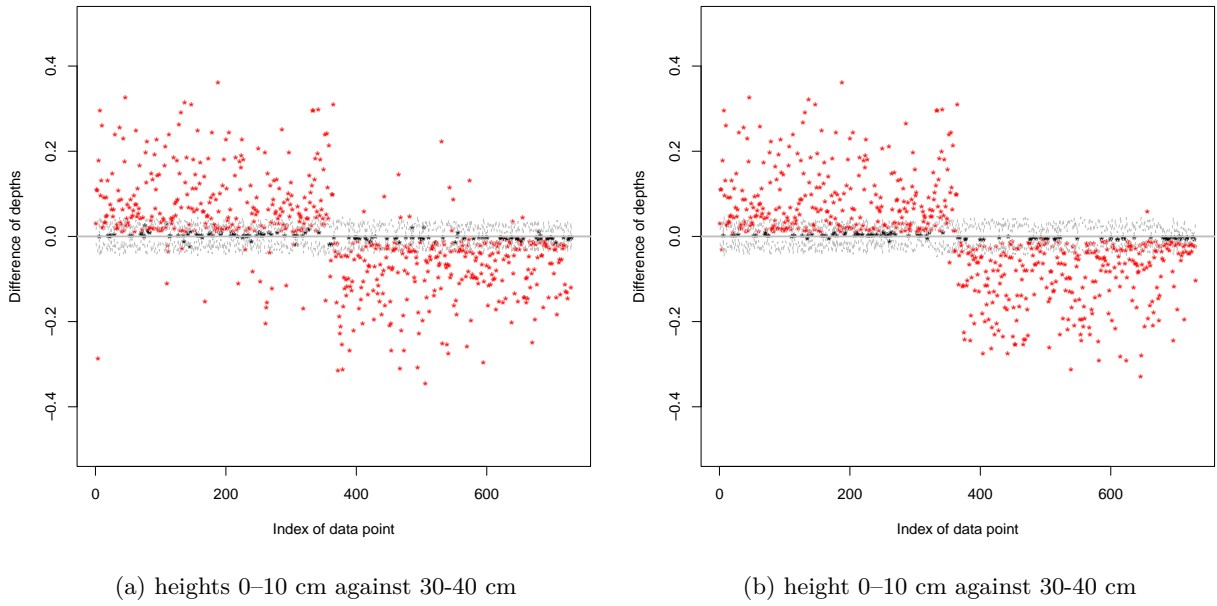

(a) heights 0–10 cm against 30-40 cm

(b) height 0–10 cm against 30-40 cm

Figure 6: Two-sample test for *gilgais* data: The data-depth discrepancy plot for heights 0–10 cm against 30-40 cm and height 0–10 cm against 30-40 cm are shown. The dotted gray curves indicate the two-sigma limits of data-depth discrepancy

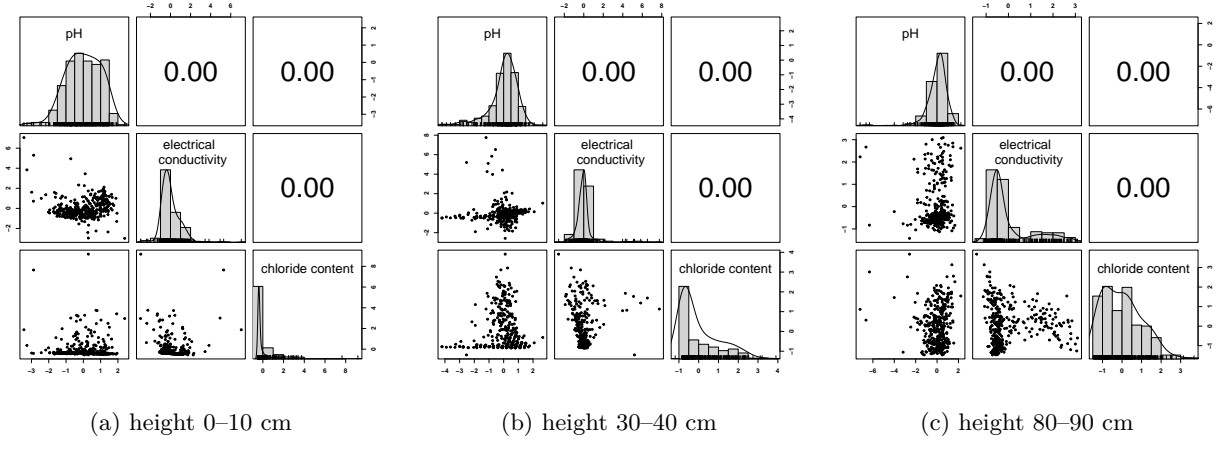

(a) height 0–10 cm

(b) height 30–40 cm

(c) height 80–90 cm

Figure 7: Scatterplots for *gilgais* data: Pairwise scatterplots (lower diagonal), marginal histograms (diagonal), and Pearson correlations (upper diagonal) for scaled measurements at depths 0–10 cm, 30–40 cm, and 80–90 cm.

decreasing cooling temperature, $m$ is the length of the Markov chain, $k$ is the number of multiple try points. In addition, some recent estimation methods for half-space depth are available at Rousseeuw & Ruts (1996); Ruts & Rousseeuw (1996); Rousseeuw & Struyf (1998); Dyckerhoff & Mozharovskyi (2016); Zuo (2019); Nagy et al. (2020); Dyckerhoff et al. (2021), though they may not be reasonably efficient for high-dimensional data. Developing more efficient algorithms for the estimation of half-space depth for high-dimensional datasets is a promising direction for future research in this domain. Note that in this work, the numerical experiments are conducted for ten-dimensional data; however, the proposed graphical tool and test statistics are readily adaptable to high-dimensional data, provided computational challenges are addressed.

**Acknowledgment:** The authors would like to thank the Editors-in-Chiefs, the Action Editor Professor Jasper C.H. Lee, and three anonymous reviewers for their constructive and stimulating comments, which significantly improved the article. Moreover, Subhra Sankar Dhar gratefully acknowledges the core research grant CRG/2022/001489, Government of India.

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

## A  Technical details

To investigate the asymptotic properties of the proposed graphical tool-kit and tests, we discuss a few propositions.

Suppose that $\mathbf{X}_1, \cdots, \mathbf{X}_n$ are the multivariate observations from an unknown distribution $F$, and the corresponding empirical distribution function is denoted by $\widehat{F}_n$. We define $\mathcal{L}_n(\mathbf{x}) = \sqrt{n}(D_{\mathcal{X}}(\mathbf{x}) - D_F(\mathbf{x}))$ for $\mathbf{x} \in \mathbb{R}^d$. Therefore, $\{\mathcal{L}_n : \mathbf{x} \in \mathbb{R}^d\}$ is a stochastic process with bounded sample path which is a map into $l^\infty(\mathbb{R}^d)$. Here $l^\infty(\mathbb{R}^d)$ is a space of bounded real valued function on $\mathbb{R}^d$ equipped with the uniform norm, viz., $\|a\|_\infty = \sup_{\mathbf{t} \in \mathbb{R}^d} |a(\mathbf{t})|$. Moreover, define $\mathcal{V}_n(H) = \sqrt{n}(\widehat{F}_n - F)(H)$, and that can be viewed as a map into space $l^\infty(\mathcal{H})$, where $\mathcal{H}$ is the class of closed half-spaces $H$ in $\mathbb{R}^d$.

Proposition 10 states the weak convergence of $\mathcal{V}_n$, which is the key component of $\mathcal{L}_n(\mathbf{x})$.

**Proposition 10.** *(Theorem 21 in section VII.5 of Pollard (1984)) Under condition (C2), $\mathcal{V}_n = \sqrt{n}(\widehat{F}_n - F)$ converges weakly to $\mathcal{V}_F$, where $\mathcal{V}_F$ is tight and $F$-Brownian bridge with mean zero and covariance function $\Sigma_F = F(H_1 \cap H_2) - F(H_1)F(H_2)$ for $H_1, H_2 \in \mathcal{H}$. Moreover, $\mathcal{V}_F$ can be chosen such that each sample path is continuous with respect to $\rho_F$.*

Let us now define $\mathcal{J}(\mathcal{V}_F)(\mathbf{x}) = \inf_{\mathbf{v} \in V(\mathbf{x})} \mathcal{V}_F H[\mathbf{x}, \mathbf{v}]$ for $\mathbf{x} \in \mathbb{R}^d$, where $V(\mathbf{x})$ is the set of all minimal directions passing through that point. Proposition 11 states the asymptotic distribution of Tukey's half-space depth.

**Proposition 11.** *(Massé (2004)) Under the conditions (C1)-(C3), $\{\mathcal{L}_n\}$ converges weakly to $\mathcal{J}(\mathcal{V}_F)$, which is tight measurable map into $l^\infty(A)$, where $A \subset \mathbb{R}^d$. Strictly speaking, $\mathcal{J}(\mathcal{V}_F)$ is a random element associated with a centered Gaussian process with covariance function $Cov\{\mathcal{J}(\mathcal{V}_F)(\mathbf{x}_1), \mathcal{J}(\mathcal{V}_F)(\mathbf{x}_2)\} = F(H[\mathbf{x}_1] \cap H[\mathbf{x}_2]) - F(H[\mathbf{x}_1])F(H[\mathbf{x}_2])$ for $\mathbf{x}_1, \mathbf{x}_2 \in A$.*

*Proof of Theroem 2.* For each direction, $\mathbf{u}_j \in \mathbb{S}^{d-1}$ is sampled by drawing a vector $\mathbf{Z}_j = (Z_{j1}, \cdots, Z_{id})^{\mathrm{T}}$ with i.i.d. component $Z_{ji} \sim N(0,1)$ and setting $u_j = \mathbf{Z}_j / \|\mathbf{Z}_j\|_2$. Thus, the computation cost is $O(d)$ (draw $d$ standard normals). Also, to compute $\|\mathbf{Z}_j\|_2$ cost $O(d)$ due to one dot product. In addition, the normalizing cost is $O(d)$. Hence, in each direction costs $O(d)$ and for all $k$ dimensions together, the cost for sampling is $O(kd)$.

Now for a fixed direction, $\mathbf{u}_j \in \mathbb{S}^{d-1}$, for each data point $\mathbf{X}_i$, compute the scalar projection $\pi_{ij} = <\mathbf{u}_j, \mathbf{X}_i>$. Thus, the computation cost is $O(d)$ per point. Due to the projection of all points in all directions, the total cost is $O(knd)$.

The rank computation for each direction, the cost is $(nd + n \log n)$.

For each evaluation point, the random half-space depth is computed by taking the minimum over k empirical cumulative distribution functions, resulting in a computational complexity of $O(k)$.

By combining the above four arguments, we have the total cost $T(n, d, k) = O(knd + kn \log n)$. $\qquad\square$

*Proof of Theorem 3.*

Note that, for every $\epsilon > 0$,

$$\lim_{n \to \infty} \mathbb{P}\left\{\widehat{\mathbb{C}}(\mathcal{X}, F_0) \subset \mathbb{C}_\epsilon(F, F_0)\right\} \geq \mathbb{P}\left\{\sup_{\mathbf{x}} |D_{\mathcal{X}}(\mathbf{x}) - D_{F_0}(\mathbf{x})| < \epsilon\right\} \to 1 \qquad (4)$$

due to the Proposition 4.4 of Massé (2004). $\qquad\square$

*Proof of Theorem 4.*

By the similar argument of Theorem 3, using the fact that $\lim_{\min(n,m) \to \infty} n/(n+m) = \lambda \in (0,1)$, observe that,

$$\lim_{\min(n,m) \to \infty} \mathbb{P}\{\widehat{\mathbb{C}}(\mathcal{X}, \mathcal{Y}) \subset \mathbb{C}_\epsilon(F, G)\} \geq \mathbb{P}\left\{\sup_{\mathbf{x}} |D_{\mathcal{X}}(\mathbf{x}) - D_{\mathcal{Y}}(\mathbf{x})| < \epsilon\right\} \to 1$$

$\square$

*Proof of Theorem 5.*

*Part 1:*

Let us define the Kolmogorov-Smirnov distance between empirical and theoretical distribution functions $\widehat{F}_n$ and $F_0$ based on Tukey's half-space depth $D$ as $d_K(\mathcal{X}, F_0) = \sup_{\mathbf{x} \in \mathbb{R}^d} |D_{\mathcal{X}}(\mathbf{x}) - D_{F_0}(\mathbf{x})| = \sup_{\mathbf{x},\mathbf{u}} |\widehat{F}_n(H[\mathbf{x}, \mathbf{u}]) - F_0(H[\mathbf{x}, \mathbf{u}])|$ where $\mathcal{X} = \{\mathbf{X}_1, \cdots, \mathbf{X}_n\}$ and $H$ is the half-space with $H[\mathbf{x}, \mathbf{u}] = \{\mathbf{y} \in \mathbb{R}^d : \mathbf{u}^{\mathrm{T}}\mathbf{x} \geq \mathbf{u}^{\mathrm{T}}\mathbf{y}\}$. It is important to note that $d_K(\cdot, \cdot)$ has the Glivenko-Cantelli property (Pollard, 1984; Donoho & Gasko, 1992), i.e., if the data samples are i.i.d. from $F_0$, $d_K(\mathcal{X}, F_0) \to 0$ as $n \to \infty$. Therefore, under an alternative $F$, we have

$$\sup_{\mathbf{x}} |D_{\mathcal{X}}(\mathbf{x}) - D_{F_0}(\mathbf{x})| \to d_K(D_F, D_{F_0}) > 0 \tag{5}$$

almost surely. Now fix any $F$ with $d_K(D_F, D_{F_0}) > 0$, then there exists some $\mathbf{x}$ with $D_{\mathcal{X}}(\mathbf{x}) \neq D_F(\mathbf{x})$. Without loss of generality, assume that $D_{\mathcal{X}}(\mathbf{x}) > D_{F_0}(\mathbf{x})$. Then

$$\begin{aligned}
\mathbb{P}_F \left\{ T_{\mathcal{X}, F_0}^{\mathrm{KS}} > s_{1-\alpha}^{(1)} \right\} &\geq \mathbb{P}_F \left\{ \sqrt{n} |D_{\mathcal{X}}(\mathbf{x}) - D_{F_0}(\mathbf{x})| > s_{1-\alpha}^{(1)} \right\} \\
&= \mathbb{P}_F \left\{ \sqrt{n}[D_{\mathcal{X}}(\mathbf{x}) - D_F(\mathbf{x}) + D_F(\mathbf{x}) - D_{F_0}(\mathbf{x})] > s_{1-\alpha}^{(1)} \right\} \\
&\geq \mathbb{P}_F \left\{ \sqrt{n}[D_{\mathcal{X}}(\mathbf{x}) - D_F(\mathbf{x})] > s_{1-\alpha}^{(1)} - \sqrt{n}[D_F(\mathbf{x}) - D_{F_0}(\mathbf{x})] \right\} \\
&\to 1 \text{ as } n \to \infty.
\end{aligned} \tag{6}$$

The last implication follows from the fact that $\{\sqrt{n}(D_{\mathcal{X}}(\mathbf{x}) - D_F(\mathbf{x}))\}$ is uniformly bounded in probability in view of Proposition 11 and an application of Prokhorov's theorem, and for finite $s_{1-\alpha}^{(1)}$, we have $s_{1-\alpha}^{(1)} - \sqrt{n}|D_F(\mathbf{x}) - D_{F_0}(\mathbf{x})| \to -\infty$ as $n \to \infty$. Hence the limiting power is one when $D_F(\mathbf{x}) > D_{F_0}(\mathbf{x})$. A similar argument can show that the limiting power is one when $D_F(\mathbf{x}) < D_{F_0}(\mathbf{x})$ for some $\mathbf{x}$. Hence, the proof is complete for KS-based test.

*Part 2:*

Due to the Glivenko-Cantelli property of $d_K(\cdot, \cdot)$, under an alternative $F$, 5 holds almost surely. Thus fix any $F$ with $d_K(D_F, D_{F_0}) > 0$, then there exists some $\mathbf{x}$ with $D_{\mathcal{X}}(\mathbf{x}) \neq D_F(\mathbf{x})$. Therefore, observe that

$$\begin{aligned}
\mathbb{P}_F \left\{ T_{\mathcal{X}, F_0}^{\mathrm{CvM}} > s_{1-\alpha}^{(2)} \right\} &= \mathbb{P}_F \left\{ n \int (D_{\mathcal{X}}(\mathbf{x}) - D_{F_0}(\mathbf{x}))^2 dF_0(\mathbf{x}) > s_{1-\alpha}^{(2)} \right\} \\
&\geq \mathbb{P}_F \left\{ n \int (D_{\mathcal{X}}(\mathbf{x}) - D_F(\mathbf{x}))^2 dF_0(\mathbf{x}) > s_{1-\alpha}^{(2)} - a_n + 2b_n \right\} \\
&\to 1 \text{ as } n \to \infty,
\end{aligned} \tag{7}$$

where $a_n = n \int (D_F(\mathbf{x}) - D_{F_0}(\mathbf{x}))^2 dF_0(\mathbf{x})$ and $b_n = n \int (D_{\mathcal{X}}(\mathbf{x}) - D_F(\mathbf{x}))(D_F(\mathbf{x}) - D_{F_0}(\mathbf{x})) dF_0(\mathbf{x})$. The last implication follows from the following facts: (i) $\{\sqrt{n}(D_{\mathcal{X}}(\mathbf{x}) - D_F(\mathbf{x}))\}$ is uniformly bounded in probability in view of Proposition 11, (ii) $s_{1-\alpha}^{(2)}$ is positive finite, and (iii) $\sup_{\mathbf{x}} |D_{\mathcal{X}}(\mathbf{x}) - D_F(\mathbf{x})| \to 0$ as $n \to \infty$ under $F$ almost surely (see Donoho & Gasko (1992) and Proposition 4.4 of Massé (2004)). An application of Prokhorov's theorem shows that the limiting power is one when $D_F(\mathbf{x}) > D_{F_0}(\mathbf{x})$. A similar argument can show that the limiting power is one when $D_F(\mathbf{x}) < D_{F_0}(\mathbf{x})$ for some $\mathbf{x}$. Hence, the proof is complete for the CvM-based test. $\square$

*Proof of Theorem 6.*

*Part 1:*

Let $F_n$ be any distribution function that satisfies $\sqrt{n} d_K(D_{F_n}, D_{F_0}) \geq \Delta_n$. By the triangle inequality, we have

$$d_K(D_{F_n}, D_{F_0}) \leq d_K(D_{F_n}, D_{\mathcal{X}}) + d_K(D_{\mathcal{X}}, D_{F_0}), \tag{8}$$

which implies $T^{\text{KS}}_{\mathcal{X},F_0} \geq \Delta_n - \sqrt{n}d_K(D_{F_n}, D_{\mathcal{X}})$. Therefore,

$$
\begin{aligned}
\mathbb{P}_{F_n}\left\{T^{\text{KS}}_{\mathcal{X},F_0} > s^{(1)}_{1-\alpha}\right\} &\geq \mathbb{P}_{F_n}\left\{\Delta_n - \sqrt{n}d_K(D_{F_n}, D_{\mathcal{X}}) > s^{(1)}_{1-\alpha}\right\} \\
&\geq \mathbb{P}_{F_n}\left\{\sqrt{n}d_K(D_{F_n}, D_{\mathcal{X}}) \leq \Delta_n - s^{(1)}_{1-\alpha}\right\} \\
&\to 1 \text{ as } n \to \infty.
\end{aligned}
\tag{9}
$$

The last implication follows from the following facts: (i) $\{\sqrt{n}(D_{\mathcal{X}}(\mathbf{x}) - D_F(\mathbf{x}))\}$ is uniformly bounded under $F_n$ in probability in view of Proposition 11 (ii) $\Delta_n \to \infty$, and (iii) $s^{(1)}_{1-\alpha}$ is positive and finite. An application of Prokhorov's theorem, we get $\mathbb{P}_{F_n}\left\{T^{\text{KS}}_{\mathcal{X},F_0} > s^{(1)}_{1-\alpha}\right\} \to 1$.

*Part 2:*

Let $F_n$ be any distribution function satisfies $\sqrt{n}d_K(D_{F_n}, D_{F_0}) \geq \Delta_n$. Therefore, using the triangle inequality equation 8, we have

$$
\begin{aligned}
\mathbb{P}_{F_n}\left\{T^{\text{CvM}}_{\mathcal{X},F_0} > s^{(2)}_{1-\alpha}\right\} &\geq \mathbb{P}_{F_n}\left\{\int \Delta_n^2 dF_0(\mathbf{x}) - n\int d_K^2(D_{F_n}, D_{\mathcal{X}})dF_0(\mathbf{x}) > s^{(2)}_{1-\alpha}\right\} \\
&\geq \mathbb{P}_{F_n}\left\{n\int d_K^2(D_{F_n}, D_{\mathcal{X}})dF_0(\mathbf{x}) \leq \int \Delta_n^2 dF_0(\mathbf{x}) - s^{(2)}_{1-\alpha}\right\} \\
&\to 1 \text{ as } n \to \infty.
\end{aligned}
\tag{10}
$$

The last implication follows from the the following fact: (i) $\{\sqrt{n}(D_{\mathcal{X}}(\mathbf{x}) - D_F(\mathbf{x}))\}$ is uniformly bounded under $F_n$ in probability in view of Proposition 11 (ii) $\Delta_n \to \infty$, and (iii) $s^{(2)}_{1-\alpha}$ is positive and finite. An application of Prokhorov's theorem, we get $\mathbb{P}_{F_n}\left\{T^{\text{CvM}}_{\mathcal{X},F_0} > s^{(2)}_{1-\alpha}\right\} \to 1$. $\qquad\square$

*Proof of Theorem 7.*

Define $a_{n,m} = \sqrt{n+m} \times d_K(D_F, D_G) = \sqrt{n+m}|D_F(\mathbf{x}) - D_G(\mathbf{x})|$.

$$
\begin{aligned}
&\mathbb{P}_{H_1}\left\{T^{\text{KS}}_{\mathcal{X},\mathcal{Y}} > t^{(1)}_{1-\alpha}\right\} \\
&\geq \mathbb{P}_{H_1}\left\{\sqrt{n+m}|D_{\mathcal{X}}(\mathbf{x}) - D_{\mathcal{Y}}(\mathbf{x})| > t^{(1)}_{1-\alpha}\right\} \\
&\geq \mathbb{P}_{H_1}\left\{\sqrt{n+m}|D_{\mathcal{X}}(\mathbf{x}) - D_F(\mathbf{x})| + \sqrt{n+m}|D_{\mathcal{Y}}(\mathbf{x}) - D_G(\mathbf{x})| > t^{(1)}_{1-\alpha} - a_{n,m}\right\} \\
&\geq \mathbb{P}_{H_1}\left\{\lambda^{-1/2}\sqrt{n}|D_{\mathcal{X}}(\mathbf{x}) - D_F(\mathbf{x})| + (1-\lambda)^{-1/2}\sqrt{m}|D_{\mathcal{Y}}(\mathbf{x}) - D_G(\mathbf{x})| > t^{(1)}_{1-\alpha} - a_{n,m}\right\} \\
&\to 1 \text{ as } \min(n,m) \to \infty.
\end{aligned}
\tag{11}
$$

The last implication follows from the fact that $\{\sqrt{n}(D_{\mathcal{X}}(\mathbf{x}) - D_F(\mathbf{x}))\}$ and $\{\sqrt{m}(D_{\mathcal{Y}}(\mathbf{x}) - D_G(\mathbf{x}))\}$ are uniformly bounded in probability in view of Proposition 11, for finite $t^{(1)}_\alpha$, we have $\sqrt{n+m}|D_F(\mathbf{x}) - D_G(\mathbf{x})| \to \infty$ as $\min(n,m) \to \infty$. By an application of Prokhorov's theorem, we get $\mathbb{P}_{H_1}\left\{T^{\text{KS}}_{\mathcal{X},\mathcal{Y}} > t^{(1)}_{1-\alpha}\right\} \to 1$ as $\min(n,m) \to \infty$.

Furthermore,

$$
\begin{aligned}
&\mathbb{P}_{H_1}\left\{T^{\text{CvM}}_{\mathcal{X},\mathcal{Y}} > t^{(2)}_{1-\alpha}\right\} \\
&\geq \mathbb{P}\left\{\lambda^{-1}n\int (D_{\mathcal{X}}(\mathbf{x}) - D_F(\mathbf{x}))^2 dH_{n,m}(\mathbf{x}) \right. \\
&\qquad \left. + (1-\lambda)^{-1}m\int (D_{\mathcal{Y}}(\mathbf{x}) - D_G(\mathbf{x}))^2 dH_{n,m}(\mathbf{x}) > t^{(2)}_{1-\alpha} - a^*_{n,m}\right\} \\
&\to 1 \text{ as } \min(n,m) \to \infty
\end{aligned}
\tag{12}
$$

where $a^*_{n,m} = a_{n,m} + 2b_{n,m} + 2c_{n,m} + 2d_{n,m}$ with

$$a_{n,m} = (n+m) \int (D_F(\mathbf{x}) - D_G(\mathbf{x}))^2 dH_{n,,m}(\mathbf{x})$$

$$b_{n,m} = (n+m) \int (D_{\mathcal{X}}(\mathbf{x}) - D_F(\mathbf{x}))(D_{\mathcal{Y}}(\mathbf{x}) - D_G(\mathbf{x})) dH_{n,m}(\mathbf{x})$$

$$c_{n,m} = (n+m) \int (D_{\mathcal{X}}(\mathbf{x}) - D_F(\mathbf{x}))(D_F(\mathbf{x}) - D_G(\mathbf{x})) dH_{n,m}(\mathbf{x})$$

$$d_{n,m} = (n+m) \int (D_{\mathcal{Y}}(\mathbf{x}) - D_G(\mathbf{x}))(D_F(\mathbf{x}) - D_G(\mathbf{x})) dH_{n,m}(\mathbf{x})$$

The last implication follows from the following facts: (i) $\{\sqrt{n}(D_{\mathcal{X}}(\mathbf{x}) - D_F(\mathbf{x}))\}$ and $\sqrt{m}(D_{\mathcal{Y}}(\mathbf{x}) - D_G(\mathbf{x}))\}$ are uniformly bounded in probability in view of Proposition 11, (ii) $t^{(1)}_{1-\alpha}$ is positive finite, (iii) $\lambda = \lim_{\min(n,m)\to\infty} \frac{n}{n+m} = \lambda \in (0,1)$, (iv) $H_{n,m} \to H$ almost surely, due to Glivenko-Cantelli's theorem, (iv) $a_{n,m} \to \infty$ $b_{n,m}, c_{n,m}$ and $d_{n,m}$ are finite (due to (i)-(iv)). Thus, with an application of Prokhorov's theorem, we get $\mathbb{P}_{H_1}\left\{T^{\text{CvM}}_{\mathcal{X},\mathcal{Y}} > t^{(2)}_{1-\alpha}\right\} \to 1$ as $\min(n,m) \to \infty$.

Let $F_n$ and $G_m$ be any distribution functions that satisfies $\sqrt{n+m}d_K(D_{F_n}, D_{G_m}) \geq \Delta_{n,m}$. By the triangle inequality, we have

$$d_K(D_{F_n}, D_{G_m}) \leq d_K(D_{F_n}, D_{\mathcal{X}}) + d_K(D_{\mathcal{X}}, D_{\mathcal{Y}}) + d_K(D_{\mathcal{Y}}, D_{G_m}) \tag{13}$$

which implies $T^{\text{KS}}_{\mathcal{X},\mathcal{Y}} \geq \Delta_{n,m} - \sqrt{n+m}d_K(D_{\mathcal{X}}, D_{F_n}) - \sqrt{n+m}d_K(D_{\mathcal{Y}}, D_{G_m})$. Therefore,

$$\mathbb{P}_{F_n, G_m}\left\{T^{\text{KS}}_{\mathcal{X},\mathcal{Y}} > t^{(1)}_{1-\alpha}\right\}$$
$$\geq \mathbb{P}_{F_n, G_m}\left\{\Delta_{n,m} - \sqrt{n+m}d_K(D_{\mathcal{X}}, D_{F_n}) - \sqrt{n+m}d_K(D_{\mathcal{Y}}, D_{G_m}) > t^{(1)}_{1-\alpha}\right\}$$
$$\geq \mathbb{P}_{F_n, G_m}\left\{\sqrt{n+m}d_K(D_{\mathcal{X}}, D_{F_n}) + \sqrt{n+m}d_K(D_{\mathcal{Y}}, D_{G_m}) \leq \Delta_{n,m} - t^{(1)}_{1-\alpha}\right\}$$
$$\to 1 \text{ as } \min(n,m) \to \infty. \tag{14}$$

The last implication follows from the following facts: (i) $\{\sqrt{n}(D_{\mathcal{X}}(\mathbf{x}) - D_F(\mathbf{x}))\}$ and $\{\sqrt{m}(D_{\mathcal{Y}}(\mathbf{x}) - D_G(\mathbf{x}))\}$ are uniformly bounded in probability under $F_n$ and $G_m$ respectively in the view of Proposition 11, (ii) $\lambda \in (0,1)$ and $t^{(1)}_{1-\alpha}$ is positive finite. (iii) $\Delta_{n,m} \to \infty$. An application of Prokhorov's theorem, we get $\mathbb{P}_{F_n, G_m}\left\{T^{\text{KS}}_{\mathcal{X},\mathcal{Y}} > t^{(1)}_{1-\alpha}\right\} \to 1$ as $\min(n,m) \to \infty$.

Moreover, by the inequality equation 13,

$$\mathbb{P}_{F_n, G_m}\left\{T^{\text{CvM}}_{\mathcal{X},\mathcal{Y}} > s^{(2)}_{1-\alpha}\right\}$$
$$\geq \mathbb{P}_{F_n, G_m}\left\{\int \Delta^2_{n,m} dH_{n,m}(\mathbf{x}) - (n+m) \int d^2_K(D_{\mathcal{X}}, D_{F_n}) dH_{n,m}(\mathbf{x})\right.$$
$$\left. -(n+m) \int d^2_K(D_{\mathcal{Y}}, D_{G_m}) dH_{n,m}(\mathbf{x}) > t^{(2)}_{1-\alpha}\right\}$$
$$\geq \mathbb{P}_{F_n, G_m}\left\{(n+m) \int d^2_K(D_{\mathcal{X}}, D_{F_n}) dH_{n,m}(\mathbf{x})\right.$$
$$\left. +(n+m) \int d^2_K(D_{\mathcal{Y}}, D_{G_m}) dH_{n,m}(\mathbf{x}) \leq \int \Delta^2_{n,m} d\mathbf{x} - t^{(2)}_{1-\alpha}\right\}$$
$$\to 1 \text{ as } \min(n,m) \to \infty. \tag{15}$$

The last implication follows from the same facts that are used in equation 14. □

*Proof of Theorem 8.*

Observe that the log-likelihood ratio for testing $H_0 : F = F_0$ against $H_n$ described in equation 2,

$$
\begin{aligned}
\mathcal{L}_n &= \sum_{i=1}^{n} \log \frac{(1 - \gamma/\sqrt{n})f_0(\mathbf{x}_i) + (\gamma/\sqrt{n})h(\mathbf{x}_i)}{f_0(\mathbf{x}_i)} \\
&= \sum_{i=1}^{n} \log \left\{ 1 + (\gamma/\sqrt{n}) \left[ \frac{h(\mathbf{x}_i)}{f_0(\mathbf{x}_i)} - 1 \right] \right\} \\
&= \frac{\gamma}{\sqrt{n}} \sum_{i=1}^{n} \left\{ \frac{h(\mathbf{x}_i)}{f_0(\mathbf{x}_i} - 1 \right\} - \frac{\gamma^2}{2n} \sum_{i=1}^{n} \left\{ \frac{h(\mathbf{x}_i)}{f_0(\mathbf{x}_i} - 1 \right\}^2 + \mathcal{R}_n \\
&= \frac{\gamma}{\sqrt{n}} \sum_{i=1}^{n} \mathcal{K}_i - \frac{\gamma^2}{2n} \sum_{i=1}^{n} \mathcal{K}_i^2 + \mathcal{R}_n
\end{aligned}
\tag{16}
$$

where $\mathcal{K}_i = \frac{h(\mathbf{x}_i)}{f_0(\mathbf{x}_i)} - 1$. Since $\sigma^2 = \mathbb{E}_{F_0} \left\{ \frac{h(\mathbf{x})}{f_0(\mathbf{x})} - 1 \right\}^2$ is finite, $\mathcal{R}_n \xrightarrow{\mathbb{P}} 0$ as $n \to \infty$. Contiguity of the sequence $H_n$ directly follows from Dhar et al. (2014) (see Theorem 6.1) since the first term of equation 16 is asymptotically normal with mean zero and variance $\gamma^2 \sigma^2$ due to central limit theorem and second term converges in probability to $\gamma^2 \sigma^2 / 2$ due to weak law of large numbers. Therefore, by Slutsky's theorem, $\mathcal{L}_n$ is asymptotically normal with mean $-\gamma^2 \sigma^2 / 2$ and variance $\gamma^2 \sigma^2$. An application of Le Cam's first lemma, we can deduce the first part of the theorem.

Now, to apply Le Cam's third lemma, we need to calculate the covariance between $D_\mathcal{X} - D_{F_0}$ and $\mathcal{L}_n$ under the null.

Let $\mathcal{T}(F)$ be a functional defined for all distributions in a stable class, then the influence function of $\mathcal{T}$ at $F$ is defined as $\mathrm{IF}(\mathbf{z}; \mathcal{T}(F)) = \lim_{\epsilon \to 0^+} \frac{\mathcal{T}((1-\epsilon)F + \epsilon \delta_\mathbf{z}) - \mathcal{T}(F)}{\epsilon}$ where $\delta_\mathbf{z}$ is the point mass probability measure at $\mathbf{z} \in \mathbb{R}^d$ and $\epsilon \in [0,1]$. For any $\mathbf{z} \in \mathbb{R}^d$, partition the set closed half-space $\mathcal{H}$ as $\mathcal{H}_\mathbf{z} = \{H \in \mathcal{H} : \mathbf{z} \in H\}$ and $\mathcal{H}_{\bar{\mathbf{z}}} = \{H \in \mathcal{H} : \mathbf{z} \notin H\}$. Thus, corresponding depths are defined as $D_F^\mathbf{z}(\mathbf{x}) = \inf_{\mathcal{H}_\mathbf{z}} F(H)$ and $D_F^{\bar{\mathbf{z}}}(\mathbf{x}) = \inf_{\mathcal{H}_{\bar{\mathbf{z}}}} F(H)$ respectively. Then the influence function of half-space depth becomes $\mathrm{IF}(\mathbf{z}; D_F(\mathbf{x})) = -D_F(\mathbf{x})\mathbf{1}\{D_F^{(\bar{\mathbf{z}})}(\mathbf{x}) < D_F^{(\mathbf{z})}(\mathbf{x})\} + (1 - D_F(\mathbf{x}))\mathbf{1}\{D_F^{(\bar{\mathbf{z}})}(\mathbf{x}) \geq D_F^{(\mathbf{z})}(\mathbf{x})\}$ (Romanazzi, 2001), where $\mathbf{1}\{a \in A\}$ takes value 1 if $a \in A$ and zero otherwise. If $\mathbf{z} = \mathbf{x}$, then, $\mathcal{H}_\mathbf{z} = \mathcal{H}$ and $\mathcal{H}_{\bar{\mathbf{z}}}$ becomes null-set. Hence $\mathrm{IF}(\mathbf{x}; D_F(\mathbf{x})) = 1 - D_F(\mathbf{x})$. Note that IF is bounded, and is a step function. For all $\mathbf{x}$ belonging to optimal half-space $\mathrm{IF}(\mathbf{z}; D_{F_0}(\mathbf{x}))$ is constant and equal to $1 - D_F(\mathbf{x})$ and for all $\mathbf{z}$ belonging to non-optimal half-spaces, $\mathrm{IF}(\mathbf{z}; D_{F_0}(\mathbf{x}))$ is constant and equal to $D_F(\mathbf{x})$. Since by the assumption optimal half-space depth associated to $\mathbf{x}$ is unique, then for suitable regularity conditions on $F_0$ (Serfling, 2009) we can write the von-Mises expansion (Romanazzi, 2001) as

$$
D_\mathcal{X}(\mathbf{x}) - D_{F_0}(\mathbf{x}) = \frac{1}{n} \sum_{i=1}^{n} \mathrm{IF}(\mathbf{X}_i; D_{F_0}(\mathbf{x})) + \mathcal{E}_n
\tag{17}
$$

where $\mathcal{E}_n$ is the remaining term. Due to central limit theorem, it can be shown that $\sqrt{n} \mathcal{E}_n \xrightarrow{\mathbb{P}} 0$ (see Appendix 2 of Romanazzi (2001)). Therefore, the asymptotic covariance function between $\sqrt{n}(D_\mathcal{X}(\mathbf{x}) - D_{F_0}(\mathbf{x}))$ and $\mathcal{L}_n$ is

$$
\begin{aligned}
&\mathrm{Cov}_{H_0} \left\{ \sqrt{n}(D_\mathcal{X}(\mathbf{x}) - D_{F_0}(\mathbf{x})), \mathcal{L}_n \right\} \\
&= \frac{\gamma}{n} \mathrm{Cov}_{H_0} \left\{ \sum_{i=1}^{n} \mathrm{IF}(\mathbf{X}_i; D_{F_0}(\mathbf{x})), \sum_{i=1}^{n} \mathcal{K}_i \right\} \\
&\quad - \frac{\gamma^2}{2n^{3/2}} \mathrm{Cov}_{H_0} \left\{ \sum_{i=1}^{n} \mathrm{IF}(\mathbf{X}_i; D_{F_0}(\mathbf{x})), \sum_{i=1}^{n} \mathcal{K}_i^2 \right\} \\
&= -\gamma \mathrm{Cov}_{H_0} \left\{ D_{F_0}(\mathbf{x}), \frac{h(\mathbf{x})}{f_0(\mathbf{x})} \right\} + \frac{\gamma^2}{2\sqrt{n}} \mathrm{Cov}_{H_0} \left\{ D_{F_0}(\mathbf{x}), \left( \frac{h(\mathbf{x})}{f_0(\mathbf{x})} - 1 \right)^2 \right\} \\
&= -\gamma \int D_{F_0}(\mathbf{x}) h(\mathbf{x}) d\mathbf{x} + o(1)
\end{aligned}
\tag{18}
$$

The last implication follows from the fact that $\text{Cov}_{H_0}\left\{D_{F_0}(\mathbf{x}), \left(\frac{h(\mathbf{x})}{f_0(\mathbf{x})}-1\right)^2\right\}$ is finite that can be shown by applying Cauchy-Schwarz inequality with the fact that $\text{Var}_{H_0}\{D_{F_0(\mathbf{x})}\}$ and $\mathbb{E}_{H_0}\left\{\frac{h(\mathbf{x})}{f_0(\mathbf{x})}-1\right\}^4$ are finite. Thus, by Le Cam's third lemma, under contiguous alternatives the empirical Tukey's half-space depth process i.e. $\sqrt{n}(D_{\mathcal{X}}(\mathbf{x})-D_{F_0}(\mathbf{x}))$ converges to $\mathcal{G}_1'$ which is Gaussian process with mean $-\gamma\mathbb{E}_{\mathbf{x}\sim h}\{D_{F_0}(\mathbf{x})\}$ and the covariance kernel $F_0(H[\mathbf{x}_1]\cap H[\mathbf{x}_2])-F_0(H[\mathbf{x}_1])F_0(H[\mathbf{x}_2])$. Moreover, $\sqrt{n}(D_{\mathcal{X}}(\mathbf{x})-D_{F_0}(\mathbf{x}))$ satisfies the tightness condition under contiguous alternatives since it is tight under the null. The tightness under null follows from the weak convergence of the empirical Tukey's depth process (see Proposition 11). Therefore, by Le Cam's third lemma, under the contiguous alternatives alternatives $H_n$, the asymptotic power of the test based on $T_{\mathcal{X},F_0}^{\text{KS}}$ and $T_{\mathcal{X},F_0}^{\text{CvM}}$ are $\mathbb{P}_\gamma\left\{\sup_{\mathbf{x}}|\mathcal{G}_1'(\mathbf{x})| > s_{1-\alpha}^{(1)}\right\}$ and $\mathbb{P}_\gamma\left\{\int|\mathcal{G}_1'(\mathbf{x})|^2 dF_0(\mathbf{x}) > s_{1-\alpha}^{(2)}\right\}$, respectively. $\quad\square$

*Proof of Theorem 9.* Observe that the likelihood ratio for testing $H_0 : F = G$ against $H_{n,m}$ described in equation 3,

$$
\begin{aligned}
\mathcal{L}_{n,m} &= \sum_{i=1}^{n}\sum_{j=1}^{m}\log\frac{f(\mathbf{x}_i)\left\{(1-\gamma/\sqrt{n+m})f(y_j)+\gamma h(\mathbf{y}_j)/\sqrt{n+m}\right\}}{f(\mathbf{x}_i)f(\mathbf{y}_i)}\\
&= \sum_{j=1}^{m}\log\left\{1+\frac{\gamma}{\sqrt{n+m}}\left(\frac{h(y_j)}{f(y_j)}-1\right)\right\}\\
&= \frac{\gamma}{\sqrt{n+m}}\sum_{j=1}^{m}\left\{\frac{h(\mathbf{y}_j)}{f(\mathbf{y}_j)}-1\right\}-\frac{\gamma^2}{2(n+m)}\sum_{j=1}^{m}\left\{\frac{h(\mathbf{y}_j)}{f(\mathbf{y}_j)}-1\right\}^2+\mathcal{R}_{n,m}\\
&= \frac{\gamma}{\sqrt{n+m}}\sum_{j=1}^{m}\mathcal{K}_j'-\frac{\gamma^2}{2(n+m)}\sum_{j=1}^{m}\mathcal{K}_j'^2+\mathcal{R}_{n,m}
\end{aligned}
\tag{19}
$$

where $\mathcal{K}_i' = \frac{h(\mathbf{y}_j)}{f(\mathbf{y}_j)}-1$. Since $\sigma^2 = \mathbb{E}\left\{\frac{h(\mathbf{y})}{f(\mathbf{y})}-1\right\}^2$ is finite, $\mathcal{R}_{n,m}\xrightarrow{\mathbb{P}}0$ as $\min(n,m)\to\infty$. Contiguity of the sequence $H_{n,m}$ directly follows from Dhar et al. (2014) (see Theorem 6.2) since the first term of equation 19 is asymptotically normal with mean zero and variance $\gamma^2\sigma^2(1-\lambda)$ due to central limit theorem and second term converges in probability to $-\gamma^2\sigma^2(1-\lambda)/2$. Therefore, by Slutsky's theorem, $\mathcal{L}_{n,m}$ is asymptotically normal with mean $-\gamma^2\sigma^2(1-\lambda)/2$ and variance $\gamma^2\sigma^2(1-\lambda)$. An application of Le Cam's first lemma, we can deduce the first part of the theorem.

Now, to apply Le Cam's third lemma, we need to calculate the covariance between $D_{\mathcal{X}}-D_{\mathcal{Y}}$ and $\mathcal{L}_{n,m}$ under null. As in the proof of Theorem 8, using von-Mises expansion described in equation 17 and the fact that $\mathcal{X}$ and $\mathcal{Y}$ are independent, we have

$$
\begin{aligned}
&\text{Cov}_{H_0}\left\{\sqrt{n+m}(D_{\mathcal{X}}(\mathbf{u})-D_{\mathcal{Y}}(\mathbf{u})),\mathcal{L}_{n,m}\right\}\\
&= \gamma\text{Cov}_{H_0}\left\{\sqrt{n+m}(D_{\mathcal{X}}(\mathbf{u})-D_{\mathcal{Y}}(\mathbf{u})),\frac{1}{\sqrt{n+m}}\sum_{j=1}^{m}\mathcal{K}_j'\right\}\\
&\quad -\frac{\gamma^2}{2}\text{Cov}_{H_0}\left\{\sqrt{n+m}(D_{\mathcal{X}}(\mathbf{u})-D_{\mathcal{Y}}(\mathbf{u})),\frac{1}{(n+m)}\sum_{j=1}^{m}\mathcal{K}_j'^2\right\}:=\gamma\mathbb{A}_1-\frac{\gamma^2}{2}\mathbb{A}_2
\end{aligned}
\tag{20}
$$

where

$$
\begin{aligned}
\mathbb{A}_1 &= \text{Cov}_{H_0}\left\{\sqrt{n+m}(D_{\mathcal{X}}(\mathbf{u})-D_{\mathcal{Y}}(\mathbf{u})),\frac{1}{\sqrt{n+m}}\sum_{j=1}^{m}\mathcal{K}_j'\right\}\\
&= -\text{Cov}_{H_0}\left\{(1-\lambda)^{-1/2}\sqrt{m}(D_{\mathcal{Y}}(\mathbf{u})-D_F(\mathbf{u})),\frac{1}{\sqrt{n+m}}\sum_{j=1}^{m}\mathcal{K}_j'\right\}
\end{aligned}
$$

$$= -(1-\lambda)^{-1/2}\mathrm{Cov}_{H_0}\left\{\frac{1}{\sqrt{m}}\sum_{j=1}^{m}\mathrm{IF}(\mathbf{y}_j; D_F(\mathbf{u})), \frac{1}{\sqrt{n+m}}\sum_{j=1}^{m}\mathcal{K}'_j\right\}$$

$$= \sqrt{\frac{\lambda}{1-\lambda}}\mathrm{Cov}_{H_0}\left\{D_F(\mathbf{u}), \frac{h(\mathbf{u})}{f(\mathbf{u})}\right\} = \sqrt{\frac{\lambda}{1-\lambda}}\int D_F(\mathbf{u})h(\mathbf{u})d\mathbf{u} \tag{21}$$

and similar to the second term of equation 18, under the condition $\mathbb{E}\left\{\frac{h(\mathbf{x})}{f(\mathbf{x})}-1\right\}^4 < \infty$, by applying Cauchy-Schwarz inequality, $\mathbb{A}_2 = o(1)$ as $\min(n,m) \to \infty$. Thus, the covariance between $\sqrt{n+m}(D_{\mathcal{X}}(\mathbf{u}) - D_{\mathcal{Y}}(\mathbf{u}))$ and $\mathcal{L}_{n,m}$ is $\gamma\sqrt{\lambda/(1-\lambda)}\mathbb{E}_{\mathbf{u}\sim h}\{D_F(\mathbf{u})\}$.

Thus, by Le Cam's third lemma, under contiguous alternatives the empirical Tukey's half-space depth process i.e. $\sqrt{n+m}(D_{\mathcal{X}}(\mathbf{u}) - D_{\mathcal{Y}}(\mathbf{u}))$ converges to $\mathcal{G}'_2$ which is Gaussian process with mean $\gamma\sqrt{\lambda/(1-\lambda)}\mathbb{E}_{\mathbf{u}\sim h}\{D_F(\mathbf{u})\}$ and the covariance kernel $\{F(H[\mathbf{u}_1] \cap H[\mathbf{u}_2]) - F(H[\mathbf{u}_1])F_0(H[\mathbf{u}_2])\}/\lambda(1-\lambda)$. Moreover, $\sqrt{n+m}(D_{\mathcal{X}}(\mathbf{u}) - D_{\mathcal{Y}}(\mathbf{u}))$ satisfies the tightness condition under contiguous alternatives since it is tight under null. The tightness under null follows from the weak convergence of the empirical Tukey's depth process (under the independence of $\mathcal{X}$ and $\mathcal{Y}$, for $\lambda \in (0,1)$, see Proposition 11). Therefore, by Le Cam's third lemma, under the contiguous alternatives alternatives $H_n$, the asymptotic power of the test based on $T_{\mathcal{X},\mathcal{Y}}^{\mathrm{KS}}$ and $T_{\mathcal{X},\mathcal{Y}}^{\mathrm{CvM}}$ are $\mathbb{P}_\gamma\left\{\sup_{\mathbf{u}}|\mathcal{G}'_2(\mathbf{u})| > t_{1-\alpha}^{(1)}\right\}$ and $\mathbb{P}_\gamma\left\{\int |\mathcal{G}'_2(\mathbf{u})|^2 dH_{n,m}(\mathbf{u}) > t_{1-\alpha}^{(2)}\right\}$, respectively. $\qquad\square$

