# OpenReview forum: "Inspecting discrepancy between multivariate distributions using half-space depth-based information criteria"
_TMLR — Rejected by TMLR_

### Review · Reviewer_vhUf · 2025-11-30

**Summary Of Contributions:**

Summary:

This paper studies hypothesis testing problems that consider whether a multivariate distribution is different from a given distribution, or whether two multivariate distributions are different from each other. This paper proposes a test based on halfspace-depth, which uses samples from the distributions to approximately compute the halfspace depth for points in $R^d$ and compares the differences of the depth for each point. This paper shows that under certain assumptions, the test based on halfspace depth can be used to asymptotically distinguish the alternative hypothesis from the null hypothesis. Furthermore, this paper uses experiments to justify that the proposed statistical test is useful.

Strength:

This paper studies the hypothesis testing problem, a fundamental problem in statistics and data science. The study includes both theoretical results and empirical experiments, which are sound and complete.

Weakness:

1. I am not very convinced about the significance of the test. It has been known that computing the halfspace depth is NP-hard. Though this paper conducts some experiments of the proposed test, the dimensions are typically very low, and it is not clear to me whether these experiments are scalable to higher dimensions.

2. The algorithms that are used to implement these statistical tests are not written clearly. For example, the algorithm generates a large number of points over a high-dimensional unit sphere to approximate the supremum in the proposed test. However, it is not clear how large this number should be. In fact, I am not convinced that using points over a unit sphere can correctly approximate the supremum in the proposed test. Furthermore, the algorithm does not give precise criteria for when to reject the null hypothesis. Clearly, the choice of this criteria can significantly affect the empirical result.

3. The writing of this paper can be improved. For example, the main theorem of this paper involves 4 assumptions. But these four assumptions are not even mentioned in the main body of the paper.

**Audience:**

Yes

**Audience Explanation:**

I think people from the statistics community and the machine learning community would be interested in this paper, because this paper studies a fundamental problem in these areas.

**Claims And Evidence:**

No

**Claims Explanation:**

1. I am not quite sure if the hypothesis testing problem is formulated properly in this paper. Usually, when we formulate a hypothesis testing problem, we need some "gap" (usually the TV distance) between the null hypothesis and the alternative hypothesis to make the problem solvable information theoretically. However, in this paper, there is no such gap assumption. So, given two distributions with TV distance $\epsilon$, it is not very clear to me whether we would need an unbounded number of examples to make the test work in the worst case.

2. In the proposed test, one needs to compute the supremum of the difference of the halfspace depths of the two distributions over $R^d$. In the implementation of the test, the paper proposes that one could draw a large number of examples over the unit sphere to approximate this supremum. It seems to me that this method should not always be a correct approximation. For example, suppose $F$ and $F_0$ both have a support on some convex body that is disjoint from the unit sphere, then one cannot use points from the unit sphere to approximate the supremum over $R^d$, because for every point in the unit sphere, there would be a halfspace contains that point but contains no points from $F$ or $F_0$.

3. Some of the proofs in the paper are not formal enough. For example, the proof of Theorem 3 only contains one sentence
> "By the similar argument of Theorem 3, using the fact...".

This is clearly a typo. Furthermore, in the proof, it is unclear to me where the assumptions C1-C4 are used and where the assumption $\lambda \in (0,1)$ is used.

**Requested Changes:**

1. I suggest that the authors state the implementation of the algorithm more clearly. In particular, I believe that without any pre-processing step, one should not hope to use points from the unit sphere to approximate the supremum.

2. I think the writing of this paper can be improved a lot. For example, assumptions C1-C4 are almost stated in every theorem, so they can be moved to the main body. Furthermore, most proofs of the paper should be written more clearly, especially, the proofs should state clearly which assumption is used. It seems to me that many assumptions made in the statement are not clearly used in the proofs.

3. I think it would be better to make a clearer comparison between this work and prior work to make it clear why the proposed test is helpful.

---

> ### Author Response · Authors · 2025-12-22
> **Response to "Requested changes" is provided in the comment below. We thank the reviewer for the valuable suggestions.**
>
> **1.** I suggest that the authors state the implementation of the algorithm more clearly. In particular, I believe that without any pre-processing step, one should not hope to use points from the unit sphere to approximate the supremum.
>
>
> **Response:** We are not fully clear on what is meant by “preprocessing steps” in this context. In our analysis, the only preprocessing applied is standardization of the data. After standardization, we directly apply the proposed algorithm: Steps 1.1–1.6 for the one-sample test and Steps 2.1–2.6 for the two-sample test. No additional filtering, smoothing, transformation, or feature extraction is performed prior to implementing the proposed procedures.
>
> **2.** I think the writing of this paper can be improved a lot. For example, assumptions C1-C4 are almost stated in every theorem, so they can be moved to the main body. Furthermore, most proofs of the paper should be written more clearly, especially, the proofs should state clearly which assumption is used. It seems to me that many assumptions made in the statement are not clearly used in the proofs.
>
>  **Response:** Note that the characterization of distribution by half-space depth (see Proposition 1) requires assumption C1-C4. Kindly see the research articles mentioned in the statement of Proposition 1. Since Proposition 1 is the backbone of all theoretical results, the assumptions C1-C4 are required for all the theoretical results. We agree with your suggestion to move it to the main body to improve readability among readers. We will incorporate this if given the opportunity to revise. Besides the assumptions C1-C4, the key moment-based (strictly speaking, fourth moment) assumption in Theorems 7 and 8 is proved with full explanation. Kindly see equation numbers (16) and (19) in the proofs of Theorems 7 and 8, respectively.
>
> **3.** I think it would be better to make a clearer comparison between this work and prior work to make it clear why the proposed test is helpful.
>
> **Response:** In Section 5 (Tables 1-4), we have provided an extensive comparison study between our work and the prior related works through simulation studies.

---

> ### Author Response · Authors · 2026-01-02
> **Response to weakness comment of the Reviewer**
>
> Reviewer's Comment 1 : I am not very convinced about the significance of the test. It has been known that computing the halfspace depth is NP-hard. Though this paper conducts some experiments of the proposed test, the dimensions are typically very low, and it is not clear to me whether these experiments are scalable to higher dimensions.
>
> **Response to Comment 1** : We agree with the reviewer that the computation of the half space depth is NP-hard. On December 22, 2025, in responding to requested change (3) to Reviewer Qn4x. In the worst situation, the computational complexity of the half space depth is $O(n^{d - 1} \log n)$, where $n$ is the sample size, and $d$ is the dimension of the data. Hence, for sample size 1000, the methodology can be implemented for 6-7 dimensional data.
>
> Reviewer's Comment 2 : The algorithms that are used to implement these statistical tests are not written clearly. For example, the algorithm generates a large number of points over a high-dimensional unit sphere to approximate the supremum in the proposed test. However, it is not clear how large this number should be. In fact, I am not convinced that using points over a unit sphere can correctly approximate the supremum in the proposed test. Furthermore, the algorithm does not give precise criteria for when to reject the null hypothesis. Clearly, the choice of this criteria can significantly affect the empirical result.
>
> **Response to Comment 2** : We thank the reviewer for giving an opportunity to explain this delicate mathematical issue. First of all, in view of topological equivalence (strictly speaking, isomorphism) between a compact space and a unit sphere, supremum over unit sphere will coincide with the supremum over a compact space. In fact, it is true in separable Hilber space as well (in that case, compact space will be replaced by normed bounded space). Kindly see Proposition 2.2 and its preceding paragraph in https://arxiv.org/abs/2301.00375.
>
> The size of the number of points depends on the dimension of the sphere. Our empirical study indicates that 500 points work reasonably well as long as the dimension is less than or equal to 6.
>
> Regarding the criteria for rejecting null hypothesis, when the $p$-value is less than the preassigned level of significance of the test (e.g., 5\%), the null hypothesis is rejected.
>
> Reviewer's Comment 3 : The writing of this paper can be improved. For example, the main theorem of this paper involves 4 assumptions. But these four assumptions are not even mentioned in the main body of the paper.
>
> **Response to Comment 3** : The reviewer's concern is much appreciated here. On December 22, 2025, we explained how the assumptions are used in the results in responding to requested changes (2). If we have an opportunity, we will be happy to move the conditions in the main body of the article. Moreover, the practicability of the assumptions are explained in the Response to Comment 1 (Reviewer eUbG) on January 1, 2026.

---

> ### Author Response · Authors · 2026-01-02
> **Response to reviewer's other comments related to "Are the claims made in the submission supported by accurate, convincing and clear evidence?"**
>
> Reviewer's comment 1: I am not quite sure if the hypothesis testing problem is formulated properly in this paper. Usually, when we formulate a hypothesis testing problem, we need some "gap" (usually the TV distance) between the null hypothesis and the alternative hypothesis to make the problem solvable information theoretically. However, in this paper, there is no such gap assumption. So, given two distributions with TV distance $\epsilon$, it is not very clear to me whether we would need an unbounded number of examples to make the test work in the worst case.
>
> **Response to Comment 1** : The reviewer's concern is much appreciated here. We agree with the reviewer to a some extent that there should be a gap between the null and alternative when the observations are non-Euclidean  in nature (e.g., functional data) because comparing two non-Euclidean objects pointwise may not make proper sense (kindly see https://doi.org/10.1214/19-AOS1811). However, for Euclidean valued observations/data, there is no as such need of gap between the null and alternative as the dimension of the data is finite. As we did, similar formulations of the hypothesis for Euclidean valued random vectors have been studied for a long time in the literature. Kindly see https://doi.org/10.1007/s11222-023-103330, https://doi.org/10.1016/j.csda.2009.04.004,  https://doi.org/10.3150/13-BEJ530, https://doi.org/10.1016/j.csda.2022.107655 and many more articles.
>
> Since the whole work here depends on the fact that the observations are Euclidean valued, hence, we followed the same principle as we discussed in the previous paragraph.
>
> Reviewer's Comment 2: In the proposed test, one needs to compute the supremum of the difference of the halfspace depths of the two distributions over $R^d$. In the implementation of the test, the paper proposes that one could draw a large number of examples over the unit sphere to approximate this supremum. It seems to me that this method should not always be a correct approximation. For example, suppose $F$ and $F_0$ both have a support on some convex body that is disjoint from the unit sphere, then one cannot use points from the unit sphere to approximate the supremum over $R^d$, because for every point in the unit sphere, there would be a halfspace contains that point but contains no points from $F$ or $F_0$
>
> **Response to Comment 2** : We appreciate the reviewer's concern. We, however, feel that there is some kind of misunderstanding. The example the reviewer has presented, note that if $F$ and $F_{0}$ have disjoint support, that means they are entirely distinct distribution (i.e.., no overlap), and hence,  the supremum of the difference of the halfspace depths of those two distributions will be maximum, i.e., either $HD_{F}({\bf x}) = 0$ or $HD_{F_{0}}({\bf x}) = 0$. Further, observe that here ${\bf x}$ is the point of evaluation of the half space depth, and when we are transforming $||{\bf x}|| \leq R$ (for some $R> 0$) to  $||{\bf x}|| \leq 1$, the random vector associated with the distribution $F$ or $F_{0}$ is also transformed by the scale factor $R$. Hence, after transformation, both convex bodies (i.e., supports of $F$ and $F_{0}$) will overlap with the unit sphere. For details, kindly see Proposition 2.2 and its preceding paragraph in https://arxiv.org/abs/2301.00375.
>
> Reviewer's Comment 3 : Some of the proofs in the paper are not formal enough. For example, the proof of Theorem 3 only contains one sentence
>
> "By the similar argument of Theorem 3, using the fact...".
>
> This is clearly a typo. Furthermore, in the proof, it is unclear to me where the assumptions C1-C4 are used and where the assumption $\lambda \in (0,1)$ is used.
>
> **Response to Comment 3** : First of all, we are sorry for this typographical error. It should be "By the similar argument of Theorem 2, using the fact...."
>
> We thank the reviewer for raising this important mathematical issue. As we responded to "requested changes"  (point number 2),  the characterization of distribution by half-space depth (see Proposition 1) requires assumption C1-C4.
>
> The assumption $\lambda\in (0, 1)$ is used since only under this condition, $(D_{\cal{X}}({\bf x}) - D_{\cal{Y}}(\bf x))$ (after appropriate normalization) will weakly converge to a non-degenerate random element, which is essential for having limiting power $= 1$. It is a common assumption in any two-sample non-parametric testing of hypothesis problems. For example, kindly see Section 15. 5 in  https://www.cambridge.org/core/books/asymptotic-statistics/A3C7DAD3F7E66A1FA60E9C8FE132EE1D or Section 6 in the classical text book https://link.springer.com/book/10.1007/b98855 or Theorems 2.2 and 6.2 in https://doi.org/10.3150/13-BEJ530.

---

> > ### Comment · Editors_In_Chief · 2026-01-06
> >
> > The reference https://doi.org/10.1007/s11222-023-103330 does not exist.
> > Could you provide the full references rather than just a (broken) hyperlink?

---

> > > ### Author Response · Authors · 2026-01-06
> > > **Response to Editors In Chief**
> > >
> > > Thank you for requesting the full reference. The full reference is https://doi.org/10.1007/s11222-023-10333-0

---

> ### Comment · Reviewer_vhUf · 2026-01-10
>
> I would like to thank the authors for addressing my concerns here. I am convinced by the response to comment 1 and comment 3. However, I am still not fully convinced by the response to comment 2. In my proposed example, I did not say the supports of $F$ and $F_0$ are disjoint. In my example, the support of $F$ and $F_0$ can be the same, but their support can be disjoint from the unit sphere. Notice that in the algorithm, one evaluates halfspace depth at points sampled uniformly from the unit sphere. Since the supports of $F$ and $F_0$ are disjoint from the unit sphere, I am not sure if points from the unit sphere are representative enough to approximate the sup of the differences of halfspace depth over $R^d$. I would like to get more clarification about this question.
>
> Furthermore, even if some points over the unit sphere can represent the sup of the differences of the halfspace depth, it is still not quite clear to me whether uniformly sampling over the sphere is enough. Is this just motivated by the empirical studies? If $F$ and $F_0$ put much mass over some low-dimensional region, then uniformly sampling over the sphere does not look that reasonable. Do I misunderstand here?

---

> ### Author Response · Authors · 2026-01-10
> **Response to Reviewer's comment on January 10, 2026**
>
> First of all, we would like to thank the reviewer for giving us further opportunity to explain the issue related to computing supremum in the KS type test statistic. Observe that we mainly interested to compute $\sup_{{\bf x}\in\mathbb{R}^{d}}|D_{F}({\bf x}) - D_{F_{0}}({\bf x})|$ (excluding normalization constant $\sqrt{n}$). We first approximate  $\sup_{{\bf x}\in\mathbb{R}^{d}}|D_{F}({\bf x}) - D_{F_{0}}({\bf x})|$ by $\sup_{||{\bf x}|| \leq R}|D_{F}({\bf x}) - D_{F_{0}}({\bf x})|$, where $R > 0$ is sufficiently large. In this step, we wanted to convey that the supremum over an unbounded set is approximated by a supremum over a large bounded set, and it makes sense since $R> 0$ can be arbitrarily large. Now, note that in view of Theorem 3.3 in https://doi.org/10.1214/17-AOS1658  or Section 2.2 in https://doi.org/10.1002/wics.70038, $D_{F}(.)$ and $D_{F_{0}}(.)$ are quasi concave functions with respect to (.), and in particular, they will be concave functions for elliptically symmetric distributions (e.g., multivariate Gaussian, multivariate Cauchy and many more) in view of https://doi.org/10.1214/17-AOS1658 (see Section 2, page 3281). Hence, $|D_{F}({\bf x}) - D_{F_{0}}({\bf x})|$ will be a convex function with respect to ${\bf x}$.
>
> Now, due to Bauer maximum principle (which states that a convex, upper semi-continuous function defined on a compact, convex set attains its maximum value at an extreme point of that set; see https://doi.org/10.1007/BF01898615 or see Theorem 1 in https://doi.org/10.1007/s005260000047), we have $$\sup_{||{\bf x}|| \leq R}|D_{F}({\bf x}) - D_{F_{0}}({\bf x})| = \sup_{||{\bf x}|| = R}|D_{F}({\bf x}) - D_{F_{0}}({\bf x})|.$$
>
> Finally, due to equivariance of half space depth under arbitrary transformation (see Theorem 2.1
> in https://doi.org/10.1214/aos/1016218226), we have $$\sup_{||{\bf x}|| = R}|D_{F}({\bf x}) - D_{F_{0}}({\bf x})| = \sup_{||{\bf x}|| = 1}|D_{F}({\bf x}) - D_{F_{0}}({\bf x})|.$$ This is the reason for which to compute the half space depth based KS statistic, the points are sampled from the boundary of the unit ball.
>
> Regarding the reviewer's example, suppose that $F$ and $F_{0}$ are supported on $[-3, -2]$, which is disjoint from $[- 1, 1]$ (considering one dimension case for ease of notation). Our idea is that let us first optimise the function over $[- 3, 3]$, and due to affine equivariance of half space depth, it will be the same as the optimization over $[-1, 1]$. This is what we have done in this work.
>
> Now, why we did generate the points uniformly, the reason is the following. Note that for computational purpose, we wanted to have as many as points from the set $\lbrace\bf x\in\mathbb{R}^{d} : ||{\bf x }|| = 1\rbrace$, and if we generate the observations uniformly on $\lbrace{\bf x}\in\mathbb{R}^{d} : ||{\bf x }|| = 1\rbrace$, as the number of observations tends to infinity, it will be dense in $\lbrace{\bf x}\in\mathbb{R}^{d}: ||{\bf x }|| = 1\rbrace$. Although, we must admit that the one can achieve some other way to have dense observations on $\lbrace{\bf x}\in\mathbb{R}^{d} : ||{\bf x }|| = 1\rbrace$ (for example, one may consider a sequence of rational numbers (component-wise), whose norm equals one).
>
> In conclusion, we would like to mention that the idea is motivated by the all aforementioned theoretical arguments, and then we implemented it in the empirical studies. Overall, it works well.
>
> If the reviewer has any further comments on this issue, we will be happy to address them.

---

### Review · Reviewer_eUbG · 2025-11-30

**Summary Of Contributions:**

The paper studies how to test whether two multivariate distributions are the same. The authors define a data-depth discrepancy (DDD) based on Tukey’s half-space depth, which is the difference between the depth functions of two distributions and can be shown in a two-dimensional plot.

Building on this DDD, the paper introduces depth-based Kolmogorov-Smirnov and Cramér-von Mises test statistics for both the one-sample and two-sample settings. The authors prove consistency of the proposed tests and derive their asymptotic distributions under contiguous alternatives, which allows them to study local asymptotic power. In their simulations, they report finite-sample level and power studies under several synthetic models as well as real data sets.

**Audience:**

Yes

**Audience Explanation:**

Hypothesis testing for multivariate distributions is closely related to several areas in machine learning, such as two-sample testing, evaluation of generative models, and detecting distribution shift. In particular, researchers who care about statistical foundations of machine learning problems may be interested in the proposed depth-based tests and graphical tools.

**Broader Impact Concerns:**

I do not see any significant broader impact concerns.

**Claims And Evidence:**

No

**Claims Explanation:**

The claims made in the submission are partly supported both by theory and by experiments, but there are some issues that the authors should address.

On the theoretical side, the paper proves consistency and local asymptotic power results for the proposed depth-based KS and CvM tests, under conditions (C1)-(C4). The theoretical conditions are clearly formulated but quite technical, so adding some examples when they are likely to be satisfied in practice could make the results more accessible.

On the empirical side, the finite sample level and power are studied through simulations under several models. However, there is one case (Model A.3 in Table 3) that shows something counter. The empirical power is decreasing in Model A.3 in Table 3 by sample size but the paper does not discuss this abnormal result. Also, there are some scenarios that all the power equal to 1, which makes it harder to compare methods in more challenging regimes.

So in summary, the evidence is generally consistent with the claims, but it is better to have a clearer discussion of the assumptions and a more careful analysis of the experiments.

**Requested Changes:**

(i) Please check and discuss the strange behavior in Model A.3 of Table 3. This result should be checked for possible issues in the writing or for Monte Carlo error. At present, this abnormal result is not discussed.

(ii) It would be better to make the experimental design more informative for power comparison. In several scenarios (for example, Model A.5), the observed relative frequency of rejecting the null is essentially 1.000 even at the smallest sample sizes.

(iii) It would be very helpful if the authors could explain more clearly why half-space depth is a particularly suitable choice in this context, and it would be better to compare DDD with some other metrics or discrepancies, for example, MMD with a characteristic kernel, energy distance, or certain probability metrics.

(iv) Also consider connecting to modern models. Since the current experiments are mostly for classical statistical distributions, it would be better if the authors could add more connections to some modern models or applications, for example, to evaluate generative models or to perform OOD detection.

---

> ### Author Response · Authors · 2025-12-22
> **Response to "Requested changes" is provided in the comment below. We thank the reviewer for the valuable suggestions.**
>
> **(i)** Please check and discuss the strange behavior in Model A.3 of Table 3. This result should be checked for possible issues in the writing or for Monte Carlo error. At present, this abnormal result is not discussed.
>
> **Response:** We apologise for the inadvertent error. The power values for Model A.3 in Table 3 should be in reverse order. If given the opportunity, we will correct it in the revised manuscript.
>
> **(ii)** It would be better to make the experimental design more informative for power comparison. In several scenarios (for example, Model A.5), the observed relative frequency of rejecting the null is essentially 1.000 even at the smallest sample sizes.
>
> **Response:** In this context, we would like to emphasise that when the distribution described in the model is a heavy-tailed distribution like the multivariate Cauchy distribution (for example, Model 5), our test performs very well in view of Proposition 1, as the heavy-tailed multivariate Cauchy distribution and light-tailed multivariate Gaussian distribution are far apart in terms of tail behaviour. Consequently, even for a small sample size, the empirical power of the test (i.e., the relative frequency of rejecting the null) approaches the maximum possible value, i.e., one.
>
>
> **(iii)** It would be very helpful if the authors could explain more clearly why half-space depth is a particularly suitable choice in this context, and it would be better to compare DDD with some other metrics or discrepancies, for example, MMD with a characteristic kernel, energy distance, or certain probability metrics.
>
> **Response:** The half-space depth is a particularly suitable choice since it characterises the distribution (see Proposition 1). Technically speaking, suppose that $HD_{A} (.)$ denotes the half space depth of a random vector $A$, then  $HD_{A} (x) = HD_{B}(x)$ for all $x$ if and only if $A = B$ in distribution (see the discussion after equation number (1)). Moreover, the proposed data depth discrepancy (DDD) differs fundamentally from metrics such as MMD, energy distance, or Wasserstein-type distances. Those metrics summarise global differences between two distributions into a single scalar and typically require kernel or metric specification. In contrast, DDD yields a localised, index-wise discrepancy, which is particularly suitable for identifying where and how the distributions differ, rather than only whether they differ. This feature is crucial for our scientific goal of detecting structured and heterogeneous departures across the sample index. Moreover, unlike MMD with characteristic kernels, which tests equality in distribution but offers limited interpretability at the individual data-point level, DDD directly reflects deviations in data depth relative to the empirical reference distribution. This makes the resulting test statistics easier to interpret and visualise. We agree that comparisons with alternative discrepancy measures are valuable.
>
> **(iv)** Also consider connecting to modern models. Since the current experiments are mostly for classical statistical distributions, it would be better if the authors could add more connections to some modern models or applications, for example, to evaluate generative models or to perform OOD detection.
>
> **Response:** Note that for object-oriented design (OOD), the objects are not Euclidean valued, whereas the fundamental assumption in this work is that the random variables are Euclidean valued random vectors, and hence, directly, this method cannot be applied to object data. In fact, strictly speaking, it is still theoretically unknown in the literature whether the half-space depth for non-Euclidean random elements characterises the measure or not. Unless the characterisation property holds (as in Proposition 1), the result cannot be used for object data.

---

> > ### Comment · Reviewer_eUbG · 2026-01-06
> > **Response to Authors**
> >
> > Thank you again for your detailed responses.
> >
> > (1) In my previous review, “OOD” was meant to refer to out-of-distribution detection. It would be helpful if the paper could briefly discuss such potential applications to make the connection to modern ML problems clearer.
> >
> > (2) I appreciate the conceptual discussion on how DDD differs from metrics like MMD, energy distance, or Wasserstein distances. However, it is still difficult to judge when one should prefer DDD-based tests over standard kernel or distance-based two-sample tests. I would strongly encourage the authors, if space permits, to include at least one comparison with a widely used baseline (e.g., an MMD-based two-sample test with a characteristic kernel, or an energy-distance test) on the same simulation models.
> >
> > (3) For cases where the empirical power is very close to 1, I believe this is mainly an issue of experimental design rather than post-hoc interpretation. It is the responsibility of the experimenter to choose additional models or parameter settings where the tests have more intermediate power, so that differences between methods become visible.

---

> > > ### Author Response · Authors · 2026-01-07
> > > **Response to Reviewer's comment on January 07, 2026**
> > >
> > > Response to Comment 1 : We thank the reviewer for raising an important issue. In the context of  out-of-distribution detection, i.e, protection against the outliers, we can say that the half space depth based methodologies mostly work well for the data having outliers. For example, the half space based location estimator's breakdown point can achieve $\frac{1}{3}$ (in the limiting sense of the sample size of the data; kindly see Propositions 3.2 and 3.3 in https://doi.org/10.1214/aos/1176348890), and this large value of breakdown point indicates that the graphical toolkit and tests based on half space depth proposed have good performance against the outliers. In fact, this fact is visible in the performance in the simulation studies for Models A.4 and A.5, when the data are generated from heavy tailed distributions.
> > >
> > > Response to Comment 2 : The reviewer's suggestion is much appreciated here. As the reviewer advised, we will definitely add at least one comparison study with MMD-based test or energy distance test in the revised version of the manuscript.
> > >
> > > Response to Comment 3 : We really appreciate the reviewer's view. In the revised manuscript, we will choose some model so that some intermediate values of the tests can be obtained, and consequently, the performance of the test will be more easy to compare.

---

> > > > ### Comment · Reviewer_eUbG · 2026-01-10
> > > > **Response to Authors**
> > > >
> > > > Thank you for your follow-up responses and for taking these suggestions seriously. These planned changes will clearly strengthen the paper if implemented carefully.

---

> ### Author Response · Authors · 2026-01-01
> **Response to reviewer's other comments related to "Are the claims made in the submission supported by accurate, convincing and clear evidence?"**
>
> Reviewer's Comment 1 : On the theoretical side, the paper proves consistency and local asymptotic power results for the proposed depth-based KS and CvM tests, under conditions (C1)-(C4). The theoretical conditions are clearly formulated but quite technical, so adding some examples when they are likely to be satisfied in practice could make the results more accessible.
>
> **Response to Comment 1**: We thank the reviewer for giving us the opportunity to explain the conditions (C1)-(C4). Note that when $F$ is an absolutely continuous distribution (e.g., Gaussian distribution), (C1) will be satisfied. Kindly see Section 2 in  https://doi.org/10.3150/bj/1089206404. Next, (C2) is a well-known condition for  empirical central limit theorems (see Chapter 5 in https://link.springer.com/book/10.1007/978-1-4612-5254-2). In this context, observe that the class of Gaussian distribution indexed by mean parameter, when the variance is fixed, satisfies (C2). Thereafter, (C3) implies the uniqueness of the halfspace depth at a certain point, and (C4) is satisfied by a wide range of distributions like multivariate Gaussian distribution, multivariate Cauchy distributions etc.
>
> Overall, Conditions (C1)-(C4) are common across the literature of statistical and ML methodologies based on halfspace depth, and if the data obtained from absolutely continuous distributions like Gaussian distributions, Conditions (C1)-(C4) will be satisfied, and hence, it has wide practical adaptability.
>
> Reviewer's Comment 2 : On the empirical side, the finite sample level and power are studied through simulations under several models. However, there is one case (Model A.3 in Table 3) that shows something counter. The empirical power is decreasing in Model A.3 in Table 3 by sample size but the paper does not discuss this abnormal result. Also, there are some scenarios that all the power equal to 1, which makes it harder to compare methods in more challenging regimes.
>
> **Response to Comment 2** : As we replied on December 22, 2025 (Response to Requested Changes (i)), we apologise for the inadvertent error. The power values for Model A.3 in Table 3 should be in reverse order. If given the opportunity, we will correct it in the revised manuscript.
>
> Regarding other point about the difficulty in comparing the tests, we would like to say that when the null and alternative are far apart in tail behaviour (e.g., multivariate Gaussian against multivariate Cauchy), any reasonable test is supposed to have discriminating ability to distinguish both distributions. For that reason, for a few cases, all tests perform well, i.e., they achieve the maximum power one.
>
> In order to resolve this issue, one may adopt an advanced technique from theory. As we replied to requested changes (2) of the Reviewer Qn4x on December 22, 2025, one may consider **local alternatives** instead of **fixed alternatives**, and we did it for the KS and the CVM tests in this article. However, since the asymptotic distributions of most of the tests **under local alternatives** are **unknown** in the literature, this exercise may not be doable in this work.

---

### Review · Reviewer_Qn4x · 2025-12-10

**Summary Of Contributions:**

**Summary** \
This paper studies the standard hypothesis testing problem of testing goodness of fit of a dataset with respect to a known base distribution, and also between two datasets. Namely, consider Problem 1: suppose we have a dataset $\mathcal{X}=(X_1,\dots,X_n)$ where each $X_i \in \mathbb{R}^d$, and each $X_i$ is drawn i.i.d. from an unknown distribution $F$. We want to test the null hypothesis that $F=F_0$ for some known base distribution $F_0$, versus the alternative hypothesis that $F \neq F_0$. Similarly, consider Problem 2: suppose we have two datasets $\mathcal{X}=(X_1,\dots,X_n)$ and  $\mathcal{Y}=(Y_1,\dots,Y_m)$, where each $X_i$ and $Y_i$ is in $\mathbb{R}^d$, each $X_i$ is drawn i.i.d. from some unknown distribution $F$ and each $Y_i$ is drawn i.i.d. from some unknown distribution $G$. We want to test the null hypothesis $F=G$ versus the alternative hypothesis that $F \neq G$.

The central quantity of interest in this paper for this task is the "data depth", and in particular, the Tukey depth of any point w.r.t. the distribution/dataset. More precisely, for any distribution $F$ and point $x$, consider the depth function $D_F(x)=\inf_{H} (P(H):x \in H)$, where the infimum is taken over all closed halfspaces $H$ that contain $x$. The empirical version of this quantity that can be computed from a dataset $\mathcal{X}=(X_1,\dots,X_n)$ is $D_{\mathcal{X}}(x)=\inf_{H} \frac{1}{n}\sum_{i=1}^n 1[X_i \in H]$. It appears that many statistical software libraries have direct support for computing this quantity. Furthermore, as the dataset size $n$ gets large, this empirical quantity accurately approximates $D_F(x)$. With this definition established, the main reason for considering this quantity for the purposes of Problems 1 and 2 is a prior result (stated as Proposition 1 in the paper), which says that $D_F(x)=D_G(x)$ for all $x \in \mathbb{R}^d$ iff $F=G$, under certain technical conditions.

So, in order to solve Problem 1, one can compare $D_{F_0}(x)$ with $D_\mathcal{X}(x)$ at a test set of points $x$ --- if these two quantities are close by uniformly, then this confirms the null that the dataset was indeed drawn from $F_0$ (i.e, $F=F_0$); otherwise, if there is significant deviation at any $x$, this would suggest the alternative hypothesis. Similarly, for Problem 2, one can compare $D_{\mathcal{X}}(x)$ with $D_{\mathcal{Y}}(x)$ at a test set of points $x$, and perform the same test. The way "comparison" is performed could be either in an $\ell_{\infty}$ sense over $x$, or an $\ell_2$ sense---this corresponds to either the Kolmogorov–Smirnov (KS) test or the  Cramér–von Mises (CvM) test. The formal details of these tests are given in Section 3.2.

The authors also comment about how this test offers a convenient "graphical tool-kit"---namely, one can scatter plot the difference $D_{F_0}(x)-D_{\mathcal{X}}(x)$ against $i$ (the index of the test point) for a test set $(x_1,\dots,x_n)$. If the points are very close to the $y=0$ horizontal axis, this supports the null, as opposed to a plot that has points far away from the $x$ axis. This is always a simple 2d plot irrespective of the dimension of the data.

Next, the authors derive asymptotic results about the test in Section 4, again under suitable technical conditions. Theorems 2 and 3 roughly establish that as $n$ gets large, the scatter plot converges to the horizontal axis iff the null holds. Theorems 5 and 6 establish that the asymptotic power of these tests is optimal. These results were for the case when the alternative hypothesis was the complement of the null. Section 4.2 establishes asymptotic results for when the alternative hypothesis is "local"/"contiguous".

The authors then perform empirical studies to validate the performance of the proposed test. Section 5 has goodness of fit experiments on synthetic data, where the base distribution is either normal/Cauchy/Laplace or a mixture of these. The authors compare the observed relative frequency of rejecting the null for their test, as compared to a variety of other tests. The qualitative summary of the numbers seems to be that the test proposed by the authors consistently has better performance.

Finally, the authors consider a couple real datasets: the Iris dataset and gilgais dataset. The objective is to check if the covariates in these dataset are drawn from a multivariate normal distribution. While the benchmark to validate the results here is unclear, from their proposed test, the authors conclude that the Iris data is multivariate normal, whereas the gilgais data is not.

**Strengths**\
The studied quantity, namely data depth, is intuitive and standard. The corresponding test based on data depth discrepancy is natural and also backed by theoretical evidence due to Proposition 1. The asymptotic theoretical results are relevant and appear sound. The "graphability" of the test seems to also be a convenient plus point. The empirical evidence provided, at least on the synthetic data, seems convincing.

**Weaknesses**\
The computational efficiency of the test has not been discussed properly. The significance of some of the theoretical results, especially the ones about local/contiguous alternatives, is unclear to me. It is also unclear to me why "index of the test point" is the correct/most reasonable x axis to plot the data depth discrepancy. Some other choices, like choosing the test points $x$ to also be the points from the dataset $\mathcal{X}$ itself (which was also use to empirical estimate $D_{F}(\cdot)$) have not been properly discussed/justified. The authors don't discuss a benchmark to validate the result of their test on the real datasets. Please see the recommended changes for more details.

**Audience:**

Yes

**Audience Explanation:**

Given that the problems studied in this paper are very foundational and basic, I believe the statistics and property testing communities will definitely be interested in the study.

**Claims And Evidence:**

Yes

**Claims Explanation:**

**With the exception of the results on the Iris and gilgais datasets, for which I do not know whether to believe that the conclusion of the proposed test is accurate.

**Requested Changes:**

Could you please respond to the following questions/comments that I have? It would greatly inform my overall evaluation.

1) One of the main selling points of the paper according to the authors is that of providing a "graphical tool-kit". Here, the authors plot the data depth discrepancy against the index of the test point $x$ in the data set. Could you comment a bit more on why “index of data point” is the most natural x-axis to consider? In particular, why do you substitute points in the dataset itself (which were used to also empirically estimate $D_F$) as the test points to compute data depth discrepancy? Why do the test points need to be from the same distribution as the dataset even? Couldn't you compute this for any $x$? In particular, it seems like a more robust and better methodology is to consider several different test sets, constructed by e.g., drawing from different distributions, independent of the dataset $\mathcal{X}$.

2) With regards to Section 4.2, what is the significance of studying local/contiguous alternative hypotheses? The authors claim in the conclusion that this is "an important theoretical step that is often overlooked”. It would be worth discussing this and elaborating on it. Also, is my understanding correct that in Section 4.1, you simply consider the alternate hypothesis $F \neq F_0$, whereas in 4.2, you consider more structured alternate hypotheses? Namely, the testing problem in 4.2 is harder, in that solving the testing problem in 4.2 implies solving that in 4.1, but not vice versa? Maybe this is worth clarifying

3) The authors mention that there is convenient software support to implement the proposed tests, in the form of direct packages in standard statistical libraries. However, what is the computational cost of implementing the test? In particular, how expensive is it to compute $D_{\mathcal{X}}(x)$ on a dataset $\mathcal{X}$ of size $n$, or $D_{F}(x)$ itself? The authors don't discuss this at all in the paper, and it deserves to be properly discussed. In particular, what is the state-of-the-art method to compute the exact/approximate Tukey depth of a point $x$ within an $n$-point dataset?

4) Following up on the above point: the authors should definitely include the running times of the proposed test in comparison to all the other tests, for example, in the empirical results in Section 5. As of now, the reader gets no sense of how the running time of the different tests compare.

5) For the results on the real datasets: the authors claim that the result of their proposed test implies the iris data is bivariate normal, while the giglais data isn't: how is the reader supposed to infer if these conclusions are valid/accurate? Is it generally known that the Iris data is normal, whereas the giglais data is not? In the absence of accepted benchmarks/wisdom, it is a little hard to associate any meaning/correctness with the result of the proposed test on these datasets. For example, it would be worth seeing what the other tests have to say on these datasets.

6) In the tables in Section 5, in each row, please format the the best numbers to be in bold. This greatly helps the reader to see how/if the numbers of the proposed test are better than the other tests. It would also be useful to include pointers like "larger is better" / "smaller is better" in the captions.

Other Minor points:

a) I would recommend the authors to bulletize/itemize the individual contributions in Section 1.1 for better readability.

b) Typo in Section 2: I believe in the definition of $\hat{S}_n(x)$, the $x^{-}$ should simply be $x$.

c) Throughout the paper, whenever the authors reference equations, the word "equation" gets repeated twice.

---

> ### Author Response · Authors · 2025-12-22
> **Response to the reviewer is provided in the comment below. We thank the reviewer for the valuable suggestions.**
>
> 1. Using “index of the data point” $i$ as x-axis is the most appropriate choice for the graphical displays in our article. Data depth discrepancy is calculated for each $i$, and plotting it against $i$ enables a clear assessment of whether the discrepancy deviates from zero across datapoints. Each discrepancy corresponds to specific datapoint $X_{i}$, plotting against $i$ provides direct visualization of how the discrepancy varies across the sample. We use the observed datapoints as test points because the objective is to assess discrepancies within the support of the underlying data-generating distribution. Evaluating data depth at points drawn from the same distribution ensures that depth values are meaningful, stable, and comparable, since data depth is inherently a relative notion defined with respect to the empirical distribution. While data depth discrepancy can, in principle, be evaluated at arbitrary test points, including those from external distributions, such choices may yield unstable or uninformative depth values, especially in regions with little data support. This can obscure interpretation and introduce variability unrelated to the hypothesis. Our approach follows standard depth-based inference practice by evaluating discrepancies at the observed sample points.
>
> 2. The main significance of the local/contiguous alternatives is associated with finding the optimal test among the consistent tests. For example, suppose that there are two tests based on $T_n$ and $V_n$, both are consistent, i.e., the power of both tests based on $T_n$ and $V_n$ converge to one (maximum possible value) when the sample size is infinity for **fixed** alternatives. Then the natural question would be which test is better among these two tests. To address this issue, the idea of **local** alternatives came to the literature. The local alternatives should be such that they are not the same as the null distribution but converge to the null as the sample size converges to infinity (see equation number (2)). Therefore, in other words, it will give us the power of the test in the **local neighborhood** of the null distribution, whereas the **fixed** alternatives provide us the power of the test when the **distance** between the alternative and null distributions remains **fixed/constant** (does not depend on the sample size $n$). In conclusion to the article, we wanted to convey that the power property of the test under **local** alternatives was often overlooked. Further, note that the volume of the **neighborhood** depends on the choice of $\gamma$ (see equation number (2)). Moreover, for $\gamma = 0$, we have $F_n = F_0$ for all $n$, and for $\gamma = \sqrt{n}$, we have $F_{n} = H$ for all $n$ (see equation number (2)). Therefore, in this sense, as the reviewer guessed, the work in Section 4.1 (i.e., under **fixed** alternative) is a special case of the work in Section 4.2 (i.e., under **local** alternatives) but not vice-versa.
>
> 3. The computational efficiency of the proposed test depends on the computational algorithm of the half-space depth. Computation of data-depth is not the main objective of our article. However, there are a few algorithms available in the literature to compute the half-space depth for high-dimensional data (see the last two lines in Section 7). The worst-case complexity is $O(n^{d−1}logn)$. There are a few algorithms, for example, **Shao and Zuo (2020)** (see proposition 2 therein), computational complexity of MTMSAC algorithm is $O(Nmknd)$ and for the MTMSAD algorithm has computation complexity $O(Nmknd + Nmkd^{3})$ where where $N$ is the length of decreasing cooling temperature, m is the length of the Markov chain, $k$ is the number of multiple try points.
>
> Reference :  https://doi.org/10.1007/s00180-019-00906-x
>
> 4. Since the competing tests do not require the computation of data depth, the running times are not directly comparable. Nevertheless, we will report the running times of the different testing procedures in the revised manuscript.
>
> 5. We believe the conclusions are valid, since both the theoretical and simulation results demonstrate that the proposed test has well-controlled size and adequate power over a range of settings. This ensures that the resulting decisions are statistically reliable. Moreover, if given the chance to revise, we will add the decisions from the benchmark tests considered in the simulation study for the real data analysis in the revised manuscript.
>
> 6. We agree that highlighting the best values in bold improves readability. Moreover, we will add in the revised manuscript “computation of size” and “computation of power” to facilitate comparison with the expected results.
>
> **Minor comments:**
>
> a.We have incorporated this into the revised version of the manuscript.
>
> b. We have corrected this in the revised version of the manuscript.
>
> c. This was a typographical error arising from LaTeX formatting, and it has been corrected in the revised version of the manuscript.

---

> ### Author Response · Authors · 2026-01-01
> **Response to reviewer's other comments related to "Are the claims made in the submission supported by accurate, convincing and clear evidence?"**
>
> Reviewer's comment : With the exception of the results on the Iris and gilgais datasets, for which I do not know whether to believe that the conclusion of the proposed test is accurate.
>
> **Response** : We thank the reviewer for raising this issue. For Iris data, many other tests also conclude that the sepal length and the sepal width of the Iris species follow multivariate normal distribution, after appropriate normalization. Kindly see Section 4.2 in the reference : https://doi.org/10.3150/13-BEJ530
>
> Regarding gilgais data, it is also well-known fact that the (pH Variables, Electrical Conductivity, Chloride Content) jointly do not follow trivariate normal distribution. Kindly see https://link.springer.com/book/10.1007/978-0-387-21706-2 If the book is not available, it is possible to check using the following simple R-code related to well-known Shapiro-Wilk test on gilgais data
>
> library(MASS)\\\\
> data(gilgais)\\\\
> qqnorm(gilgais pH00); qqline(gilgais pH00)\\\\
> shapiro.test(gilgais\$pH00)\\\\
>
> Regarding two-sample problems for gilgais data, the variables are entirely different in nature, and it is unlikely to have the same distributional feature, which is reflected in the results obtained by our test. If we have the opportunity to revise the paper, we can add a simple scatter plot to support the results.

---

> ### Author Response · Authors · 2026-01-01
> **Response to weakness comment of the Reviewer**
>
> Weakness : The computational efficiency of the test has not been discussed properly. The significance of some of the theoretical results, especially the ones about local/contiguous alternatives, is unclear to me. It is also unclear to me why "index of the test point" is the correct/most reasonable x axis to plot the data depth discrepancy. Some other choices, like choosing the test points $x$ to also be the points from the dataset $\mathcal{X}$ itself (which was also use to empirical estimate $D_{F}(\cdot)$) have not been properly discussed/justified. The authors don't discuss a benchmark to validate the result of their test on the real datasets. Please see the recommended changes for more details.
>
> **Response** : We thank the reviewer for giving us further opportunity to clarify a few important issues. Regarding computation efficiency, local alternatives, ``index of the test point", we replied on December 22, 2025. For the last part of the comment about implementing the test on real data sets, we would like to clarify that we implemented the tests on the real data sets, namely, Fisher’s Iris data and gilgais data. Kindly see Section 6 in the article.

---

> ### Comment · Reviewer_Qn4x · 2026-01-06
> **Response to authors**
>
> Thanks a lot for the detailed response.
> 1) I understand that the training dataset itself appears to be a canonical dataset to compare data depth discrepancy, but I am still not entirely convinced why that should be the best dataset for comparing this. The authors say that this is a "stable" and "meaningful" choice, but ultimately, it seems there isn't a formal reason that certifies it to be the "most informative" dataset as compared to some other test set. I believe it is worth clarifying in the paper that the choice of using the training data itself for the purposes of graphing the data depth discrepancy is more canonical/heuristic than principled.
> 2) Thanks for elaborating more on the significance of local alternatives. It would definitely be worth including this discussion in the revised manuscript.
> 3) Please also include the discussion on the computational efficiency of computing data depth in the paper.
> 4) "Since the competing tests do not require the computation of data depth, the running times are not directly comparable." I disagree with this. The point is not about comparing different statistical tests that all use data depth, but simply about comparing the computational power required by different statistical tests. Given that you claim via your experiments that your proposed test outperforms other statistical tests in statistical power, it is natural to study the computational tradeoffs. For example, one would want to know if the performance improvement in the statistical power arises as a result of a running time blowup of orders of magnitude, which would decrease the merit of using the proposed method. For this reason, I believe it is essential to include the running times in the tables.
> 5) Thank you for providing the references that support the results of your test on the real datasets. Again, this discussion should definitely be included in the manuscript.
> 6) I am not sure if the changes for making the best numbers bold were already done, but this would make the tables more readable for sure.

---

> ### Author Response · Authors · 2026-01-07
> **Response to Reviewer's comment on January 07, 2026**
>
> Response to Comment 1 : We agree with the reviewer's view to some extent, and if we dig in, it is possible to conclude that the reviewer's view and considering the data set itself as the training data set have only negligible difference when the size of the data set is large enough. To  explain it, consider $[0, 1]$ on x-axis and consider a **discrete** random variable $X$ such that $P[X = \frac{i}{n}] = \frac{1}{n}$ for $i = 1, \ldots, n$. Note that as $n\rightarrow\infty$, $X$ will weakly converge to a **continuous** random variable $Y$, where $Y$ follows uniform distribution over $[0, 1]$ (for the proof, kindly see Example 25.3 in the textbook https://www.google.co.in/books/edition/Probability_and_Measure/z39jQgAACAAJ?hl=en). In view of this fact, when the size of the data is sufficiently large, considering the index of the data set on the $X$-axix (i.e., horizontal axis) or taking any arbitrary observation $x$ on the $X$-axis has only negligible difference. Overall, in principle, the reviewer is correct since ideally one may consider arbitrary observation $x$ on the $X$-axis to get the best possible graphical toolkit from a theoretical point of view. However, from a practical/computational point of view, computing half space depth at the data point is less demanding than computing halfspace depth at arbitrary point. Therefore, combining all these aforementioned facts (i.e., limiting result as well as to avoid computational burden), we consider the data set itself as the training data set.
>
> Response to Comment 2 : We thank the reviewer for a nice suggestion. Following the reviewer's advice, we will definitely include this discussion in the revised manuscript.
>
> Response to Comment 3 : We again thank the reviewer for a nice suggestion. Following the reviewer's advice, we will definitely include this discussion in the revised manuscript.
>
> Response to Comment 4: We are extremely sorry for misunderstanding the comment of the reviewer in the previous round. We completely agree with the reviewer that the computational running time should be provided to have complete study. If we get an opportunity to revise the article, we will definitely incorporate the running time.
>
> Response to Comment 5 : We really appreciate the reviewer's suggestion, and we will definitely add the relevant discussion in the revised version of the manuscript.
>
> Response to Comment 6 : The reviewer's concern is much appreciated here. In the revised version of the manuscript, we will definitely make the best numbers bold.

---

### Comment · Action_Editor_XyvT · 2025-12-31

Hi Authors,

Hope you're enjoying the holidays.

I want to encourage you to respond to the reviews beyond the "Requested Changes" sections. TMLR's acceptance criteria are "whether the claims of the paper are supported by clear and convincing evidence" and "whether some parts of the community would be interested in the results". For the former criterion, reviewers raised a number of issues that are currently not addressed by the rebuttals.